
# The Acidity of Atmospheric Particles and Clouds

Havala O. T. Pye[1], Athanasios Nenes[2,3], Becky Alexander[4], Andrew P. Ault[5], Mary C. Barth[6], Simon L. Clegg[7], Jeffrey L. Collett, Jr.[8], Kathleen M. Fahey[1], Christopher J. Hennigan[9], Hartmut Herrmann[10], Maria Kanakidou[11], James T. Kelly[12], I-Ting Ku[8], V. Faye McNeill[13], Nicole Riemer[14], Thomas Schaefer[10], Guoliang Shi[15], Andreas Tilgner[10], John T. Walker[1], Tao Wang[16], Rodney Weber[17], Jia Xing[18], Rahul A. Zaveri[19], Andreas Zuend[20]

[1]Office of Research and Development, U.S. Environmental Protection Agency, Research Triangle Park, NC, 27711, USA
[2]School of Architecture, Civil and Environmental Engineering, Ecole Polytechnique Fédérale de Lausanne, Lausanne, CH-1015, Switzerland
[3]Institute for Chemical Engineering Sciences, Foundation for Research and Technology Hellas, Patras, GR-26504, Greece
[4]Department of Atmospheric Science, University of Washington, Seattle, WA, 98195, USA
[5]Department of Chemistry, University of Michigan, Ann Arbor, MI, 48109-1055, USA
[6]National Center for Atmospheric Research, Boulder, CO, 80307, USA
[7]School of Environmental Sciences, University of East Anglia, Norwich NR4 7TJ, UK
[8]Department of Atmospheric Science, Colorado State University, Fort Collins, CO, 80523, USA
[9]Department of Chemical, Biochemical, and Environmental Engineering, University of Maryland Baltimore County, Baltimore, MD, 21250, USA
[10]Leibniz Institute for Tropospheric Research (TROPOS), Atmospheric Chemistry Department (ACD), Leipzig, 04318, Germany
[11]Department of Chemistry, University of Crete, Voutes, Heraklion Crete, 71003, Greece
[12]Office of Air Quality Planning & Standards, U.S. Environmental Protection Agency, Research Triangle Park, NC, 27711, USA
[13]Department of Chemical Engineering, Columbia University, New York, NY, 10027, USA
[14]Department of Atmospheric Sciences, University of Illinois at Urbana-Champaign, Urbana-Champaign, Illinois, 61801, USA
[15]State Environmental Protection Key Laboratory of Urban Ambient Air Particulate Matter Pollution Prevention and Control, Nankai University, Tianjin, 300071, China
[16]Department of Civil and Environmental Engineering, The Hong Kong Polytechnic University, Hung Hom, Kowloon, Hong Kong, China
[17]School of Earth and Atmospheric Sciences, Georgia Institute of Technology, Atlanta, GA, 30332, USA
[18]School of Environment, Tsinghua University, Beijing, 100084, China
[19]Atmospheric Sciences & Global Change Division, Pacific Northwest National Laboratory, Richland, WA, 99352, USA
[20]Department of Atmospheric and Oceanic Sciences, McGill University, Montreal, Quebec, H3A 0B9, Canada

*Correspondence to*: Havala O. T. Pye (pye.havala@epa.gov)

**Abstract.** Acidity, defined as pH, is a central component of aqueous chemistry. In the atmosphere, the acidity of condensed phases (aerosol particles, cloud water, and fog droplets) governs the phase partitioning of semi-volatile gases such as $HNO_3$, $NH_3$, and $HCl$, as well as chemical reaction rates. It has implications for the atmospheric lifetime of pollutants, deposition, and human health. Despite its fundamental role in atmospheric processes, only recently has this field seen a growth in the number of studies on particle acidity. Even with this growth, many fine particle pH estimates must be based on thermodynamic model calculations since no operational techniques exist for direct measurements. Current information indicates acidic fine particles



are ubiquitous, but observationally-constrained pH estimates are limited in spatial and temporal coverage. Clouds and fogs are also generally acidic, but to a lesser degree than particles, and have a range of pH that is quite sensitive to anthropogenic emissions of sulfur and nitrogen oxides, as well as ambient ammonia. Historical measurements indicate that cloud and fog droplet pH has changed in recent decades in response to controls on anthropogenic emissions, while the limited trend data for

aerosol particles indicates acidity may be relatively constant due to the semi-volatile nature of the key acids and bases and buffering in particles. This paper reviews and synthesizes the current state of knowledge on the acidity of atmospheric condensed phases, specifically particles and cloud droplets. It includes recommendations for estimating acidity and pH, standard nomenclature, a synthesis of current pH estimates based on observations, and new model calculations on the local and global scale.

## 1   The importance of atmospheric acidity

Human activity and natural processes result in emissions of sulfur, nitrogen, ammonia, dust, and other compounds that affect the composition of the Earth's atmosphere. The acidity of suspended atmospheric media, particles and droplets, influences many processes that involve the atmosphere and all aspects of the Earth system (e.g., watersheds, marine and terrestrial ecosystems) that interface with it, see Fig. 1. Aerosols (also referred to as particulate matter, PM) and cloud droplets throughout

the atmosphere exhibit a wide range of acidity, each spanning five orders of magnitude or more in molality units, or five units of pH (Fig. 2). Some anthropogenic emissions (sulfur dioxide, nitrogen oxides, organic acids) increase acidity while others (ammonia, non-volatile cations (NVC), amines) reduce acidity. The orders of magnitude differences in water content between aerosols and clouds leads to distinctly different acidity levels in these media, as well as their response to changes in precursor concentrations. The ability of chemical species to affect particle or cloud droplet acidity is driven by both its degree of acidity

(or basicity), reflected in their dissociation (or association) constants, and by volatility, with less volatile compounds partitioning to a greater degree into liquid aerosols and cloud droplets. Semi-volatile species, for which significant fractions typically exist in both the gas and condensed phases, include ammonia ($NH_3$), nitric acid ($HNO_3$), hydrochloric acid (HCl), and low molecular weight organic acids. Sulfuric acid ($H_2SO_4$), by contrast, has extremely low volatility and can be treated as entirely in the condensed phase for most applications. Metal cations, including those found in dust and sea salt, are also

essentially non-volatile. The abundance of these various constituents is a function of emission source and atmospheric processing and ultimately dictates the pH of fine particles (Fig. 1, 2).

Although aerosol and cloud acidity are distinct in many ways, aerosol forms in part from cloud evaporation, and so aerosol composition and acidity may be directly affected by cloud chemistry. Similarly, cloud droplets and ice crystals nucleate on

pre-existing particles, and therefore much of the material that modulates cloud acidity originates from the precursor aerosol. In addition, cloud droplets can collide with surfaces resulting in occult deposition (Dollard et al., 1983) or precipitation in the



form of rain. With these connections in mind, both aerosol and cloud acidity are important to human health, ecosystem health and productivity, climate, and environmental management.

The acidity of atmospheric deposition for dry, wet, and occult pathways is directly affected by aerosol and cloud pH (Fig. 1).

Thus, programs designed to reduce acid rain (e.g., the Acid Rain Program under Title IV of the 1990 Clean Air Act Amendments in the U.S.), have had implications for particle and cloud droplet acidity. In terrestrial ecosystems, direct effects of acid deposition to foliage include leaching of cations, altered stomatal function and changes in wax structure (Cape, 1993). Acid deposition can exacerbate soil acidification (Binkley and Richter, 1987), resulting in loss of soil base cations, leaching of nitrate and mobilization of aluminium, affecting terrestrial ecosystem health and the quality of water delivered to streams

and lakes (Driscoll et al., 2007). Apart from reactive nitrogen, atmospheric deposition is also a significant source of limiting and trace nutrients such as phosphorus, iron, and copper, especially in the remote oceans (Mahowald et al., 2008; Myriokefalitakis et al., 2018). While mineral dust is a major source of these nutrients, combustion sources also emit iron, copper and other trace metals (Reff et al., 2009; Ito et al, 2019). Acid processing of aerosol prior to deposition, may greatly enhance the solubility of all these compounds increasing their bioavailability and ecosystem impacts (Meskhidze et al., 2003;

Nenes et al., 2011; Kanakidou et al., 2018). For example, dust aerosols coated by acidic sulfate and nitrate show increased Fe solubility compared to fresh dust particles, particularly in the fine mode, the deposition of which may promote phytoplankton blooms in nutrient-limited regions of the oceans (Meskhidze et al., 2005). The same process occurs for P (e.g., Nenes et al., 2011; Stockdale et al., 2016); however, the extent to which particle pH may similarly increase the solubility and amount of organic forms of nitrogen and phosphorous, a potentially large source to ecosystems (Jickells et al., 2013), is not well known

(Kanakidou et al., 2018). Deposition of trace nutrients from acid-promoted dissolution into regions of the ocean where the nutrients are not limiting the biological productivity may enhance the productivity in other regions that are nutrient-limited by means of long-range transport by ocean currents. Such a redistribution of nutrients can have important implications for the biogeochemistry of the ocean, the oxygenation state, and the carbon cycle (Ito et al., 2016).

Aerosol acidity is also a governing factor for atmospheric dry deposition of inorganic reactive nitrogen species, which is a key nutrient driving primary productivity in terrestrial and marine ecosystems. The hydrogen ion activity in aqueous aerosols affects the partitioning of total nitrate ($TNO_3 = HNO_3 + NO_3^-$) and total ammonium ($TNH_4 = NH_3 + NH_4^+$) between the gas and aerosol phases. Given the much larger deposition velocity of gases compared to submicron aerosols, pH-mediated partitioning influences the effective deposition velocity and lifetime of $TNO_3$, $TNH_4$, and total inorganic N ($TNO_3 + TNH_4$).

Acidity therefore also affects the magnitude and spatial patterns of inorganic N deposition to terrestrial and aquatic ecosystems. Lower aerosol pH favors partitioning of $TNO_3$ toward gaseous $HNO_3$ rather than aerosol $NO_3^-$ thus shortening its lifetime (Weber et al., 2016). In contrast, $TNH_4$ ($NH_3 + NH_4^+$) partitions toward gaseous $NH_3$ at higher pH. Conditions of aerosol pH that promote a short residence time and local dry deposition of $TNO_3$ may conversely result in longer range transport of $TNH_4$ and a more spatially extensive pattern of deposition and influence from source regions. The presence of dust and sea salt can





influence not only pH but the size distribution (Lee et al., 2008, Lee et al., 2004) and resulting deposition velocity of nitrate aerosol due to higher deposition velocities of coarse compared to fine mode particles (Slinn, 1977). Variations in this scavenging efficiency, dependent upon cloud pH, can also affect atmospheric lifetimes and spatial deposition patterns of $TNH_4$. $HNO_3$, due to its strong acidity, is essentially partitioned entirely to cloud droplets for typical cloud pH values (>2). $NH_3$ also

mostly partitions into cloud drops for pH values below 6, but an appreciable fraction can remain in the gas phase for higher pH values. Biases in pH in atmospheric models can therefore influence the amount, speciation, and location of N deposition, with implications for determining ecosystem critical load exceedances for nutrients and acidity (Bobbink et al., 2010).

PM$_{2.5}$ is associated with adverse human health effects, including premature mortality (Di et al., 2017; Lepeule et al., 2012;

Pope et al., 2009; EPA, 2009). Aerosol acidity is associated with health effects of air pollution through its influence on atmospheric processes that affect the amount and composition of PM$_{2.5}$ (Fig. 1). The concentration of fine particulate matter (PM$_{2.5}$) is directly modulated by pH through its effects on gas–particle partitioning and pH-dependent condensed-phase reactions. The "strong acidity" property of aerosol (Koutrakis et al., 1988) has historically been associated with adverse health effects (Dockery et al., 1993; Dockery et al., 1996; Thurston et al., 1994; Raizenne et al., 1996; Spengler et al., 1996; Gwynn

et al., 2000; EPA, 2009). One reason for this could be that aerosol acidity influences solubilization and the concentrations of toxic forms of trace species, such as transition and heavy metals, that have been linked to negative health effects (Kelly and Fussell, 2012; Lippmann, 2014; Rohr and Wyzga, 2012; Chen and Lippmann, 2009, Frampton et al., 1999). Transition metal ions (TMI), such as soluble Cu and Fe from acid dissolution, contribute significantly to the oxidative potential of particles (Fang et al., 2017; Poschl and Shiraiwa, 2015), which has been linked to cardiorespiratory emergency department visits with

a stronger association than PM$_{2.5}$ mass (Abrams et al., 2017; Bates et al., 2015). Ye et al. (2018) report a strong association between soluble Fe, which is modulated by particle acidity and aerosol water content, and cardiovascular endpoints. The mechanistic link between acidity, TMI dissolution, and health outcomes recently proposed by Fang et al. (2017) may help explain why sulfate in the ambient atmosphere is associated with adverse health outcomes – in contrast to studies that show little role for sulfate in negative health endpoints (Schlesinger, 2007; Reiss et al., 2007).

Aerosol acidity can affect the gas–particle partitioning of semi-volatile toxic organic pollutants and therefore their environmental fate and pathways for exposure (Vierke et al., 2013). Some per- and polyfluoroalkyl substances (PFASs), including perfluoroalkane sulfonic acids (PFSAs) and perfluoroalkyl carboxylic acids (PFCAs), are strongly acidic and likely to be at least partially dissociated (ionized) under pH conditions typical of most atmospheric aerosols (Ahrens et al., 2012).

Once in the particle phase, they are vulnerable to hydrolysis, which shortens their lifetime in the environment but may lead to the formation of toxic degradation products (Tebes-Stevens et al., 2017). Aerosol acidity was also recently shown to enhance airborne nicotine levels and resulting thirdhand smoke exposure (DeCarlo et al., 2018). Furthermore, aerosol acidity may also affect particle toxicity on a per mass basis (increase or decrease) by influencing organic aerosol composition (Arashiro et al., 2016; Tuet et al., 2017). Many organic compounds that are toxic (e.g., nitrosamines) can also be formed in the aerosol phase



under acidic conditions; at the same time, other potentially toxic compounds (e.g., organonitrates) may hydrolyze under strongly acidic conditions (Rindelaub et al., 2016a). Even for non-toxic organic aerosol facilitated by acidity, the enhanced (or conversely reduced) formation of inert organic mass in the particle promotes the partitioning (or evaporation) of toxic species, such as polycyclic aromatic hydrocabons (PAHs) (Liang et al., 1997), from the gas phase to the particle and thereby alter the

location of deposition in the respiratory airways. For highly soluble organic species, uptake to the aerosol phase can also potentially extend their atmospheric lifetimes by protecting the species against deposition on vegetation and ground surfaces.

Since acidity impacts the mass and chemical composition of atmospheric aerosols, which scatter and absorb radiation and serve as cloud condensation nuclei (CCN), acidity can also affect climate. First, particle pH influences, and is related to, the

water uptake properties (hygroscopicity) of particles, which in turn can modulate both visibility and the radiative balance throughout the atmosphere (the aerosol direct climate effect). Cloud pH has been linked to the amount and speciation of aerosol upon evaporation, with important radiative effects (Turnock et al., 2019). Changes in acidity can also affect the number of chromophores contained within aerosol (so-called brown carbon) and their efficiency to absorb sunlight in the near-UV range (Hinrichs et al., 2016; Teich et al., 2017; Phillips et al., 2017). Acidity-induced changes in aerosol affect the ability of particles

to act as CCN and contribute to the formation of droplets in warm and mixed-phase clouds. For example, insoluble particles, such as dust, facilitate the production of ice crystals in mixed-phase and cold clouds (Seinfeld and Pandis, 2016); acidification of these particles can modify the active sites that ice is formed upon, and thereby affect the distribution of ice and liquid water throughout the atmosphere (Sullivan et al., 2010; Reitz et al., 2011). The distribution of droplets and ice also may in turn regulate the riming efficiency in mixed-phase clouds (Pruppacher and Klett, 2010) and distribution of clouds throughout the

atmosphere. Changes in cloud distribution strongly modulate Earth's radiative balance and the hydrological cycle (IPCC, 2007).

Atmospheric acidity also plays an important role in new particle formation, which is thought to contribute up to 50% of the CCN concentrations in the atmosphere, thus acting as a climate regulator (Gordon et al., 2017). Sulfuric acid likely plays a

critical role in the formation of stable clusters upon which new particles are formed (Weber et al., 1997; Weber et al., 1998), while bases such as amines and ammonia (Jen et al., 2016) can facilitate stabilization and growth of such clusters. Uptake of organic acids through acid-base chemistry (Zhang et al., 2004; Hodshire et al., 2016) and acid-mediated secondary organic aerosol formation (e.g., McNeill, 2015) impact the aerosol size distribution with implications for CCN concentrations, cloud droplet formation and climate.

Understanding particle acidity can facilitate improved air quality management strategies and policy planning to mitigate the health and environmental effects of air pollution. Consideration of different policy options and the development of emission reduction strategies often relies on chemical transport model (CTM) simulations of future conditions. Such modeling depends on the capability of CTMs to adequately simulate responses to policy scenarios. The predictive capability of CTMs is closely

linked to their ability to track particle acidity through the pathways shown in Fig. 1. Some studies have pursued the development of observation-based indicators for the sensitivity of pollutants to precursor emissions for use in CTM evaluation and air quality management (e.g., Gas Ratio, Sect. 4). However, the use of the sensitivity indicators has been limited because their robustness has not been well established. Recent work (Shah et al., 2018; Vasilakos et al., 2018) has begun to explore the

influence of particle acidity on the simulated responsiveness of $PM_{2.5}$ to emissions changes. Vasilakos et al. (2018) demonstrated that reliable predictions of particle pH in CTMs are key to modeling the response of $PM_{2.5}$ components to precursor emission changes. Furthermore, pH biases may propagate to biases in nitrate partitioning, dissolved metal concentrations, inorganic and organic aerosol amount and composition, aerosol size distributions, and ultimately could affect predicted impacts of emissions on ecosystem productivity and public health.

This study reviews the current understanding of aerosol and cloud acidity in the atmosphere. The work is motivated by the central role of aerosol and cloud acidity in numerous complex atmospheric processes of importance to human health and welfare as well as the rapid growth in literature on aerosol acidity in recent years. Despite decades of research on these processes, relatively few observational constraints exist for model evaluation. This review aims to collect values of fine aerosol

and cloud pH as well as discuss the approaches used to determine them. We provide an overview of the range of pH acidity scales and methods of approximating pH as well as discuss their challenges and advantages (Sect. 2). In addition, we discuss the role of chemistry in driving and being modulated by pH (Sect. 3), proxies of pH (Sect. 4), observations of particle (Sect. 5.1) and cloud (Sect. 5.2) pH, insights from box modeling of particle pH and its approximations and proxies (Sect. 6), the role of size and composition (Sect. 7), and regional and global model representations of pH (Sect. 8).

**2    The definition of pH and pH scales**

Aerosol acidity is generally not directly measured, despite some recent progress (see Sect. 5.1.1-5.1.2). Instead, estimates are obtained from thermodynamic models that involve assumptions, which can vary according to the completeness of the atmospheric dataset being considered. The numerical value of pH also differs according to the concentration scale in use. Furthermore, pH has a number of different definitions – each devised for a particular application – and can only be measured

accurately and with metrological traceability for dilute solutions. These facts are not always well understood. This section summarizes the formal definition of pH and operational definitions and approximations. Thermodynamic models used to calculate fine particle pH are also discussed.

**2.1    Definition of acidity in terms of the pH**

The degree of acidity or basicity of a solution can be quantified based on the thermodynamic activity (the effective

concentration, including non-ideal behavior) of dissolved hydrogen ions ($H^+$). In its most common form, this measure of acidity



is reported as a dimensionless quantity known as the pH. The International Union of Pure and Applied Chemistry (IUPAC) defines pH as (Buck et al., 2002; IUPAC, 1997):

$$\text{pH} = -\log_{10}(a_{\text{H}^+}) = -\log_{10}\left(\frac{m_{\text{H}^+}}{m^{\ominus}}\,\gamma_{\text{H}^+}\right), \tag{1}$$

where $a_{\text{H}^+}$ denotes the activity of H$^+$ in aqueous solution on a molality basis, $m_{\text{H}^+}$ is the molality of H$^+$ (mol kg$^{-1}$, i.e. moles of

H$^+$ ions per kg of solvent, typically pure water), and $\gamma_{\text{H}^+}$ is its molal activity coefficient (see Table 1 for a summary of definitions). The quantity $m^{\ominus} = 1$ mol kg$^{-1}$ is the standard state (unit) molality used to achieve a dimensionless quantity in the logarithm (Covington et al., 1985) (omitted for simplicity in future equations). For solid particles or ice clouds, and potentially for glassy particles, a single pH value is undefined due either to the lack of a liquid aqueous phase or the potential for long intraparticle mixing timescales. In Eq. (1), both $a_{\text{H}^+}$ and $\gamma_{\text{H}^+}$ are molality-based with a reference state of infinite

dilution in pure water ($\gamma_{\text{H}^+} \rightarrow 1$ as $m_{\text{H}^+} \rightarrow 0$). In most calculations involving natural systems the solvent is pure water, and therefore the molality, $m_i$ (mol kg$^{-1}$), of solute species $i$ is given by $m_i = \frac{n_i}{n_w M_w}$, where $n_i$ is the number of moles of $i$ in the aerosol or cloud water particles, $n_w$ the number of moles of water and $M_w$ the molar mass of water. For some applications involving solutions containing large fractions of organic material that are miscible with water, the definition of the 'solvent' may be altered to include all non-ionic (organic) species. This is largely for practical reasons and because some thermodynamic

models of activities in solutions and liquid mixtures (e.g., Yan et al., 1999; Zuend et al., 2008) require it. The activity coefficients used in the calculation of pH must be consistent with both the definition of what constitutes the solvent and also the concentration scale used. Further explanation is given in the Supplementary Information (Sect. S1).

The IUPAC definition of pH (Eq. 1) is regarded as a notional definition, because it involves the activity coefficient of a single

ion (Buck et al., 2002; Covington et al., 1985). These are inaccessible experimentally because electrolyte solutions (of any relevant amount of substance) always contain both cations and anions, in proportions yielding an overall electroneutral system.

Only mean activity coefficients of neutral cation–anion combinations are measurable quantities, such as $\gamma_{\pm,\text{HCl}} = \left[\gamma_{\text{H}^+}\gamma_{\text{Cl}^-}\right]^{\frac{1}{2}}$ in the case of the 1:1 electrolyte HCl (e.g., Prausnitz et al., 1999; Robinson and Stokes, 2002). Several, but not all, thermodynamic activity coefficient models used in atmospheric science and geochemistry provide a computation of single-ion activity

coefficients within their mathematical framework (see later discussion of aerosol models, Sect. 2.6). However, these single-ion values are purely conventional in that they depend on assumptions inherent in the derivation of the model equations and, unlike mean activity coefficients, are not necessarily comparable between models.

## 2.2   Alternative pH concentration scales

Older definitions of pH by IUPAC, alongside Eq. (1), define the pH value on a molarity scale (pH$_c$) (Covington et al., 1985):

$$\text{pH}_c = -\log_{10}\left(a_{\text{H}^+}^{(c)}\right) = -\log_{10}\left(\frac{c_{\text{H}^+}}{c^{\ominus}}\,\gamma_{\text{H}^+}^{(c)}\right). \tag{2}$$





The superscript $(c)$ indicates the molarity basis for the activity ($a_{H^+}^{(c)}$) and activity coefficient ($\gamma_{H^+}^{(c)}$), distinct from the molality

basis. The reference state is still infinite dilution in pure water ($\gamma_{H^+}^{(c)} \to 1$ as $c_{H^+} \to 0$), and the quantity $c^{\ominus} = 1$ mol dm$^{-3}$ is the

standard state molarity. The quantity $c_{H^+}$ denotes the molarity or molar concentration of H$^+$ in an aqueous solution (i.e., mol

of H$^+$ per dm$^3$ of aqueous solution; IUPAC, 1997). For dilute solutions, $c_{H^+}$ is practically equivalent to the molar amount of

ion per dm$^3$ of pure water. Covington et al. (1985) point out that for most applications involving dilute aqueous solutions, the

pH and pH$_c$ values obtained from molality and molarity scales (for the same mixture) are of negligible numerical difference.

The pH difference depends mainly on the density of water, and the difference in molal vs molarity-based pH is approximately

0.001 pH units at 298.15 K, increasing to about 0.02 pH units at 393.15 K (with larger differences expected for concentrated

aqueous electrolyte solutions and/or those with mixed solvents).

The mole fraction concentration scale is used by the Extended Aerosol Inorganics Model (*E-AIM*) of Clegg and co-authors

(Clegg et al., 2001; Wexler and Clegg, 2002, and references therein). The pH on a mole fraction basis, pH$_x$, is given by:

$$\text{pH}_x = -\log_{10}\left(a_{H^+}^{(x)}\right) = -\log_{10}\left(x_{H^+}\, f_{H^+}^*\right), \tag{3}$$

where $x_{H^+}$ is the mole fraction of H$^+$ in the solution, $a_{H^+}^{(x)}$ and $f_{H^+}^*$ are the mole-fraction-based activity and the (rational) activity

coefficient, respectively, both defined with respect to an infinite dilution reference state in pure water (superscript $*$ or $(x)$).

The mole fractions of all species $i$, including water, are calculated as $x_i = n_i/\Sigma_j\, n_j$, where the summation is calculated over all

solution species $j$ (ions, uncharged (e.g., organic) solutes, and water).

Conversions among pH values calculated using different concentration scales is necessary to compare model predictions, and

to report acidity on a consistent basis. Generally, formulae for the conversion of pH are derived based upon the equivalence of

the chemical potentials of solution entities irrespective of concentration scale (see, for example, Robinson and Stokes, 2002).

The conversions to pH from the equivalent values on the mole fraction and molarity scales are given below, and the derivations

are given in the Supplementary Information (Sect. S1):

$$\text{pH} = \text{pH}_x + \log_{10}\left(M_w \cdot 1\, \tfrac{\text{mol}}{\text{kg}}\right) \approx \ \text{pH}_x - 1.74436, \tag{4}$$

$$\text{pH} = \text{pH}_c - \log_{10}\left(\tfrac{1}{\rho_0} \cdot 10^3\, \tfrac{\text{kg}}{\text{m}^3}\right). \tag{5}$$

where, $\rho_0$ (kg m$^{-3}$) is the density of the reference solvent (pure water for the normal case, i.e. when activity coefficients on

molality and molarity scale are defined with reference state of infinite dilution in pure water, regardless of a presence of

organics in the solution). Because the density $\rho_0$ depends weakly on temperature, the exact relation between pH and pH$_c$ is

non-linear. In the usual case of water as the reference solvent ($\rho_0$ close to 1000 kg m$^{-3}$), the logarithmic difference is small

(typically < 0.02 pH units), resulting in pH$_c \approx$ pH (Jia et al., 2018).


## 2.3 Approximations of pH

Approximate values of pH, based upon the definition of pH (Eq. 1) but making simplifying assumptions, can be obtained in several ways. For example, the activity coefficient $\gamma_{H^+}$ could be set to unity and pH computed based on only the *free* $H^+$ molality symbol $pH_F$:

$$pH_F = -\log_{10}(m_{H^+}) \tag{6}$$

The assumption of $\gamma_{H^+} = 1$ is appropriate only in highly dilute aqueous solutions, corresponding to ambient relative humidities close to 100%.

Another approach is to use the mean molal ion activity coefficient of an $H^+$–anion pair in place of $\gamma_{H^+}$, i.e. $\gamma_{H^+} \approx \gamma_{\pm,H,X}$, where

X is a monovalent anion such as $HSO_4^-$, $NO_3^-$, or $Cl^-$ (e.g., Wright, 2007). This approximation can be expected to capture the typical increase in $\gamma_{H^+}$ with increasing $H^+$ liquid phase concentration (decreasing ambient RH), although only semi-quantitatively. The approximate pH determined in this way is labeled as $pH_{\pm}(H,X)$:

$$pH_{\pm}(H,X) = -\log_{10}\left(m_{H^+}\gamma_{\pm,HX}\right). \tag{7}$$

The deviation of $pH_{\pm}(H,X)$ from pH is related to the ratio of the specified single-ion activity coefficients via $pH_{\pm}(H,X) -$

$pH = \frac{1}{2}\log_{10}\left(\frac{\gamma_{H^+}}{\gamma_{X^-}}\right)$. Consequently, the pH approximation by $pH_{\pm}(H,X)$ is very good for $\frac{\gamma_{H^+}}{\gamma_{X^-}} \approx 1.0$, which is the case in the highly dilute limit of aqueous electrolyte solutions. Further, it may also hold approximately towards higher electrolyte concentrations if both single-ion activity coefficients tend to deviate from 1.0 to a similar degree (which depends on aerosol composition).

An alternative to $pH_F$ would be to use the *total* $H^+$ molality, which can be defined as the sum of dissolved $H^+$ and $HSO_4^-$ molalities:

$$pH_T = -\log_{10}(m_{H^+} + m_{HSO4^-}) \tag{8}$$

The use of this definition may be appropriate in contexts where the amount of free $H^+$ is not of interest and/or the computation of bisulfate dissociation (which will vary with RH, aerosol composition, and temperature) is impractical. The use of $pH_F$ and

$pH_T$ as alternatives to pH, as well as the assumption that $pH \approx pH_{\pm}(H,X)$, are tested by the intercomparison of thermodynamic model predictions for fine particles in Sect. 6.

## 2.4 Acidity and the pH scale

Expressing the acidity of a solution in terms of pH leads to a scale that has two important characteristics: an increase in acidity is accompanied by a decrease in pH and vice-versa; and it is a logarithmic scale, meaning that a decrease by one pH unit

corresponds to a ten-fold increase in $H^+$ activity. Hence, apparently modest changes in pH represent relatively large changes in acidity.



A pH of 7 represents a neutral aqueous solution with values less than 7 generally considered acidic and values larger than 7 basic in nature. The characterization of pH equal to 7 as neutral is based on the chemical equilibrium between $H^+$ and $OH^-$ ions arising naturally in aqueous solutions. The auto-dissociation of water ($H_2O \rightleftharpoons OH^-_{(aq)} + H^+_{(aq)}$) is described by the temperature

($T$)-dependent equilibrium constant on molality basis, $K_w(T)$. The value of $pK_w$ ($= -\log_{10}[K_w]$) is 14.95 at 0°C and 13.99 at 25°C for pressures encountered in the atmospheric (Bandura and Lvov, 2005) resulting in corresponding pH values of about and 7.475and 6.995, respectively, for highly dilute aqueous systems. Both systems are neutral, although the pH values differ.

The pH scale is commonly considered to span values from 0 to 14, but larger and smaller values are also possible as the scale

has no specific limits. Current large-scale models (Sect. 8) and observations (Sect. 5) indicate that cloud pH has a global mean somewhere between 4 and 6 and ranges from around 2 to above 7 (Fig. 2). The global distribution of fine particle (nominally particles of 2.5 μm in diameter and below, $PM_{2.5}$) pH is bimodal with a population of particles having a mean pH of 1 – 3 and another population, influenced by dust, sea spray, and potentially biomass burning, having an average pH closer to 4 – 5. Fine particle pH can be negative (Sect. 5.1), particularly when sulfate is a major component, and is rarely predicted to exceed 7.

**2.5    Measuring pH and operational definition of pH**

The small sizes and associated liquid volumes of single particles (which are in chemical equilibrium with vapours) prevent the application of standard pH measurement techniques for single aerosol particle and cloud droplet acidity. Instead, samples of larger volumes must be collected (e.g., a population of droplets in the case of cloud water, Sect. 5.2) or other methods employed, including measurements of aerosol and gas phase compositions and the application of thermodynamic models (Sect.

2.6) to compute pH values via Eq. (1).

With sufficient sample volume, particularly in the case of cloud droplets which typically have low ionic strengths, traditional pH measurement techniques can be used. The operational definition of pH is based on the principle of determining the difference between the pH of a solution of interest and that of a reference (buffer) solution of known pH by measuring the

difference in electromotive force, using an electrochemical cell (e.g., a combination electrode coupled to a pH meter). High-precision measurement of absolute pH values of the reference buffer solution used for calibration are made with a so-called primary method using electrochemical cells without transference (Harned cells, see Buck et al. 2002). The uncertainty associated with typical pH measurements, which use glass electrodes, is of the order of 0.014 for ionic strengths < 0.1 mol kg$^{-1}$ and is expected to increase towards higher ionic strength (Buck et al., 2002). Further details on pH measurement methods

and their relationship to Eq. (1) are provided in the SI.





The measurement of aerosol pH is problematic because of the difficulty in collecting sufficient sample material without perturbing its acidity and also due to the mismatch between ionic strengths present in atmospheric fine particles ($> 1$ mol kg$^{-1}$ and sometimes exceeding 100 mol kg$^{-1}$, Herrmann et al., 2015) and the molal ionic strength for which normal operational techniques are appropriate (no more than about 0.1 mol kg$^{-1}$). Values of pH based upon primary measurement methods, which

are used for instrument calibration, cannot be readily defined at high ionic strength. This is because assumptions regarding the activity coefficient of the Cl$^{-}$ ion (which is needed to establish the pH value of the buffer) are limited to very dilute solutions. In addition, the calibration of the pH electrodes requires the ionic strengths of the buffer and test solution (e.g. an aerosol sample) to be low and similar in magnitude. A pH electrode, calibrated for dilute conditions, will yield a measured pH that is systematically in error to an unknown degree if placed in a solution of higher ionic strength (e.g., Wiesner et al., 2006). Similar

considerations apply to colorimetric methods (see Sect. 5.1): the equilibria involving the chemical species that provide the color response depend not solely on $a_{H^+}$, but on the thermodynamic activities of the sensing species themselves. These will vary with the chemical composition and concentration of the solution. Thus, colorimetric methods also require calibration that is relevant to the solution media they will be used to measure.

## 2.6    Thermodynamic models for pH calculation

Given the operational difficulties associated with measuring aerosol pH (Sect. 2.5), estimates of the degree of acidity of particles generally depend upon the use of thermodynamic models. In atmospheric science, a number of different thermodynamic models are used to predict equilibrium gas–particle partitioning, liquid-phase activity coefficients, solid–liquid and liquid–liquid equilibria, dynamic mass transfer of semi-volatile species, aerosol liquid water content (ALWC), and pH. Most models can treat both metastable (supersaturated) solutions or stable states (where solids have formed). Here, some of

the most widely used models are described, focusing on their general approach, special features, and relevant species in the context of pH calculations.

The thermodynamic modeling approach inherent in some models (e.g., ISORROPIA, MOSAIC, and EQUISOLV II) does not yield single ion activity coefficients that allow for calculation of pH via Eq. 1. For example, ISORROPIA II and MOSAIC

nominally output information for pH$_F$ and studies published prior to this work using those models (see Sect. 5.1), were approximating acidity by reporting pH$_F$. In this work (Sect. 6), model source codes were modified to use the mean molal activity coefficients for different cation–anion pairs (e.g., (H$^+$, HSO$_4^-$) or (H$^+$, Cl$^-$)) in the estimation of pH using pH$_\pm$(H, X). Section 6 and observationally constrained pH estimates focus on equilibrium conditions, although MOSAIC is often used to dynamically calculate the transient H$^+$ amount.

### 2.6.1    Extended Aerosol Inorganics Model (E-AIM)

The Extended Aerosol Inorganics Model (E-AIM) is a thermodynamic model to calculate gas/liquid/solid equilibrium in aqueous aerosol systems containing inorganic ions, water, and an arbitrary number of organic compounds with user-defined





properties. It uses the Pitzer–Simonson–Clegg equations (Pitzer and Simonson, 1986; Clegg et al., 1992; Clegg et al., 1998) for the calculations of solvent and solute activity coefficients (single ion values) on the mole fraction scale. There are four principal models that differ in terms of the species and temperature range considered (Wexler and Clegg, 2002, and references therein; Friese and Ebel, 2010). The models include some or all of the following ions and the solid salts and gases that can be

formed from them: $H^+$, $NH_4^+$, $Na^+$, $SO_4^{2-}$, $HSO_4^-$, $NO_3^-$, $Cl^-$, and $Br^-$. The possible calculations include the properties of an aqueous solution of defined composition, as well as the equilibrium state of a gas and particle system at defined RH and temperature. In chemical systems containing inorganic ions, the aqueous phase equilibria $H^+ + SO_4^{2-} \rightleftharpoons HSO_4^-$, $NH_4^+ \rightleftharpoons NH_3 + H^+$, and $H_2O \rightleftharpoons H^+ + OH^-$ are solved as well as those between aerosol species $NH_3(aq)$, $H^+(aq)$, $NO_3^-(aq)$, $Cl^-(aq)$ with the gases $NH_3$, $HNO_3$, and $HCl$. Analogous equilibria (i.e., acid dissociation, gas/liquid equilibrium) can also be solved for user-

specified mono- and di-carboxylic acids, and mono- and di-amines. The model can be used for both acidic and alkaline aerosols.

The activity coefficients, and contributions to the water activity, of uncharged (or undissociated forms of) organic solutes are calculated using the Universal Quasi-Chemical Functional group Activity Coefficients (UNIFAC) model (Fredenslund et al.,

1975). Organic anions are assumed to have the same activity coefficient model interaction parameters as $HSO_4^-$ or $SO_4^{2-}$ (according to their charge), and amine cations are assigned the same parameters as $NH_4^+$.

The *E-AIM* model is based upon thermodynamic data for pure aqueous solutions and mixtures over a wide range of temperatures. This basis in measurements, and the calculation of ionic activities in terms of interactions between pairs and

triplets of solute species, makes *E-AIM* generally the most accurate (inorganic) thermodynamic model used in atmospheric science. Nevertheless, it has some known weaknesses: predictions of rising equilibrium RH with concentration in some aqueous $NH_4^-$–$NO_3^-$–$SO_4^{2-}$–$H_2O$ aerosols at about 250 K and below (Model II); and similar errors in aqueous aerosols at low RH and containing high concentrations of $NH_4^+$ and $Cl^-$ (Model IV). To address the latter case some restrictions have been placed on the types of calculations that can be carried out (see http://www.aim.env.uea.ac.uk/aim/model4/input4a.html). Most

relevant to aerosol pH is the fact that calculated molalities of free $H^+$, $HSO_4^-$ and $SO_4^{2-}$ in aqueous $H_2SO_4$ – and therefore in mixtures containing the three ions – deviate somewhat from measurements of the stoichiometric dissociation constant of $HSO_4^-$ obtained spectroscopically (Knopf et al., 2003; Myhre et al., 2003). Myhre et al. (2003) show that there is good agreement between modeled and measured degrees of dissociation of $HSO_4^-$ ($\alpha_{HSO4}$) at room temperature up to 30 – 40 wt% acid (equivalent to 75 to 56% equilibrium RH), within the relatively large scatter in the data. At 50 wt% acid, and above, the

calculated $\alpha_{HSO4}$ are too low, meaning that the molality of free $H^+$ is also too low. These differences increase for $H_2SO_4$ concentrations above 35 wt% (5.5 mol $kg^{-1}$) and temperatures below about 240 K (Knopf et al., 2003). These errors do not necessarily lead to errors in pH as the stoichiometric activity of $H^+$ (used in determination of pH, Eq. 1) in aqueous solution is accurately reproduced as indicated by accurate predictions of equilibria with acid gases ($HNO_3$ and $HCl$, Fig. 6 to 12 of Carslaw et al., 1995).





### 2.6.2    AIOMFAC-based equilibrium model

The Aerosol Inorganic–Organic Mixtures Functional groups Activity Coefficient (AIOMFAC) model is a thermodynamic activity coefficient model treating liquid mixtures containing water, inorganic ions and organic compounds. The model combines a Pitzer-type aqueous ion interaction model with a modified UNIFAC model (Fredenslund et al., 1975; Hansen et al., 1991), which was originally designed for organic mixtures. As in UNIFAC, AIOMFAC applies a group-contribution approach to cover a wide variety of organic compounds by a relatively small set of organic functional groups (~16 main groups). The AIOMFAC expressions, parameterization and validation based on experimental data, as well as known limitations are described in detail elsewhere (Zuend et al., 2011; Zuend et al., 2008).

AIOMFAC presently includes the following inorganic ions: $H^+$, $Li^+$, $Na^+$, $K^+$, $NH_4^+$, $Mg^{2+}$, $Ca^{2+}$, $Cl^-$, $Br^-$, $NO_3^-$, $HSO_4^-$, $SO_4^{2-}$, ($I^-$ forthcoming). Most of these ions can be present simultaneously in an aqueous solution; limitations exist for $Li^+$ in the presence of bisulfate ($HSO_4^-$) ions due to a lack of experimental data required for determining associated model parameters. However, using a less-rigorous analogy approach, AIOMFAC can approximate those parameters so that all listed ions can be treated in solution. The bisulfate dissociation equilibrium is solved numerically using the temperature-dependent equilibrium constant parameterization by Knopf et al. (2003). Other inorganic electrolyte species are considered completely dissociated when in liquid solution – with deviations from that assumption accounted for implicitly by activity coefficients. In contrast to the E-AIM model, AIOMFAC does not solve the $H_2O \rightleftharpoons H^+ + OH^-$ dissociation equilibrium (which is acceptable when the pH is at least an order of magnitude lower than the neutral value). The organic functional groups available in calculations for mixed organic–inorganic systems include carboxyl, hydroxyl, ketone, aldehyde, ether, ester, alkenyl, alkyl, hydroperoxide, peroxyacid, peroxide, and aromatic functional groups. However, only a subset of these groups is currently available when $HSO_4^-$, $Mg^{2+}$ or $H^+$ are present; see https://aiomfac.lab.mcgill.ca/about.html (Fig. 4 on that website). A few species are available exclusively for a select set of organic or inorganic systems, e.g., an organonitrate group in non-electrolyte systems (Zuend and Seinfeld, 2012) and the methanesulfonate ion in certain organic-free aqueous solutions ($CH_3SO_3^-$ with $H^+$, $Na^+$, $NH_4^+$; Fossum et al., 2018).

An online version of AIOMFAC is available at https://aiomfac.lab.mcgill.ca (and http://www.aiomfac.caltech.edu). Note that the online AIOMFAC model is simply an activity coefficient model, not a complete gas–liquid thermodynamic equilibrium model. A limitation of AIOMFAC is that the composition-dependent degree of dissociation of organic acids (via the carboxyl group) is not accounted for explicitly in the determination of acidity. As such, the pH calculations are only meaningful in the presence of some amount of inorganic $H^+$. Due to the weak temperature-dependence of activity coefficients, the model is applicable over a temperature range of about $298 \pm 30$ K, while most of the experimental training data were for temperatures $\geq 293$ K. AIOMFAC variants with a more sophisticated temperature dependence have been parameterized for electrolyte-free aqueous organic systems (Ganbavale et al., 2015).





A unique feature of AIOMFAC is its ability to represent non-ideal interactions between organic molecules and inorganic ions in liquid solutions up to high concentrations, a feature that is important for the prediction of liquid–liquid phase separation. Thermodynamic equilibrium models have been developed with AIOMFAC as their core module, including efficient numerical methods for the prediction of liquid–liquid equilibria (Zuend and Seinfeld, 2013) and the equilibrium gas–particle partitioning

of water and semi-volatile organic compounds (Zuend and Seinfeld, 2012; Zuend et al., 2010).

Recent work further extends the AIOMFAC-based gas–particle partitioning model by consideration of the gas–liquid equilibria of the following inorganic acids and bases: $HNO_3$, $HCl$, $HBr$, and $NH_3$, while $H_2SO_4$ is treated as non-volatile (Ma and Zuend, *in preparation*). This equilibrium model is referred to as AIOMFAC–GLE hereafter. For given input in the form of molar

amounts per unit volume of air at given pressure and temperature, the AIOMFAC–GLE model predicts the compositions of co-existing phases (gas phase plus up to two liquid phases) and associated activity coefficients of all species in all (liquid) phases. This enables a straightforward calculation of phase-specific pH values using Eq. (1).

### 2.6.3     MOSAIC

The Model for Simulating Aerosol Interactions and Chemistry (MOSAIC) is a sectional aerosol model that treats aerosol

thermodynamics, size-resolved dynamic gas–particle partitioning, heterogeneous chemistry, and coagulation (Zaveri et al., 2008). It includes all major inorganic salts and electrolytes composed of $H^+$, $NH_4^+$, $Na^+$, $Ca^{2+}$, $SO_4^{2-}$, $HSO_4^-$, $CH_3SO_3^-$, $NO_3^-$, $Cl^-$, and $CO_3^{2-}$. Ions such as $K^+$ and $Mg^{2+}$ are represented by equivalent amounts of $Na^+$ while other unspecified inorganic species such as silica, other inert minerals and trace metals found in soil dust aerosols are lumped together as "other inorganic mass" (OIN). MOSAIC also includes carbonaceous species such as black carbon, primary organics, and secondary organics.

Although organic–inorganic interactions are not presently treated explicitly in MOSAIC, organics and OIN species can absorb water, which indirectly affects the overall particle pH. The gas-phase species that can partition to the particle phase include $H_2SO_4$, $CH_3SO_3H$ (methanesulfonic acid), $HNO_3$, $HCl$, $NH_3$, and any number of secondary organics.

At a given time step, the thermodynamics submodule MESA (Multicomponent Equilibrium Solver for Aerosols; Zaveri et al.,

2005a) first determines the equilibrium phase state in each size section as a function of particle-phase composition, particle size (accounting for the Kelvin effect), relative humidity and temperature, with aerosol water content calculated using the Zdanovskii–Stokes–Robinson (ZSR) method (Stokes and Robinson, 1966; Zdanovskii, 1948). The dynamic gas–particle partitioning module ASTEM (Adaptive Step Time-split Euler Method) then calculates the driving forces for mass transfer of the gas-phase species over each bin and integrates the associated mass transfer differential equations for all size sections

(Zaveri et al., 2008). The mean stoichiometric activity coefficients of electrolytes for the equilibrium phase state and mass transfer driving force calculations are estimated using the Multicomponent Taylor Expansion Method (MTEM; Zaveri et al., 2005b). Briefly, MTEM calculates the mean molal activity coefficient of an electrolyte in a multicomponent solution on the basis of its values in binary solution for all the electrolytes present in the mixture at the solution water activity ($a_w$), assuming





that $a_w$ is equal to the ambient RH. For self-consistency most of the MTEM and ZSR parameters are determined using the comprehensive Pitzer-Simonson-Clegg model (PSC) (Clegg and Pitzer, 1992; Pitzer and Simonson, 1986; Clegg et al., 1998) at 298.15 K. The PSC model is the basis of E-AIM.

In partially or fully deliquesced aerosols, the hydrogen ion molality ($m_{H^+}$) plays a central role in both equilibrium phase state and mass transfer calculations. For computational efficiency, two solution domains are considered on the basis of the so-called molal sulfate ratio, $X_T$:

$$X_T = \frac{m_{NH_4^+} + m_{Na^+} + 2m_{Ca^{2+}}}{m_{SO_4^{2-}} + m_{HSO_4^-} + m_{CH_3SO_3^-}} . \tag{9}$$

In the sulfate-rich domain (i.e., $X_T < 2$), the partial dissociation of the bisulfate ion ($HSO_4^- \rightleftharpoons H^+ + SO_4^{2-}$) and electroneutrality

equations are simultaneously solved to determine $m_{H^+}$, which is subsequently used to determine the equilibrium gas-phase concentrations of $HNO_3$, $HCl$, and $NH_3$ at the particle surface for computing their driving forces for mass transfer. In the sulfate-poor domain (i.e., $X_T \geq 2$), $HSO_4^-$ is assumed to completely dissociate to $SO_4^{2-}$, and the use of equilibrium $m_{H^+}$ to calculate the driving forces produces spurious oscillations in the mass transfer of $HNO_3$, $HCl$, and $NH_3$. This problem is solved by introducing the concept of dynamic $m_{H^+}$, which is determined by simultaneously solving surface equilibrium equations

together with the acid-base coupled condensation approximation. At a given time, the dynamic $m_{H^+}$ is thus a function of the gas/liquid equilibrium constants and mass transfer coefficients of $HNO_3$, $HCl$, and $NH_3$ along with their gas- and particle-phase concentrations. When the gases and particles reach a steady state, the dynamic $m_{H^+}$ in each size section is equal to the equilibrium $m_{H^+}$. See Sect. 7 for a further discussion of the role of particle size and mass transfer on pH.

### 2.6.4    ISORROPIA II

ISORROPIA II (Fountoukis and Nenes, 2007; http://isorropia.epfl.ch) is a computationally efficient code that treats the thermodynamics of inorganic $K^+$–$Ca^{2+}$–$Mg^{2+}$–$NH_4^+$–$Na^+$–$SO_4^{2-}$–$NO_3^-$–$Cl^-$–$H_2O$ aerosol systems. $NH_3$, $HNO_3$, and $HCl$ are considered present in the solution. The current version, version 2.3, of the code is used in this work (see code website for version history). The discrete adjoint of ISORROPIA, called ANISORROPIA, has also been developed (Capps et al., 2012) using a combination of automatic differentiation of ISORROPIA II and post-convergence treatments (to account for

discontinuities in the information flow during solution) to compute the sensitivities of all output parameters of the code to their relevant inputs with analytical precision.

ISORROPIA II can compute the equilibrium composition for two types of inputs: (a) forward, closed-system problems, in which the temperature, relative humidity and total concentrations (gas + aerosol) of aerosol precursors are known, and (b)

reverse, open-system problems, in which the temperature, relative humidity, and the concentrations of aerosol $NH_4^+$, $SO_4^{2-}$, $Na^+$, $Cl^-$, $NO_3^-$, $Ca^{2+}$, $K^+$, and $Mg^{2+}$ are used as input.

To reduce the computational complexity and increase solver speed, ISORROPIA II uses a segmented solution approach, where depending on the relative amounts of each aerosol precursor, major and minor species are defined. The equilibria of the major species together with conservation of mass and electroneutrality provide the equilibrium composition. ISORROPIA II uses mean activity coefficients for the cation–anion pairs in solution. For this, the Kusik and Meissner (1978) model for specific
ionic pairs is applied in combination with the Bromley (1973) mixing rule for activity coefficients in the multicomponent mixtures found in the aerosol. ALWC is computed as a function of RH, using the Zdanovskii–Stokes–Robinson relation (Stokes and Robinson, 1966; Zdanovskii, 1948), using a water activity database computed from the E-AIM thermodynamic model, and incorporating the effect of temperature. Although the mean activity coefficients of all major cation–anion pairs are considered, for certain species (e.g., OH$^-$, and undissociated ammonia, nitric and hydrochloric acid $NH_{3(aq)}$, $HNO_{3(aq)}$, $HCl_{(aq)}$)
unity activity coefficients are assumed due to a lack of corresponding data. Also, the first dissociation of sulfuric acid in solution is always assumed to be complete.

### 2.6.5    EQUISOLV II

EQUISOLV II is a model for calculating gas–aerosol equilibrium in atmospheric systems that contain water vapour, gases including $NH_3$, $HNO_3$, and HCl, and soluble inorganic electrolytes distributed across multiple particle size bins (Jacobson,
1999). Equilibrium is solved using a mass flux iteration technique. The model contains the major inorganic ions $H^+$, $NH_4^+$, $Na^+$, $K^+$, $Mg^{2+}$, $Ca^{2+}$, $SO_4^{2-}$, $HSO_4^-$, $NO_3^-$, $Cl^-$, and $CO_3^{2-}$ (Jacobson et al., 1999) as well as some minor and trace constituents. The thermodynamic treatment is summarized in chapter 17 of Jacobson (2005b).

Mean activity coefficients of the cations and anions in single electrolyte solutions are calculated using polynomials fit to
available data and reference values at 25 °C, supplemented by similar fits to values of enthalpies and apparent molar heat capacities of the solutions. Together, using standard relationships, these enable mean activity coefficients to be calculated for different temperatures. Mean ionic activity coefficients in mixtures are estimated using the approach of Bromley (1973), based on values for the constituent pure aqueous solutions at the total ionic strength of the mixture. The equilibrium $HSO_4^- \rightleftharpoons H^+ + SO_4^{2-}$ is calculated explicitly and is based on the same thermodynamic treatment as in E-AIM (Clegg and Brimblecombe,
1995). The approach of Kusik and Meissner (1978) is used to estimate the mean molal activity coefficients $\gamma_\pm$ ($H^+$, $HSO_4^-$) and $\gamma_\pm$ ($H^+$, $SO_4^{2-}$) in mixtures in order to obtain the equilibrium concentrations of $H^+$, $HSO_4^-$, and $SO_4^{2-}$ (see Sect. 4.2 of (Jacobson et al., 1996). This approach will not yield the same values as the treatment of Clegg and Brimblecombe (1995); see Sect. 6.1.

The relationship between the water content of aqueous aerosols containing multiple electrolytes and RH is estimated with the Zdanovskii-Stokes-Robinson relation (Stokes and Robinson, 1966; Zdanovskii, 1948), using polynomials representing single-solute molalities as a function of water activity (equivalent to equilibrium RH), and incorporating the effect of temperature, based upon the same enthalpy and heat capacity data as for the solutions referred to above.





### 2.6.6     Other thermodynamic models

Many other thermodynamic models have been developed for the prediction of atmospheric aerosol hygroscopicity and related properties, including the Gibbs free energy minimization model GFEMN (Ansari and Pandis, 1999) for inorganic aerosol
systems, ADDEM (Topping et al., 2005a,b) which emphasizes consideration of droplet size (Kelvin effect), and UHAERO (Amundson et al., 2007; Amundson et al., 2006) which allows for the computation of complex phase diagrams of both inorganic and organic systems. These models are based on the Pitzer–Simonson–Clegg model for activity coefficient and pH calculations, either directly or via polynomial expressions fitted to that model (and are thus related to E-AIM). The models SCAPE and SCAPE 2 (with NVC), for inorganic aerosol thermodynamics (Kim and Seinfeld, 1995; Kim et al., 1993a, b),
implement several activity coefficient methods, including Bromley's method (Bromley, 1973), the Kusik and Meissner method (Kusik and Meissner, 1978) and a Pitzer model. The Equilibrium Simplified Aerosol Model (EQSAM) (Metzger et al., 2002; 2006) computes gas/liquid/solid partitioning for aqueous inorganic aerosol systems, including crustal cations and iron (II, III) species. The numerical complexity of thermodynamic equilibrium models has also led to work focused on the design of computational solvers for high efficiency; for example, HETV (Makar et al., 2003) is a vectorized solver for the $SO_4$–$NO_3$–
$NH_4$ system based on the ISORROPIA algorithms. All these thermodynamic models provide a theoretical basis and mechanism to link cations and anions in aqueous solution to pH or one of its approximations.

### 3     Interactions of aerosol and cloud chemistry with acidity

The previous section highlighted how cations and anions (along with ambient conditions) drive pH in condensed phases with a focus on equilibrium conditions and models. In addition, kinetic processes such as cation and anion dissolution influence
acidity. Furthermore, the acidity of aerosols, clouds, and fogs is tightly coupled with their chemical reactivity. The pH of the atmospheric aqueous phase affects the partitioning of weakly acidic and basic gases to the condensed phase, and the rate of many multiphase chemical reactions. The chemical reactions in the atmospheric aqueous phase, in turn, modulate the pH of the aqueous phase. As a result, the acidity of aerosols, cloud droplets, and fog droplets is not only determined by thermodynamic equilibrium, but also multiphase chemical kinetics. Because of the complex nature of these couplings between
acidity and atmospheric aqueous phase chemistry, these issues are presented in more detail in a companion paper currently in preparation (Tilgner et al., 2019 *in prep*). The section highlights some important systems where acidity interacts with, and is influenced by, condensed-phase chemical reactions.

One important example of a system with chemistry-acidity feedbacks is the multiphase oxidation of sulfur dioxide ($SO_2$) to
form particulate sulfate (S(IV)→S(VI) conversion, also referred to as 'sulfur' oxidation). Sulfate makes up 15% of $PM_{2.5}$ mass globally (Sofiev et al., 2018) and is a major component of $PM_{2.5}$ in areas affected by emissions from combustion of sulfur-



containing fossil fuels. Multiphase reactions are the primary driving force for oxidation of $SO_2$ to sulfate (Calvert et al., 1985). The ionization of $SO_2$ in the aqueous phase under basic conditions enhances its uptake; the effective Henry's Law constant for $SO_2$ varies three orders of magnitude (from 17 M atm$^{-1}$ to $1.7 \times 10^4$ M atm$^{-1}$) between pH 3 and pH 6 (Sander, 2015). Therefore, sulfate production, especially under acidic conditions, is largely limited by the amount of $SO_2$ that can partition to the aqueous

phase. Meanwhile, sulfate formation is a major source of acidity in aerosols, fog and cloud droplets (Calvert et al., 1985). In the absence of buffering, S(IV)→ S(VI) oxidation pathways which are more effective at higher pH, such as oxidation of $SO_3^{2-}$ by $O_3$ (Maahs, 1983; Lagrange et al., 1994) or $NO_2$ (Lee and Schwartz, 1983; Clifton et al., 1988), will become quenched with increasing sulfate production (Fig. 3, Supplement Sect. S2). However, buffering may be significant in atmospheric waters; Collett et al. (1999), for example, demonstrated that buffering in a California fog permitted the fog pH to stay 0.3 to 0.7 pH

units higher than expected, enhancing the amount of sulfate aerosol present after the fog episode by 50%.

Another important process controlled by the acidity of the aqueous phase is the solubility of transition metal ions such as Fe(III) and Mn(II), which can catalyze S(IV) oxidation. Transition metals are ubiquitous in the atmosphere, having been observed in aerosol samples and cloud/fog/rain water collected around the globe (e.g., Bianco et al., 2017; Hsu et al., 2010).

Transition metals (particularly Fe, Cu, and Mn) are active in the aqueous-phase chemistry of clouds, fogs, and deliquesced aerosols, catalyzing reactions and affecting the oxidative capacity of the condensed phase (Deguillaume et al., 2005). Due in part to variation in how transition metal emissions are generated from different source types (e.g., mechanically generated mineral dust vs. condensation/gas-to-particle conversion of gases emitted during combustion), transition metal composition/concentration and source contributions vary across the aerosol size distribution (Deguillaume et al., 2005). This

has implications for the chemical environment and acidity that metals from different sources are exposed to, the reactions they participate in, and their potential impacts on human health.

The degree to which transition metals contribute to condensed phase reactions depends on their solubility. TMI solubility typically increases as pH decreases, although the relationship between pH and metal solubility is a complex one (Spokes et al.,

1994). Transition metals are often emitted as largely insoluble chemical species, and their solubility increases as the emitted particles 'age' via exposure to acidic gases in the atmosphere. The degree to which pH affects TMI solubility depends on the origin of the particles, degree of particle aging which alters a particle's physicochemical characteristics, and the specific metal (Deguillaume et al., 2005; Deguillaume et al., 2010). Several laboratory studies have attempted to elucidate the pH dependence of transition metal solubilization for different species. Spokes et al. (1994) found that for a Saharan dust sample, the

solubilization of Al and Fe, while strongly enhanced at lower pH, was nearly completely reversible with increasing pH. For an urban aerosol sample, some of the solubilized metals remained in solution with increasing pH, possibly due to complexation of the metal species with organic ligands. Manganese was found in both dust and urban particles to be soluble with decreased pH, with only limited reversibility as the pH was increased.





Acid-catalyzed reactions of hypohalous acids (HOX, where X = Br, Cl or I) in sea salt aerosols influence the oxidative capacity of the troposphere (Saiz-Lopez and von Glasow, 2012; von Glasow and Crutzen, 2014; Simpson et al., 2015). The reactions of HOX with other halogen ions can lead to the release of reactive halogen gases, which are involved in many key tropospheric reaction cycles. Reactions of S(IV) with HOX are another major contributor to sulfate formation in sea salt aerosols (Chen et

al., 2016; Vogt et al., 1996; von Glasow et al., 2002). These reactions acidify the aerosol (Chen et al., 2016) but may be considered a sink of reactive halogens, in that they convert HOX to their less-reactive acid form. Reactive halogen gases act directly as important sinks of key oxidants, such as $O_3$ and $HO_2$, and therefore indirectly influence other linked systems $HO_x$ (=OH + $HO_2$) and $NO_x$ (=NO + $NO_2$) (Oltmans et al., 1989; Schmidt et al., 2016; Sherwen et al., 2016). Moreover, reactive halogen gases, especially the Cl atom, can be powerful oxidants that can rapidly react with important tropospheric organic

trace gases, such as non-methane volatile organic compounds (VOCs), and dimethylsulfide (Barnes et al., 2006; Hossaini et al., 2016).

Acidity also impacts the partitioning of weak acids, including organic acids, into aqueous aerosols and cloud/fog droplets by controlling their ionization state in the aqueous phase. The hydration of carbonyl groups in compounds that also contain pH

sensitive moieties, such as α-oxocarboxylic acids, is also highly influenced by acidity (Kerber and Fernando, 2010). Increasing acidity leads to a decrease of the effective partitioning towards the particle phase of acids and to an increase in the effective partitioning of bases, and vice versa (see Fig. 4). In the case of ionizable organic species such as organic acids, key aqueous-phase oxidants, such as OH, $NO_3$ and $O_3$, can react via different possible reaction pathways and kinetics with the protonated and deprotonated forms (Buxton et al., 1988; Herrmann et al., 2010; Herrmann et al., 2015; Bräuer et al., 2019). Accordingly,

the overall reaction rate constant for oxidation of dissociating compounds can be largely pH dependent, especially for reaction with nitrate radicals. For organic acids, the overall rate constant typically increases with increasing pH and more efficient oxidation can be expected under less acidic conditions. For example, the overall second order rate constant for the reaction of nitrate radicals with formic acid (and its ionized, dissociated form) varies from close to $3.8 \times 10^5$ $M^{-1}s^{-1}$ at pH = 2.0 to $5.1 \times 10^7$ $M^{-1}s^{-1}$ at pH = 5 (Exner et al., 1994). Additionally, the increased partitioning of organic acids under less acidic conditions leads

to even higher oxidations rates.

Changes in the pH of a cloud or fog droplet can result from addition of acids or bases to the solution, through partitioning from the gas phase, collision/coalescence of droplets or aqueous reactions. The magnitude of the pH change can be strongly affected by the presence and ability of weak acids or bases to buffer against that change through proton uptake or release. A buffer is a

mixture of a weak acid and its conjugate base (e.g., formic acid and formate) or a mix of a weak base and its conjugate acid (e.g., ammonia and ammonium). The magnitude of an internal buffering effect is greatest when the solution pH is equal to the $pK_a$ ($pK_b$) of the weak acid (base) buffer. External buffering can also be important, perhaps best illustrated by the uptake of additional ammonia from the gas phase in response to a decrease in solution pH (Liljestrand, 1985; Jacob et al., 1986a; Jacob et al., 1986b).





The formation of secondary organic aerosol material in atmospheric aerosols via multiphase processes is strongly related to the acidity. Many atmospheric organic accretion reactions, such as aldol condensation (Noziere and Esteve, 2007; Noziere et al., 2010; Sareen et al., 2010; Li et al., 2011), hemiacetal and acetal formation (Jang et al., 2002; Kalberer et al., 2004; Shapiro et al., 2009; Loeffler et al., 2006), and esterification of carboxylic acids (Barsanti and Pankow, 2006) are acid-catalyzed. The acid-catalyzed reactive uptake of epoxide species, especially isoprene epoxydiols (IEPOX) (Paulot et al., 2009; Surratt et al., 2010), to aerosol water has also emerged as a significant source of secondary organic aerosol material (Lin et al., 2012; Marais et al., 2016b, Pye et al., 2013). Because the epoxidic oxygen must be protonated in concert with ring-opening, the reactive uptake of IEPOX to aqueous media is strongly pH-dependent, with the reactive uptake coefficient decreasing rapidly with

increasing pH for pH > 1 (Gaston et al., 2014). Therefore, the rate of IEPOX secondary organic aerosol (SOA) formation is slow in cloud water because of the generally higher pH compared to particles (McNeill, 2015), but given the relatively large liquid water content of clouds, which promotes dissolution, IEPOX uptake could be significant in more acidic cloud droplets (pH 3-4) (Tsui et al., 2019).

The trend of decreasing sulfate content in clouds and aerosols across North America and Europe may have implications for partitioning of inorganic (Vasilakos et al., 2018; Shah et al., 2018) and organic gases between the gas and condensed phases, and the dominant mechanisms and rates of multiphase chemical processes in the atmosphere which produce $PM_{2.5}$ mass, under future conditions. Implications of acidity changes for partitioning of semi-volatile compounds and their multiphase chemical processing are outlined in more detail in a companion paper (Tilgner et al., 2019 *in prep*).

**4    Proxies of aerosol pH**

pH has been referred to as a master variable as a result of its fundamental role in the condensed phase environment (Strum and Morgan, 1996) including the processes highlighted in the previous section. This role suggests that any study interested in understanding processes influenced by particle acidity (e.g., gas–particle partitioning, acid-catalyzed reactions, metal dissolution) should examine pH. Due to the lack of direct measurements of aerosol pH (see Sect. 2.5, 5.1) as well as

requirements for data used to calculate pH, multiple methods have been employed in the literature as surrogates or proxies for fine particle acidity. In some cases, proxies are used to infer a pH and whether particles are acidic or basic. In other cases, concepts related to pH, such as PM sensitivity to ammonia vs oxidized nitrogen, are interpreted via molar ratios of species, but pH itself is not discussed as a central concept. Since the underlying processes affecting the endpoint of interest (e.g., gas–particle partitioning of semi-volatile species that contribute to PM mass) is often directly dictated by pH, pH is implicitly

contained in such analysis. However, since proxies have only an indirect connection to the system's acidity, interpretation of results without information on pH can be challenging, incomplete, and in the worst case, incorrect.



In this section, the most common aerosol pH proxies are defined and historical context for their development and use provided. Proxies are indirectly related to pH and thus differ from the approximations highlighted in Sect. 2.3. Section 6 expands upon the information presented here by evaluating the effectiveness of each proxy using box model predictions of $pH_F$ based on ambient data.

**4.1    Proxies based on electroneutrality**

Two of the most commonly used proxy methods for aerosol pH, the cation/anion equivalent ratio (also called the cation/anion equivalence ratio or molar ratio, Hennigan et al., 2015), and the charge balance (or strong acidity, Table 2), are based upon the principle of solution electroneutrality. In both approaches, $H^+$ is assumed to balance with an excess of anions. In the case of a molar ratio, the amount of $H^+$ is assumed to scale inversely with the level of the cations relative to anions.

Application of the molar equivalent ratio to infer acidity began with studies comparing measured strong acidity to $SO_4^{2-}$ in aerosol samples (Lee et al., 1993; Lee et al., 1999; Liu et al., 1996), because direct measurements of strong acidity (e.g., via extraction and measurement by pH probe) have biases associated with sample collection and challenges with data interpretation. The concept of the cation/anion equivalent ratio was first applied to $NH_4^+$ and $SO_4^{2-}$ measurements to infer the

chemical forms of these abundant ions (Junge, 1963; Wall et al., 1988; Moyers et al., 1977; Lewis and Macias, 1980; Macias et al., 1981).

Electroneutrality and the cation/anion equivalent ratio require the incorporation of all ionic species present in solution in a given particle. However, practical measurement limitations and assumptions have led to variations of the cation/anion

equivalent used throughout the years. One such assumption is that $H^+$ accounts for the charge deficit between the water-soluble species present in particles that can be readily measured with an ion chromatograph. This typically includes five cation ($NH_4^+$, $Na^+$, $Ca^{2+}$, $Mg^{2+}$, and $K^+$) and three anion ($Cl^-$, $NO_3^-$, $SO_4^{2-}$) species (e.g., Quinn et al., 2006; Sun et al., 1998), though it occasionally includes a limited group of organic anions (Kerminen et al., 2001). A similar definition has also been applied to the non-refractory inorganic species measured with an Aerodyne Aerosol Mass Spectrometer ($NH_4^+$, $NO_3^-$, $SO_4^{2-}$, and partial

$Cl^-$) (Zhang et al., 2005; Quinn et al., 2006). If the aerosol is near neutral (pH ≈ 7) or acidity is dictated by components other than the inorganic ions in the balance (e.g., organic acids) then charge balance will not provide meaningful information (Trebs et al., 2005; Lawrence and Koutrakis, 1996). Further, the carbonate system ($CO_{2(aq)}$–$H_2CO_3$–$HCO_3^-$–$CO_3^{2-}$) must be included in the ion balance when the pH is near neutral, or above (Winkler, 1986).

The molar equivalent ratio is frequently simplified to consider only $NH_4^+$, $NO_3^-$, and $SO_4^{2-}$ (Pathak et al., 2009). The justification for this assumption is that these species represent the dominant fraction of inorganic ions, thereby controlling acidity in environments with low levels of crustal species and minor marine influence (Zhang et al., 2005). Non-volatile cations have the potential to drive large-scale patterns in $pH_F$ (see Fig. 2 as well as Sect. 8), thus making proxies without non-volatile





cations potentially invalid over large parts of the globe. Several variations of the molar equivalent ratio using these three species have been developed. The degree of sulfate neutralization (DSN), introduced by Pinder et al. (2008a), is defined as:

$$DSN = ([NH_4^+] - [NO_3^-])/[TSO_4] \tag{10}$$

where each term represents the molar concentration of each species in the particle phase. Measurements of sulfate do not

usually distinguish between bisulfate and sulfate (Solomon et al., 2014), so total sulfate, $[TSO_4]$, is used (where $[TSO_4] = [HSO_4^-] + [SO_4^{2-}]$). The bisulfate anion is often more abundant than the sulfate ion in fine particles; however, $TSO_4$ is often conceptualized has having an effective charge of negative 2.

Similar to DSN, the degree of neutralization (DON, Adams et al., 1999) has been suggested, defined as:

$$DON = [NH_4^+]/(2[TSO_4] + [NO_3^-]) \tag{11}$$

Other names have been applied to the DON, including e.g., the neutralization ratio (Lawal et al., 2018). The DON represents the ammonium associated with sulfate + nitrate while DSN represents the ammonium associated only with sulfate. In principle, this suggests that they account for different aspects of particle acidity (Pinder et al., 2008a); however, DSN and DON were highly correlated in California ($R^2 = 0.961$) and the southeastern U.S. ($R^2 = 0.978$) (this work, not shown), two locations with

very different $NO_3^-$ levels, suggesting that they represent the same physical parameter. The importance of this correlation as an indicator of pH, and its general validity, remains to be studied.

Finally, the simplest form of the cation/anion molar equivalent ratio considers particle acidity based solely on the $NH_4^+ / SO_4^{2-}$ ratio. This simplification further requires that $NO_3^-$ is relatively low (Xue et al., 2011). Acidic conditions are inferred when the

measured $NH_4^+/ SO_4^{2-}$ molar ratios is less than two (Turpin et al., 1997), as this is assumed to indicate when mildly acidic ammonium sulfate particles begin containing progressively larger amounts of acidic ammonium bisulfate. Particle acidity and pH are assumed to scale with $NH_4^+/ SO_4^{2-}$, with decreasing ratios corresponding to decreasing pH (Zhang et al., 2007).

More recently, the total ammonium to sulfate ratio has been proposed as an indicator of pH (Murphy et al., 2017):

$$TNH_4 : TSO_4 = \frac{([NH_3] + [NH_4^+])}{[TSO_4]} \tag{12}$$

Thermodynamic predictions from E-AIM were interpreted using this ratio to show that pH and ammonia volatilization increases as $TNH4 : TSO_4$ varies from below 1 to above 2 (coinciding with formation of sulfate over bisulfate). While the $TNH_4 : TSO_4$ proxy may not give a precise pH estimate, it can provide general information. For example, the ratio of $NH_3:SO_2$ emissions had been used to predict that aerosol pH may increase in the near future (Murphy et al., 2017) despite relatively

constant levels in the recent past (Weber et al., 2016).

In the case of charge balance, the amount of $H^+$, is determined from the total cation/anion deficit; for this reason, it is expressed as hydrogen ion concentration in air, i.e. total molar $H^+$ amount per unit volume of air containing aerosol particles and/or cloud



droplets, $H^+_{air}$ (nmole $m^{-3}$ air or similar). $H^+_{air}$ is related to pH; however, the two are not expected to correlate since the former is an extensive property while the latter is an intensive property of an aerosol distribution. For example, a highly acidic (pH < 1) particle with a low mass may have a lower $H^+_{air}$ than a much larger, moderately acidic (pH > 4) particle. Further, $H^+_{air}$ lacks direct modulation by particulate volume (liquid water, the solvent for $H^+$) and activity coefficients. For example, diurnal

variations in RH cause $pH_F$ variations on the order of 1 pH unit in the eastern U.S. (Guo et al., 2015; Battaglia et al., 2017; see Sect. 5.1). Like the cation/anion equivalent ration, the charge balance metric also assumes specific forms of dissociation state for multivalent ions (particularly for sulfate) and is strongly influenced by measurement uncertainty for each ionic species, especially when the aerosol is mildly acidic. These are all reasons why Guo et al. (2015) found only a weak correlation between $H^+_{air}$ and $pH_F$ ($r^2 = 0.36$).

In the 1980s – 1990s, experimental methods were developed to estimate the apparent net fine particle (<2.5 µm) strong acidity (Koutrakis et al., 1988; Koutrakis et al., 1992; Purdue, 1993). These methods, synthesized in a workshop report (EPA, 1999; Purdue, 1993), resulted in an officially documented method for estimating $H^+_{air}$ from measurements. EPA Method IO-4.1 used filter extracts in combination with measurements from a commercially available pH probe (titration methods are another

option) to calculate an estimate of $H^+_{air}$ or $H^+$ as equivalent mass of sulfuric acid. The method relies on efficient particle collection with minimal filter artifacts and low amounts of nitrate compared to sulfate (EPA, 1999). For example, ammonia could displace $H^+$ and neutralize acidic particles during collection without an efficient denuder (Koutrakis et al., 1988). Extraction and dilution of ambient samples modifies their chemical environment such that the conditions during measurement are different from those in the ambient atmosphere, potentially affecting gas–particle partitioning of total ammonium and

dissociation of weak acids including bisulfate (Purdue, 1993). Variations of this method were developed to limit the extent of dilution by measuring the pH of droplets on the surface of a hydrophobic filter using microelectrodes (Winkler, 1986; Keene et al., 2004). However, the extent of dilution is still enough to shift the sulfate–bisulfate equilibrium outside of conditions present in many atmospheric particles. The use of a strong acidity proxy in many past health studies complicates their interpretation regarding the role of acidity since $H^+_{air}$ is only a proxy for pH.

Measured aerosol composition can also be used to create charge balance estimates of $H^+$ in air. This method is also referred to as ion equivalents and sometimes strong acidity in the literature (e.g., Ito et al., 1998). When charge balance is performed on observations, it usually means a summation of all charge equivalent anions and cations in the particle. Excess molar charge equivalents of anions compared to cations are assigned to $H^+_{air}$:

$$H^+_{air,cb} = 2[TSO_4] + [NO_3^-] + [Cl^-] - ([NH_4^+] + [Na^+] + [K^+] + 2[Ca^{2+}] + 2[Mg^{2+}]). \qquad (13)$$

Thermodynamic equilibrium models use charge balance (reflecting the requirement for solution electroneutrality) as an equation or constraint together with all other considerations of species equilibria across the phases present, to obtain a unique solution. Some thermodynamic equilibrium models also use the cation/anion equivalent ratio to enhance computational





efficiency by identifying major ionic species and compositional domains (e.g., Pilinis and Seinfeld, 1987; Kim et al., 1993b; Nenes et al., 1998). However, thermodynamic models do not use charge balance from input data as a proxy for pH. Thermodynamic models (either manually, or automatically) evaluate inputs, in terms of a charge balance, to ensure that the solution obtained is atmospherically relevant – as e.g., an excess surplus of non-volatile cations may imply a strongly

alkaline solution that is not found in the atmosphere. This aspect is discussed in more detail in Sect. 6.

Operationally, $H^+_{air}$ from charge balance is subject to errors associated with measuring aerosol phase composition which are likely to be large and affect interpretation of ambient conditions. These are the same challenges that arise in thermodynamic calculations based on particulate-only inputs where biases or uncertainty in the measured species can propagate to errors in

acidity (Hennigan et al., 2015; Song et al., 2018). When $H^+_{air}$ is small in concentration or incomplete consideration of species contributing to the charge balance, charge balance may return a value of zero or negative (indicating a surplus of cations) owing to limited precision and accuracy in the measurements. Using measured aerosol composition, Murphy et al. (2017) and Hennigan et al. (2015) typically found negative values (indicating no $H^+$, theoretically impossible) of $H^+$ from charge balance and an error range large enough to span both positive and negative strong acidity estimates at almost all times. State-of-the-art

measurements are not sufficiently precise to overcome this limitation (Murphy et al., 2017). The limitations of charge balance are highlighted in Sect. 6 where ion balance estimates near zero show no correlation with pH.

Proxies for acidity have been used in a variety of applications. In the past, proxies were specifically applied in the context of acid-catalysed SOA. Evidence for this phenomenon was sought in ambient data based upon landmark studies that demonstrated

an important role of acid-catalysed reactions in forming SOA in laboratory systems (Jang et al., 2002; Paulot et al., 2009; Surratt et al., 2010). However, due to a lack of direct particle acidity measurements (see Sect. 5.1 for details and discussion), researchers used different acidity proxies to characterize the impacts on SOA formation in the atmosphere. The $NH_4^+/SO_4^{2-}$ molar ratio was used in several studies (Zhang et al., 2007; Peltier et al., 2007; Tanner et al., 2009) to infer that more acidic conditions did not enhance SOA concentrations. Froyd et al. (2010) used airborne measurements of the $NH_4^+/SO_4^{2-}$ ratio to

classify acidic and neutral particle regimes, and inferred a causal effect of acidity on isoprene organosulfate formation at low $NO_x$ in the eastern U.S. Strong associations between SOA concentrations or SOA marker compounds and $H^+$ derived from the charge balance were inferred in multiple locations (Pathak et al., 2011; Feng et al., 2012; Budisulistiorini et al., 2013; Nguyen et al., 2014). Several other studies investigated the relationship between particle acidity and atmospheric SOA using different predicted or derived measures of acidity. $H^+_{air}$ estimates based on operational extraction methods were used extensively in the

literature to link SOA formation to acidity (Surratt et al., 2007; Offenberg et al., 2009; Zhang et al., 2012). In Beijing, Guo et al. (2012b) found evidence of acid-catalysed SOA based upon correlations between secondary organic carbon (SOC) concentrations and $H^+$ concentrations (nmol m$^{-3}$) from ISORROPIA, run without gaseous inputs. Li et al. (2013) also observed correlations between SOA markers and acidity in China using pH predictions from E-AIM with aerosol inputs only. No evidence for acid-catalysed SOA formation was found in a separate study in Pittsburgh, PA that analysed correlations between





modeled $H^+$ and measured SOA concentrations (Takahama et al., 2006). Takahama et al. (2006) used gas and particle phase inputs for predictions with the GFEMN model and also observed no correlations between $pH_c$ (with $\gamma_{H^+}^{(c)}$ set to unity) and SOA. The studies employing proxies likely suffered from problems with the proxies, themselves (as discussed above), and confounding factors such as correlations between organic aerosol and sulfate, a major source of acidity, that often occur in regional pollution (Sun et al., 2011; Nguyen et al., 2015). These seemingly contradictory results were resolved once $pH_F$ was used, and the conclusions reached in these prior studies have been revisited based upon detailed understanding of the underlying chemical mechanisms and additional insight suggesting that the aerosol acidity is frequently not a limiting factor in catalysing SOA formation (Surratt et al., 2010; Xu et al., 2015; Weber et al., 2016). Acid-catalyzed isoprene SOA has now been implemented in a wide variety of box model and chemical transport model applications that now rely exclusively on thermodynamic models for acidity estimates (Pye et al., 2013; Marais et al., 2016b; Riedel et al., 2016; Budisulistiorini et al., 2017). So, while proxies can provide some information on the PM system, they should not be over interpreted as a measure of pH.

## 4.2  Gas Ratio

The Gas Ratio (GR) was defined by Ansari and Pandis (1998) to address the realization that inorganic PM concentrations do not always respond linearly to changes in sulfate concentrations (a common assumption at the time) and sought to develop a parameter that can be used for policy requiring only measurement network data. The underlying reason for this nonlinear response is that gas–particle partitioning of ammonium and nitrate is sensitive to pH. The Gas Ratio (GR) is defined in molar units as follows:

$$GR = ([TNH_4] - 2[TSO_4])/[TNO_3] \tag{14}$$

and uses total ammonia ($NH_3 + NH_4^+$), and total nitrate ($HNO_3 + NO_3^-$). The numerator of the GR is sometimes referred to as free ammonia, because it is the amount of $TNH_3$ that would be available to form $NH_4NO_3$ under the simplistic assumption that appreciable $NH_4NO_3$ does not form when the molar ratio of $TNH_4$ to $TSO_4$ is less than two (i.e., the stoichiometric ratio of $(NH_4)_2SO_4$). The concept of "free ammonia" is discussed in greater detail below (Sect. 6.2.3). The GR has not been used explicitly as a proxy for aerosol pH, but it has been used extensively to define aerosol and composition regimes that relate to acidity.

Ansari and Pandis (1998) characterized inorganic PM response for changes in $TNH_4$, $TNO_3$, and $TSO_4$ as functions of GR, temperature, RH, and system concentrations. Their analysis determined critical values of GR that defined boundaries of the PM response regimes. As West et al. (1999) showed, the GR still requires complementary thermodynamic modeling to robustly explore the PM response for large sulfate reductions. Other applications of the GR include calculation of GR as a function of altitude (Adams et al., 1999), characterization of the sulfate-nitrate-ammonia-water aerosol system in the context of natural and transboundary pollution over the U.S. (Park et al., 2004), and exploration of the sensitivity of aerosol nitrate to changes in temperature, RH, $TNH_4$, $TNO_3$, and sulfate for Pittsburgh (Takahama et al., 2004). In addition, Blanchard and Hidy (2003)



considered the GR in a study of the response of nitrate to changes in TNH$_4$, TNO$_3$, and sulfate in the southeastern U.S. A relationship was demonstrated between the GR and an excess-NH$_3$ indicator that resembles free ammonia, but accounts for chloride and non-volatile cations. Pun and Seigneur (2001) used box model simulations to demonstrate that nitrate concentrations in California's San Joaquin Valley (SJV) would be sensitive to HNO$_3$ levels but not NH$_3$.

Pinder et al. (2008a) investigated the response of PM to emissions of NOx, SO$_2$, and NH$_3$, and demonstrated with AIM thermodynamic modeling that NH$_4$NO$_3$ can form at low temperature even when the GR is less than zero, in contrast to previous assumptions. To address this limitation, they developed an adjusted GR (*adjGR*) that modified the calculation of free ammonia:

$$\mathrm{adjGR} = \frac{[\mathrm{TNH_4}] - \mathrm{DSN} \times [\mathrm{TSO_4}]}{[\mathrm{TNO_3}]} = \frac{[\mathrm{TNH_4}] - ([\mathrm{NH_4^+}] - [\mathrm{NO_3^-}])}{[\mathrm{TNO_3}]} \tag{15}$$

where DSN is the degree of sulfate neutralization. Using a chemical transport model, Pinder et al. (2008a) demonstrated that the response of nitrate concentrations to changes in SO$_2$ and NH$_3$ emissions could be reasonably represented as a function of the adjGR and GR, but the adjGR provided a better fit for cases where DSN differs significantly from a value of two. In a separate study, Pinder et al. (2008b) found a strong relationship between the adjGR and the sensitivity of inorganic PM concentrations to NH$_3$ levels at sites in the eastern U.S.

Additional metrics similar to GR have been defined. Wang et al. (2011) considered the GR and adjGR in a study of the sensitivity of inorganic aerosols to NH$_3$ in mainland eastern China. They also defined a new indicator, the Flex Ratio (FR), calculated based on predictions of a statistical response-surface model developed from about 100 CTM simulations probing the sensitivity of PM to NH$_3$ and NO$_x$ emissions (See Xing et al. (2018) for a precise definition). Nitrate concentrations are
more sensitive to NH$_3$ than NO$_x$ emissions for FR > 1, and nitrate concentrations are more sensitive to NO$_x$ than NH$_3$ emissions for FR < 1. The FR provides a relatively precise estimate of the transition point between NH$_3$-rich and NH$_3$-poor conditions for existing NOx emissions levels. However, use of the FR remains limited due to its dependence on the availability of response-surface model predictions, which are currently limited to regions in Asia (e.g., Xing et al., 2011; Zhao et al., 2015). Wang et al. (2011) reported that the nitrate response regimes indicated by the GR and FR were qualitatively consistent in their
study.

The studies described above, and other studies (e.g., Campbell et al., 2015; Dennis et al., 2008; Lee et al., 2006; San Martini et al., 2005; Zhang et al., 2009), have used the GR and adjGR to understand the response of the sulfate-nitrate-ammonium-water aerosol system to changes in precursor concentrations and emissions. The GR provides a reasonable indication of the
sensitivity of inorganic PM to changes in TNH$_4$, TNO$_3$, and sulfate in many cases. However, they do not consider the role of non-volatile cations and universally applicable ranges of GR for demarcating response zones are difficult to define due to the dependence on other factors (temperature, RH, and system concentrations). The GR has not been used explicitly as a proxy for particle acidity, but the response of nitrate to precursor concentrations can be represented as a function of GR using S-





shaped curves (Pinder et al., 2008a) that resemble the sigmoid curves reported for gas–particle partitioning of $TNO_3$ as a function of pH (Guo et al., 2016). Therefore, some relation is expected between these indicators and particle acidity which is shown in Sect. 6.2.3.

### 4.3 Semi-volatile species partitioning

Aerosol pH affects the gas–particle partitioning of semi-volatile acidic and basic compounds in the atmosphere, including inorganic ($HNO_3$, $NH_3$, HCl) and organic (amines, formic, acetic, and oxalic acid) species. The underlying reason why pH affects partitioning is that the protonated and deprotonated forms of the species vary considerably in their volatility (Keene et al., 1998; Meskhidze et al., 2003; Guo et al., 2017b). Based on this insight, estimates of aerosol pH can be derived from simultaneous measurements of the abundance of a compound in the gas and condensed phases, assuming that the species in
question are in thermodynamic equilibrium. For example, $HNO_3$ partitioning is determined by:

$$HNO_3 \text{ (g)} \rightleftharpoons HNO_3 \text{ (aq)} \tag{R1}$$

$$HNO_3 \text{ (aq)} \rightleftharpoons H^+ + NO_3^- \tag{R2}$$

Reactions R1 and R2 are characterized by the Henry's law constant ($K_H$) and the acid dissociation constant ($K_a$), respectively. These equilibrium expressions can be combined (see also the derivations provided in the supporting information of Guo et al.,
2017b and Nah et al., 2018) and rearranged to yield:

$$a_{H^+} = \frac{K_H K_a p_{HNO3}}{a_{NO_3^-}} \tag{16}$$

where $p_{HNO3}$ is the partial pressure of nitric acid, $a_{NO_3^-}$ is the nitrate activity in deliquesced aerosols, and $a_{H+}$ is the $H^+$ activity in the aqueous aerosols. pH can be expressed as the fraction of total nitrate in the particle ($F_{p,NO3} = [NO_3^-]/[TNO_3]$ by moles), the gas constant ($R$), temperature, equilibrium constants, and molality-based activity coefficients for species $i$ ($\gamma_i$):

$$pH = -\log_{10}\left(\frac{K_H K_a (1-F_{p,NO3})RT}{\gamma_{NO3} - F_{p,NO3}}\right) \tag{17}.$$

An analogous version of Eq. (17) could be applied to any monovalent acid/anion pair (e.g., hydrochloride acid/chloride partitioning). pH based on $F_{p,NH4}$ ($[NH_4^+]/[TNH_4]$ by mole) is slightly different.

$$pH = -\log_{10}\left(\frac{\gamma_{NH4} + F_{p,NH4}}{K_H K_a (1-F_{p,NH4})RT}\right) \tag{18}$$

where $K_a$ is the acid dissociation constant for $NH_4^+$:

$$NH_4^+ \text{ (aq)} \rightleftharpoons H^+ \text{ (aq)} + NH_3 \text{ (aq)} \tag{R3}$$

pH from Eq. (17) and (18) becomes uncertain when $F_p$ is in the vicinity of one or zero, especially when considering the effects of observational uncertainty.

Current analytical techniques allow for the direct measurement of nitric acid while the aqueous aerosol nitrate concentration,
can be derived from the aerosol nitrate mass concentration (µg m$^{-3}$, directly measured) by using the ALWC (measured or





estimated). Conversion from aqueous concentrations to activity requires activity coefficients, which can be computed (e.g., Clegg et al., 1992), obtained from one of the aerosol thermodynamic equilibrium models, or approximated with a relevant ion pair ($pH_\pm$) (Sect. 2). Most studies to date have simplified the above expressions by assuming activity coefficients of unity (molal basis), with aqueous concentrations replacing species activities in Eq. (16) above (Meskhidze et al., 2003; Keene et al.,

2004) and the resulting pH (Eq. 17–18) becoming $pH_F$.

Gas–particle partitioning of $TNH_4$, total chlorine ($TCl = HCl + Cl^-$), and $TNO_3$ are all candidates for estimating pH. The approach was first discussed in relation to the pH of sea salt aerosols in the marine boundary layer (Keene et al., 1998). However, the phase partitioning behaviors of HCl and $HNO_3$ were inconsistent, as measured $HNO_3/NO_3^-$ implied a pH in the

~1 – 2 range, but $HCl/Cl^-$ levels implied a pH much higher (Keene et al., 1998). These discrepancies were postulated to result from positive biases in the $HNO_3$ (*g*) measurements, uncertainties in the thermodynamic constants, and kinetic limitations to mass transfer (deviation from equilibrium); however, no mention about the effects of mixing state and ability to predict liquid water content was discussed. The first quantitative estimates of aerosol $pH_c$ (molarity basis, see Eq. 2) via partitioning were done by Keene and Savoie (1998) and used $HCl/Cl^-$ partitioning to characterize sea salt particles mixed with anthropogenic

pollution. Meskhidze et al. (2003) used measured $HNO_3/NO_3^-$ partitioning to quantify aerosol $pH_F$, with specific applications to Fe solubility. Keene et al. (2004) extended the analysis and compared the size-dependent aerosol $pH_c$ predicted by the phase partitioning of $NH_3$, HCl, and $HNO_3$ in marine air. They observed general agreement in the pH predictions based on $HNO_3$ and HCl partitioning, while acidity based on $NH_3$ partitioning was systematically lower by ~1 – 2 pH units. Keene et al. (2004) assumed $\gamma_{H^+}^{(c)}$ was unity in their calculations and noted this as a likely source of uncertainty. In a study a decade later, Young

et al. (2013) compared the aerosol $pH_c$ (by size, with $\gamma_{H^+}^{(c)}$ computed by E-AIM) predicted by $NH_3$, $HNO_3$, and HCl phase partitioning at a continental location near Denver, CO. In this study, aerosol $pH_c$ derived from $NH_3$ and $HNO_3$ partitioning generally agreed, while $pH_c$ predicted from HCl partitioning was systematically higher by ~1 – 2 pH units than the other methods. The authors attributed these differences to order-of-magnitude uncertainties in the Henry's constant of HCl (Sander, 2015). Similar problems with $HCl/Cl^-$ partitioning were observed in the northeast U.S., potentially due to uncertainties in the

thermodynamic properties of HCl or non-volatile cation measurement artifacts (Haskins et al., 2018). Aerosol pH ($pH_c$ and $pH_x$ both evaluated) derived from $NH_3$ partitioning agreed well with E-AIM and ISORROPIA predictions under highly polluted conditions in Mexico City (Hennigan et al., 2015). In this study, $HNO_3$ and HCl data were not available to compare with the $NH_3$ partitioning calculations, so an evaluation of differences, as was performed by Keene et al. (2004), was not possible.

The partitioning of semi-volatile carboxylic acids should also provide insight into aerosol pH conditions (Keene et al., 2004). Nah et al. (2018) found that oxalic acid partitioning was consistent with its known thermodynamic properties, and thus represented a reasonable proxy for aerosol pH in a study in the southeast USA. However, in the same study, the partitioning





of formic and acetic acid implied aerosol pH levels that were ~5 – 6 pH units higher than that predicted by ISORROPIA or other semi-volatile species, including oxalic acid (Nah et al., 2018). The reasons for such dramatic differences are not known, but the formation of organic salts may be one explanation (Paciga et al., 2014; Hakkinen et al., 2014; Tao and Murphy, 2019a).

Efforts to reconcile some of the above differences using model simulations are challenged by large uncertainties in the emissions (Kelly et al., 2016) and secondary formation (Millet et al., 2015) of key species. Although few other comparisons of aerosol pH based upon direct measurements of semi-volatile partitioning have been conducted, semi-volatile species partitioning is used as a key evaluation of thermodynamic model predictions of pH. Guo et al. (2015) compared ISORROPIA predictions of ALWC and $NH_3$ partitioning with direct measurements in the southeastern U.S. during SOAS. ALWC is

required for accurate calculations of $a_{H^+}$ (hence pH) while $NH_3$ partitioning (for aqueous particles) is dependent upon pH. Guo et al. (2015) observed excellent model–measurement agreement for both ALWC and $NH_3$ partitioning, suggesting that their pH predictions were similarly accurate. Comparisons of modeled and measured semi-volatile species partitioning are now regularly used to check thermodynamic model predictions of aerosol pH (e.g., Guo et al., 2016; 2017b; 2018b; Murphy et al., 2017; Nah et al., 2018; Song et al., 2018).

Practical measurement limitations have precluded more extensive evaluations of direct pH predictions with partitioning predictions of pH. First, the method requires measurable concentrations of a compound in both gas and condensed phases. Conditions in which a species is partitioned almost entirely to one phase preclude the application of this method. For example, Keene et al. (2004) were unable to make quantitative estimates of aerosol pH based upon formic or acetic acid partitioning

since the aerosol concentrations were frequently below method detection limits. In certain environments, $HNO_3$ is partitioned almost entirely in the gas phase, limiting its use for aerosol pH determinations. This limitation also extends to the ability to test and validate the thermodynamic models, whose phase partitioning predictions are seen as a key model check since direct pH measurements are not yet applied to ambient particles (Liu et al., 2017). Second, this approach applies the instantaneous equilibrium assumption, which does not always hold (see Sect. 7 for a discussion on deviations from equilibrium). The

partitioning method is susceptible to biases associated with sampling semi-volatile components in either the gas or particle phases. Due to their semi-volatile nature, measurements of these compounds can suffer positive (overestimation) or negative (underestimation) artifacts, and the challenges associated with measuring organic (Turpin et al., 2000; Eatough et al., 2003; Lipsky and Robinson, 2006) and inorganic (Ashbaugh and Eldred, 2004; Talbot et al., 1990; Pathak et al., 2004; von Bobrutzki et al., 2010) semi-volatile compounds have been well documented. Despite major advances in analytical capabilities, such

challenges persist (Dhawan and Biswas, 2019; Guo et al., 2016; Tao and Murphy, 2019a). Finally, the assumption regarding the phase state of the aqueous aerosol (i.e., whether it is on the efflorescent or the deliquescent branch of the water uptake curve, given hysteresis) has a profound impact on the amount of liquid water present in the aerosol, which in turn affects the ionic strength and ions speciation in solution (hence pH). Although studies have confirmed that the liquid water content can be in agreement with observations (Guo et al., 2015), and in combination with semi-volatile partitioning measurements provide





a well-constrained estimate for aerosol pH, the ALWC predictions central to the pH calculations are not routinely evaluated. While organics do not appreciably affect pH in particles consisting of a single aqueous phase (Battaglia et al., 2019), the changes in acidity as particle viscosity increases, shifting the phase towards semi-solid and glassy forms and creating a heterogeneous distribution of $H^+$ across the particle, are unknown. All these factors eventually limit the precision and range of

atmospheric conditions for which pH estimates based on semi-volatile species partitioning can be used. The accuracy of partitioning as well as other proxies an estimate of pH are further discussed in Sect. 6 (specifically Sect. 6.2.3 and 6.3) based on box model calculations.

## 5    Atmospheric observations of acidity

The preceding sections have alluded to some of the distinct challenges associated with particle and cloud pH measurements.

Measuring the pH of nominally sub-10 µm atmospheric aerosols is not routinely possible, as the methods typically available for bulk condensed phases (e.g., electrochemical pH probes) cannot be applied to the liquid phase of a single particle or a population of particles as a result of the extremely low levels of liquid water. Another issue is the highly concentrated nature of aerosol solutions, whose ionic strengths are often orders of magnitude above the maximum ionic strength currently accepted by the IUPAC definition (0.1 M). As a result, limited direct measurements of aerosol pH exist (Sect. 5.1.1 – 5.1.2) and

observationally-constrained estimates are usually created from thermodynamic models (Sect. 5.1.3 – 5.1.4). The same limitations do not exist for fog/cloud water or precipitation, where larger sample volumes can be collected from accessible clouds, the associated water is not in equilibrium with the gas phase (although evaporation artifacts may still cause biases), and solutions are dilute enough to allow for a direct pH measurement. The latter has been done with electrochemical pH probes for decades (Sect. 5.2).

**5.1    Observed aerosol acidity**

Challenges associated with measuring semi-volatile species (Sect. 4.3) and maintaining aqueous concentrations found in ambient particles have limited direct measurements of aerosol pH for many years. The abundance of many non-volatile ionic components of the atmospheric aerosol (e.g., sulfate or sodium) can be measured. However, unperturbed equilibrium contact with the gas phase cannot be easily maintained. Determining $H^+$ activity (or molality in the case of $pH_F$) requires knowledge

of water content, which for non-glassy aerosol is in chemical equilibrium with the gas phase (e.g., Seinfeld and Pandis, 2016). Thus, if the ALWC changes between sampling and analysis, as often occurs in routine monitoring networks, the pH of that particle can shift. A second challenge is that one of the most important cations present in submicron aerosol, $NH_4^+$, is largely in equilibrium with gas-phase $NH_3$, so perturbations during collection and processing may result in large evaporation/condensation biases (e.g., Guo et al., 2018a).





Despite the obstacles related to measuring the pH of aerosols, the importance of aerosol pH has motivated efforts to more directly probe the pH of aerosols. Direct measurements may either provide an ensemble or bulk average pH value (Li and Jang, 2012; Jang et al., 2008; Ganor et al., 1993; Craig et al., 2018), single particle values (Craig et al., 2017; Rindelaub et al., 2016b), or intra-particle pH values (Wei et al., 2018). Considerable effort has also been spent to develop pH estimates using a

combination of thermodynamic modeling with measurements of aerosol and gas-phase composition (e.g., Guo et al., 2018a; Guo et al., 2017a; Guo et al., 2015; Bougiatioti et al., 2016; Song et al., 2018; and others).

### 5.1.1 Bulk pH measurements

For bulk pH values to be reasonable, composition and partitioning of semi-volatiles among the particles in the population should be relatively uniform (see Sect. 7 for a discussion on the role of particle mixing state). For a set of submicron particles

with homogeneous composition (i.e. an internal mixture, as often found in aged aerosols), this assumption is often satisfied, particularly at higher RH. The simplest bulk method, first utilized in the late 1980s involved adding a known volume of water to a filter and then utilizing a standard electrochemical pH probe to infer so-called strong acidity (Koutrakis et al., 1988). As discussed here and in Sect. 4.1, given the semi-volatile nature of water and non-conservative nature of the $H^+$ ion (Saxena et al., 1993), this approach has significant shortcomings.

The use of pH sensitive indicators has been one of the most common approaches to avoid sample modification. Although only a handful of studies have used indicators for aerosol studies, multiple approaches have been applied. Jang et al. (2008) used a Teflon filter where the dye metanil yellow was taken up by the filter prior to sampling aerosol. Particle mass was determined gravimetrically, and pH sensitive indicators were analysed with a UV-Visible (UV-Vis) spectrometer. Through a calculation

combining mass, absorption features of the protonated (at 545 nm) and deprotonated (at 420 nm) form, and particle volume from a simultaneous aerosol size distribution, the mass of $H^+$ was determined and then converted to pH. The RH had to be precisely controlled, since any change in water content would limit the reliability of the results. The authors noted the need for an online approach to avoid these complications. This work was expanded in Li and Jang (2012) with the use of an optical flow chamber to help control the RH and improve transfer from the collection point in a Teflon aerosol chamber to the UV-

Vis spectrometer for measurement. For a tightly controlled system, the dyed filter approach provided particle pH, but the method has not been reliably applied in the complex ambient atmosphere.

Colorimetric methods have been utilized at different points to determine the pH of aerosols and cloud droplets. Ganor et al. (1993) used pH paper on a four-stage impactor (>10 µm, 3.0-10.0 µm, 0.9-3.0 µm, and < 0.9 µm) to probe larger "haze"

particles in Israel under conditions with RH exceeding 80%, cloud droplets, and fog droplets. Two types of pH paper were used covering two pH ranges, 0.5-5.0 and 5.0-9.0 (indicator not given), with 0.5 pH unit resolution. For these measurements, a size-dependence in acidity was observed, with pH decreasing from 5 to 2 from cloud/fog to large particles using visual identification. Submicron haze aerosol pH was characterized as having an overall pH of 1.5 – 2.5 in Israel (Ganor et al., 1993).



Craig et al. (2018) recently quantified the measurement of aerosol pH with a precision of 0.1 pH units using particles impacted on pH paper followed by rapidly taking a cell phone picture later analysed with a simple image processing script in MatLab. For their work, thymol blue (pH = 0 – 2.5) and methyl orange (pH = 2.5 – 4.5) indicator dyes on paper were used for both model aerosol in the laboratory and ambient samples at a forested site in Northern Michigan (pH = 1.5 – 3.5) and in Ann Arbor,

Michigan (pH = 3.0 – 3.5). Water and ammonia volatilization due to increased surface area to volume ratios in smaller particles explains the observed increase of particle acidity with decreasing particle size but also the inadequacy of pH paper method to measure the pH of very small particles (Craig et al., 2018). Ganor et al. (1993) and Craig et al. (2018) mentioned that for the colorimetric approach to be effective, the particles must be aqueous with sufficient aerosol water to substantially wet the indicator paper, which was not always the case for ambient sampling. The atmospheric samples in Northern Michigan (Craig

et al., 2018) covering three size ranges (2.5-5.0, 0.4-2.5, and < 0.4 µm) and measuring to smaller sizes than in Ganor et al. (1993), showed a distinct decrease in pH toward smaller size. For the smallest stage in Craig et al. (2018), variation in pH was observed across the samples with values ranging from 1.5 – 3.0, possibly due to differences in pH between individual particles in that size range, though further investigation is needed (see Sect. 7 for a discussion on the relationship between particle size and pH). Craig et al. (2018) made comparisons to the bulk solutions with the thermodynamic model E-AIM, finding good

agreement. When applied to the particle data, the thermodynamic model ISORROPIA predict a $pH_F$ lower than measured by roughly a pH unit, while E-AIM was roughly 2 pH units lower. Further testing is needed between thermodynamic models and colorimetric methods to explore differences, particularly since the high ionic strengths in particles that may affect organic dye activity (e.g., via issues raised in Sect. 2.5).

### 5.1.2    Single particle pH measurements

Several emerging methods have the potential to provide even greater insight into the pH of individual particles but have been focused on model systems. Determining single particle pH is desirable as the variation of pH values for individual particles from the population-level average is not well known. Even a few acidic particles can dominate the average pH value for a population in an environment with fresh emissions (e.g., urban area) where the particles have not reached equilibrium with the gas concentrations surrounding them (Craig and Ault, 2018). This heterogeneity may be less important in a regional

background that has experienced significant atmospheric processing (Guo et al., 2018a). One of the first approaches to single droplet pH measurement was Ganor (1999), which impacted cloud and fog droplets collected on the four stages of a cascade impactor on to a cleaved calcite ($CaCO_3$) crystal. If an acidic droplet containing sulfate was impacted, the microchemical reaction produced gypsum ($CaSO_4 \cdot 2H_2O$) crystals. Fog droplet pH values of 3.0 and 4.0 were observed in Israel for sizes of 1-2 µm and 3-9 µm, respectively. A second single particle method (Dallemagne et al., 2016), used for studying liquid–liquid

phase separation (LLPS) in larger particles (10-30 µm) with optical microscopy methods, applied a fluorescent indicator (Oregon Green 488 carboxylic acid, succinimidyl ester) to probe pH before and after phase separation via fluorescence microscopy (see Sect. 7.3 for discussion of organic species and their role in acidity).



A recently developed approach for probing the pH of individual droplets is the acid-conjugate base method (Rindelaub et al., 2016b), which calibrates the peak area for the acid and conjugate base measured with Raman microspectroscopy to molar concentrations, which along with the acid dissociation constant ($K_a$) and activity coefficient calculations can be used to determine the activity of the $H^+$ ion. Rindelaub et al. (2016b) originally applied this to monitoring sulfate and bisulfate in proximity to the $pK_a$ of 2. Craig et al. (2017) expanded this method to cover a range of systems (nitric acid–nitrate, bioxalate–oxalate, acetic acid–acetate, and bicarbonate–carbonate, as well as an inorganic–organic mixture). From these systems, a pH of -1 to 10 could be probed, covering the full range of atmospherically- relevant particle pH (Fig. 2). The acid-conjugate base method originally made use of activity coefficients from the extended Debye-Hückel method (Rindelaub et al., 2016b), but has been expanded to other methods in subsequent publications (Craig et al., 2017). In addition to using simplistic model systems, recent work has shown that the protonation state of hydroxyl functional groups (–OH) in organic molecules from SOA formation can be used to estimate pH. Bondy et al. (2018) showed that by modulating pH the protonation state of 2-methylglyceric acid or 2-methylglyceric acid sulfate ester could be used to roughly identify the pH of a system, though further quantification is necessary to assess the accuracy of this approach. This was shown by monitoring the carbonyl (C=O) stretch of the carboxylic acid group on 2-methyl glyceric acid when protonated versus the asymmetric and symmetric vibrations of the carboxylate group ($COO^-$) when deprotonated. While these vibrations have been successfully measured in ambient aerosols collected during the SOAS summer 2013 campaign, none of the acid-conjugate base methods has been applied on ambient aerosol to evaluate aerosol pH.

Wei et al. (2018) recently probed intraparticle pH variation using Surface-Enhanced Raman Spectroscopy, and showed pH is not always uniform within an individual particle. The researchers took the acid-conjugate base method one step further with nanoprobes by functionalizing a gold nanoparticle dimer with an indicator molecule (4-mercaptobenzoic acid) and monitoring the acid versus conjugate base form of the indicator molecule in 20 µm phosphate buffer solution droplets (pH = 7 − 11). They showed that within a single particle, a gradient of up to 3.6 pH units could be observed between the core (higher pH) and the exterior few microns of the particle (lower pH) due to accumulation of protons at the air/water interface. Such substantial differences in pH within a single particle are unexpected (as they imply a large chemical potential difference of $H^+$ ions) and further independent measurements are needed to confirm this behavior in atmospheric particles.

### 5.1.3   Considerations for the development of observationally-derived pH estimates

Only very recently have methods for direct measurement of aerosol pH become available, and they require considerable development before they become routine and generally applied. Until now, most information generated on aerosol acidity relies on measurements of particle composition and gas-phase semi-volatiles, in addition to thermodynamic equilibria or kinetic modeling (see Sect. 2.6). $pH_F$ (Table 1) is the most comment approximation of pH reported in literature. The accuracy of these estimates depends on and is evaluated by the agreement between observed and modeled gas–particle partitioning of pH-sensitive species – typically $TNH_4$, $TNO_3$, and $TCl$, as well as accuracy of predicted aerosol liquid water (Guo et al., 2015;





Guo et al., 2017b; Guo et al., 2018b; Meskhidze et al., 2003; Song et al., 2018). $NH_3$ is an ideal species to measure since gas–particle partitioning of $TNH_4$ is sensitive to most ambient pH levels and ammonium is often the dominant particle cation. However, ammonia measurements are challenging and often not co-located with aerosol composition measurements. Bougiatioti et al. (2016) estimated that neglecting gas-phase $NH_3$ levels of about 0.1 to 0.7 μg m$^{-3}$ in the thermodynamic

equilibrium calculation of fine aerosol pH could lead to an underestimation in the $pH_F$ of around 0.5 units, while Guo et al. (2015) found neglecting gas-phase $NH_3$ leads to an underestimation of $pH_F$ by one unit. Although this magnitude of underestimation is not universally applicable when $NH_3$ is missing from the thermodynamic calculations, it may be a reasonable bound for most of the atmosphere. Weber et al. (2016) and Guo et al. (2017a) estimated that on average, a 5-fold to 10-fold increase in the $NH_3$ levels leads a one unit change in pH. For cases were more than 90% of total ammonium is in

the aerosol, neglecting gas-phase $NH_3$ should give similar (1 unit) underestimations in pH. Accurately bounding the error requires aerosol pH calculations to be evaluated against observations in both the aerosol and gas phases.

Routine air quality monitoring networks provide limited opportunity for pH estimation. European networks (European Monitoring and Evaluation Programme (EMEP)/EBAS, http://ebas.nilu.no/, and Research Infrastructure for the observation

of Aerosol, Clouds and Trace Gases (ACTRIS), http://actris.nilu.no/Data/Policy/) provide mainly bulk $PM_{10}$ aerosol chemical composition data which prohibits equilibrium assumptions due to the contributions from coarse mode mass. Most routine networks in the U.S. measure $PM_{2.5}$, but the Interagency Monitoring of Protected Visual Environments (IMPROVE, http://vista.cira.colostate.edu/Improve/) network lacks measurements of ammonium and the Chemical Speciation Network (CSN, https://www.epa.gov/amtic/chemical-speciation-network-csn) ammonium is biased low compared to other networks

and measurements (Pye et al., 2018; Silvern et al., 2017). The Clean Air Status and Trends Network (CASTNET, https://www.epa.gov/castnet) provides measurements of $NH_4^+$, $NO_3^-$, and $SO_4^{2-}$, along with $HNO_3$, $SO_2$, and base cations, at approximately 92 sites across the U.S.. However, CASTNET does not size select particles. For IMPROVE and CSN, concurrent relevant gas-phase measurements are generally not available. Up until 2015, the Southeastern Aerosol Research and Characterization Study (SEARCH) in the United States provided complete hourly particle and gas-phase semi-volatile

measurements with high accuracy and precision (Edgerton et al., 2006; Hansen et al., 2003). Recent instrumental developments (MARGA, Liu et al., 2014) enable concurrent measurements of inorganic substances that influence aerosol pH and are present in the gas phase (namely $HCl$, $HNO_3$, $HNO_2$, $SO_2$, $NH_3$) or in the aerosol phase ($Cl^-$, $NO_3^-$, $SO_4^{2-}$, $NH_4^+$, $K^+$, $Ca^{2+}$, $Mg^{2+}$). Online $NH_3$ instruments are becoming more available and with sensitivity down to very low concentrations (von Bobrutzki et al., 2010). In combination with aerosol data, gas-phase measurements generate datasets that can constrain aerosol pH.

For submicron aerosol ($PM_1$), the development and operation of aerosol mass spectrometers (AMS, Zhang et al. 2007; Jimenez et al.; 2009) and aerosol chemical speciation monitors (ACSM, Ng et al. 2011) during the last decade provides a powerful tool to build a database of nonrefractory submicron aerosol composition. This database could constrain aerosol pH when complemented by gas-phase measurements, mainly of $NH_3$, as well as measurements of the non-volatile, refractory aerosol





components. Such studies, usually performed on a campaign basis, have enabled the estimation of aerosol pH at various locations around the globe, including the southeastern US, Greece, and mainland China (Table S6), and ACSM measurements could be more routinely available in the future (e.g., ACTRIS, Schmale et al., 2017). However, the contribution of organosulfates and organonitrates to AMS measured total sulfate and nitrate (Farmer et al., 2010; Dovrou et al., 2019) must

be considered to provide robust inorganic aerosol predictions. In locations such as the eastern U.S. in summer where organosulfates already account for 15% of total sulfate (Riva et al., 2019), AMS measured total sulfate, when used in a thermodynamic model as inorganic sulfate, can lead to erroneous predictions of particle composition and thus pH (Pye et al., 2018).

The largest challenge using network or campaign data to estimate pH is that simultaneous information on $NH_3$ and $NH_4^+$ is often not available. The National Atmospheric Deposition Program/Ammonia Monitoring Network (AMoN) measures $NH_3$ on a biweekly schedule at 104 active sites in the U.S. (as of July 29, 2019) (http://nadp.isws.illinois.edu/AMoN). Colocation with CASTNET $NH_4^+$ measurements provides $NH_3 + NH_4^+$ at approximately 70 sites (Puchalski et al., 2019). The U.K. National Ammonium Monitoring Network (NAMN) has been established to measure the spatial distribution and long-term

trends in atmospheric gaseous $NH_3$ and aerosol $NH_4^+$ (Sutton, 2001; Sutton et al., 1998). In 2016, the network measured gaseous $NH_3$ on a monthly basis by DEnuder for Long Term Atmospheric (DELTA) sampling at 56 sites and by Adapted Low-cost Passive High-Absorption (ALPHA) samplers at a further 38 sites, 9 of these sites used for calibration, in order to quantify the spatiotemporal varibility of $NH_3$ and $NH_4^+$ concentrations and deposition across the UK, http://www.pollutantdeposition.ceh.ac.uk/content/ammonia-network). Observations show spatially variable changes in $NH_3$

with both reductions, mainly to the north, and increases, to the south between 1997 and 2007. In nature reserve areas in the Netherlands, atmospheric ammonia concentrations have been monitored by the Measuring Ammonia in Nature (MAN) network (http://man.rivm.nl) since 2005 (Lolkema et al., 2015). In 2015 that network contained 60 natural areas with a total of 236 sampling points were $NH_3$ was monitored using passive samplers. While no significant trend has been found on average, at 6 stations a significant increasing trend was recorded.

The amount of knowledge on the atmospheric distribution of $NH_3$ has increased rapidly in the satellite era since the $NH_3$ tropospheric column observations from space by the Atmospheric Infrared Sounder (AIRS) sensor on-board the Aqua satellite (Warner et al., 2016), the Infrared Atmospheric Sounding Interferometer (IASI) (Clarisse et al., 2009), and the Cross-track Infrared Sounder (CrIS) (Shephard and Cady-Pereira, 2015) became available. These satellite observations have shown the

high $NH_3$ levels associated with animal feeding operations and fertilizer applications as well as biomass burning (especially wild fires). These data provided a global view of $NH_3$ column distribution, construction of which was inhibited by the spatial and temporal variability of $NH_3$ concentrations reflecting its spatially varying sources and its short tropospheric lifetime of up to a couple of hours (Dentener and Crutzen, 1994). While there is not yet established methodology to derive aerosol pH from space observations, the improvement of near surface information on atmospheric composition in combination with ground-



level observation network data (perhaps even augmented by model fields of such data) will likely advance our understanding. Considerably more challenging, however, is constraining the vertical distributions of aerosol pH – especially since the lower temperatures and less abundant water progressively challenge the assumption of thermodynamic equilibrium and may require the treatment of particle history and hysteresis (e.g., Wang et al., 2008).

### 5.1.4    Spatial and temporal variability of aerosol pH

Current observationally-constrained estimates indicate fine mode aerosol is ubiquitously acidic. During winter with low temperature and high relative humidity, aerosol $pH_F$ is higher than during summer following the liquid water availability and temperature (Fig. 5a). This seasonal trend has been widely observed in the eastern U.S. (Guo et al., 2016; Guo et al., 2015),

Beijing (Tan et al., 2018), Inner Mongolia (Wang et al., 2019a), Hong Kong (Xue et al., 2011), Po Valley, Italy (Squizzato et al., 2013), Cabauw, Netherlands (Guo et al., 2018b), and eastern Canada (Tao and Murphy, 2019b) with $pH_F$ differences between seasons spanning from 0.6 to 2.3 $pH_F$ units. Wang et al. (2019b) reported the lowest mean aerosol $pH_F$ in summer and attributed it to the higher contribution of secondary sulfate than in the other seasons and the highest mean aerosol $pH_F$ in spring likely associated with the influence of dust. The most complete dataset containing seasonality comes from Canada, where

observationally-derived monthly mean pH values for $PM_{2.5}$ were constructed for 6 sites over 10 years (Tao and Murphy, 2019b). The Canadian dataset shows summertime minimum pH and wintertime maximum with 1 pH unit of difference (~2 versus ~3, respectively). Aerosol acidity increases with increasing temperature (0.1 unit increase in pH per 2 K decrease in temperature) and decreasing relative humidity. Summer pH is largely dictated by temperature while both meteorological factors and aerosol composition affect winter pH. Beijing shows a similar $pH_F$ trend with winter having higher $pH_F$ than

summer ($pH_F$ of 4.1 vs 1.8) (Tan et al., 2018). However, Beijing data also shows summer 2016–2017 $pH_F$ (Ding et al., 2019) being almost 2 units higher than that in summer 2014 (Tan et al., 2018), potentially indicating effective air pollution mitigation strategies. The summer minimum in fine aerosol pH is a common feature of all available pH datasets and is associated with the effects of high ambient temperatures and low aerosol water content. Temperature has also been shown to strongly affect the partitioning of total ammonium through its effects on solubility and dissociation (Hennigan et al., 2015). Composition can

also play a role in seasonality as shown in data for Inner Mongolia (Wang et al., 2019a) and at Po Valley (Squizzato et al., 2013) where maximum fine aerosol $pH_F$ was found in spring likely due to the influence of desert dust aerosol from Gobi and from Sahara deserts respectively during those time periods. The absolute values in Mongolia show partial neutralization of the aerosol with $pH_F$ between 5 and 6.1, while those for Po Valley are more acidic, being 1.3 $pH_F$ units higher in spring ($pH_F$ = 3.6) than in the summer ($pH_F$ = 2.3).

Similarly, resulting from diurnal changes in temperature and relative humidity, higher $pH_F$ is observed during the night compared to that during the day (Fig. 5b). For example, acidity shows diurnal variation in China of almost 2 $pH_F$ units (Cheng et al., 2015), in southern Canada of 0.5 – 5 pH units (Murphy et al., 2017), in the U.S. of 0.65 – 1.5 $pH_F$ units (Battaglia et al.,



2017; Guo et al., 2015; Nah et al., 2018), and about 1 $pH_F$ unit in southern California (Guo et al., 2017b). Finokalia experiences ~1 $pH_F$ unit lower $pH_F$ during the day than night because of low aerosol water content and high temperatures (Bougiatioti et al., 2016). This pattern is amplified by the urban Heat Island Effect through its impact on temperature (Battaglia et al., 2017).

Figure 6 summarizes the current estimates of ambient fine aerosol pH based on literature data summarized in Table S6. Studies that used only aerosol composition for calculating pH ("reverse mode" aerosol calculations, which are uncertain, e.g., see Hennigan et al., 2015) or ion-balance based approaches are excluded from the figure. Mean fine aerosol $pH_F$ ranges from around 1 to 6 although specific locations and episodes may experience higher or lower acidity. Highly acidic fine aerosols are found in South East Asia, the eastern U.S., and other locations. Mainland China, Europe, Canada, Mexico, and the western

U.S. have on average similar levels of aerosol acidity (2.5 to 3). This spatial variability in pH reflects variability in the chemical composition of fine aerosols that result from the combined effect of changes in sources and meteorology.

Overall, the eastern U.S. aerosol is predicted to be one of the most acidic locations, with average $pH_F$ near a value of 1 (Battaglia et al., 2017; Craig et al., 2018; Fang et al., 2017; Weber et al., 206; Pye et al. 2018; Xu et al., 2015; Guo et al., 2016; Guo et

al., 2015), and higher $pH_F$, by about 1 unit, observed in locations of intensive agriculture with high $NH_3$ concentrations (Nah et al., 2018) and those influenced by larger particles (Fang et al., 2017; Craig et al., 2018). Higher aerosol pH ($2 - 3$ in $pH_F$) was estimated for Los Angeles in summer (Guo et al., 2017b), similar to observationally derived values for the Eastern Mediterranean ($0.5 - 2.8$ $pH_F$, Bougiatioti et al., 2016). The $pH_F$ of $PM_1$ and $PM_{2.5}$ in Pasadena during the CalNex-2010 campaign were slightly different with the larger $PM_{2.5}$ particles having $pH_F$ 0.8 units higher than $PM_1$ (Guo et al., 2017b) due

to the larger water content and more abundant NVC at larger sizes. Greater acidity in submicron ($PM_1$) versus larger ($PM_{2.5}$) fine-mode particles is a robust feature in multiple data sets (e.g., Bougiatioti et al., 2016; Fang et al., 2017; Fridlind and Jacobson, 2000; Ding et al., 2019; see Sect. 7.1 for a discussion of pH as a function of particle size). Guo et al. (2016) estimated the mean $pH_F$ at $0.77 \pm 0.96$ for $PM_1$ aerosol aloft based on aerosol chemical composition measurements during the Wintertime Investigation of Transport, Emissions, and Reactivity (WINTER) campaign in the northeastern U.S. and thermodynamic

modeling.

For mainland China, fine aerosol $pH_F$ estimates vary, but tend to be mildly acidic (average $pH_F$ approximately 4) and span from negative values (in Chengdu) to as high as 6.1 for $PM_{2.5}$ aerosol (Liu et al., 2017; Jia et al., 2018; Song et al. 2018; Tian et al., 2018; Cheng et al., 2015; He et al., 2018; Tan et al., 2018; Shi et al., 2017; Guo et al., 2017a; Ding et al., 2019; Jia et al.,

2018b; Wang et al., 2019), while in Southeast Asia (Singapore and Hong Kong) fine aerosol is highly acidic (average $pH_F$ approximately 1) (Behera et al., 2013; Yao et al., 2007). Ding et al. (2019) estimated coarse particles were generally neutral or alkaline, based on observations in Beijing and modeling with ISORROPIA II. The strong acidity in Southeast Asia is consistent with reported high solubility of particulate iron sampled in the South China sea (Li et al., 2017). Altogether, these



data suggest that there is a large spatial gradient of pH across China and Southeast Asia, reflecting the highly variable sources of acidity and alkalinity within each region (Shi et al., 2019).

Higher aerosol $pH_F$ of 4.60 and 4.75 has been inferred from observations for Hawaii (Pszenny et al., 2004) and for Sao Paulo (Vieira-Filho et al., 2016). In the case of Hawaii, the higher $pH_F$ is due to the neutralizing effect of non-volatile cations from sea salt. Sao Paulo is affected by combustion sources and thus emissions of nitrate and NVC that increase the aerosol $pH_F$. Bougiatioti et al. (2016) reported that under the influence of biomass burning, aerosol $pH_F$ increases to values around 3, indicative of the impact of non-volatile cations, particularly potassium, as well as ammonia and nitrate emitted from wood burning. In general, aerosol $pH_F$ increases when gas-phase $NH_3$ increases and can further be elevated by co-condensation of nitrate and water and the presence of non-volatile ions (Guo et al., 2018a; Shi et al., 2019).

Information on pH trends over time is limited, due to the scarcity of relevant data. pH data for mainland China published prior to 2010 (Tables S6, S7) showed highly acidic aerosol with $pH_F$ of -0.16 ± 0.75 in contrast to more recent estimates (mean $pH_F$ of 3.42±1.75 for 2011 – 2016). Pre-2010 estimates are subject to large uncertainty resulting from their calculation method of relying only on aerosol composition information as input. Thermodynamic analysis of data, when carried out in a way that minimizes pH biases (mostly focused on using total gas and particle composition inputs and higher RH conditions) suggests acidity trends that may, at first glance, seem counterintuitive. One of the few such examples published that provides important insights can be found for the eastern U.S. during summer. Emissions controls over the last 20 years lead to significant reduction in sulfate aerosol, and ammonia levels remained constant or even slightly increased. Despite these important changes, summertime aerosol acidity remained the same (Weber et al., 2016). The insensitivity of aerosol pH to changes in emissions controls in this region is largely driven by the semi-volatility of ammonium, which requires a fraction of it to remain in the gas phase as dictated by thermodynamic equilibrium. Similar behaviour was found by Tao and Murphy (2019b) in Canada, where summertime aerosol pH did not increase over 10 years despite substantial decreases of sulfate and constant levels of ammonia. In the same study, the seasonality and interannual variability of $pH_F$ was found to be strongly driven by the temperature changes and the resulting shift in thermodynamic partitioning and water uptake. The modeling study of Lawal et al. (2018) also showed little response of aerosol $pH_F$ throughout the Continental U.S. to emission reductions despite the considerable improvements in air quality over the period 2001-2011. For this, they used a thermodynamic (ISORROPIA II) and the chemical transport (CMAQ) model together with the aerosol pH relevant observations from three monitoring networks (AMoN, SEARCH, and CASTNET). However, if sulfate aerosol continues to decrease, aerosol pH may eventually begin increasing (as proposed by Tao and Murphy, 2019b). In the southeastern US, when sulfate approaches the 0.2 – 0.3 µg m$^{-3}$, level, small amounts of NVCs start affecting pH causing it to increase (Weber et al., 2016). Weber et al. (2016) also calculated that only large increases in $NH_3$ together with sulfate reductions can lead to an increase in pH. Thus, for sulfate between 0.1 and 10 µg m$^{-3}$, pH approaches 2.5 when $NH_3$ is over 10 µg m$^{-3}$ and ammonium nitrate is formed. In any location, unusually high levels of $NH_3$ (an order of magnitude above the background or higher) associated with localized emissions, (e.g., confined animal feeding operations)



can also increase pH (Nah et al., 2018). For locations characterized by high levels of ammonia, strong emissions of nitrate, and/or high levels of NVCs, pH may be driven by the water uptake and the mild acidity associated with ammonium nitrate aerosol. Furthermore, meteorology (RH, T) is an important driver of pH and meteorological trends influenced by climate change or interannual variability, can dominate over any composition changes (Tao and Murphy, 2019b). The eventual

response of aerosol pH to changing emissions and meteorology can be determined with models – but careful evaluation of them with in-situ data is critical to ensure that they are in the correct acidity regime (e.g., Vasilakos et al., 2018; Shah et al., 2018).

## 5.2    Observed cloud and fog acidity

Sample collection is usually the largest challenge associated with measuring cloud droplet pH. Once obtained, the pH of collected cloud and fog water is typically measured using an electronic pH meter and a combination glass electrode. The approach to pH measurement in cloud and fog water has been similar over the past several decades. Semi-micro or micro-electrodes are available to analyze small volumes of available fog/cloud water, with some pH microelectrodes capable of measuring as little as 10 µL of sample. The electrodes are typically calibrated using pH 4 (phthalate-based) and 7 (phosphate-

based) buffer solutions, although higher and lower pH calibration buffers are also available (see also supplementary information Sect. S1). Buck et al. (2002) provide an overview of key buffer requirements (stability, ionic strength, certification, low pH change with temperature) and a list of primary buffer standard compositions.

### 5.2.1    What determines the pH in a cloud/fog droplet?

The pH of a fog or cloud drop is determined by the balance between acids and bases in solution. The initial composition of a droplet is determined by the dissolution of soluble material contained within an aerosol particle that serves as the CCN. Further changes to composition come from subsequent scavenging of other, non-activated, interstitial particles and from uptake of water-soluble gases and aqueous-phase reactions (Sect. 3). While early measurements of cloud and fog composition focused on inorganic species, it has become increasingly clear that organic matter also contributes significantly to droplet composition

(Herckes et al., 2013) and, potentially, to droplet pH.

Uptake of gaseous carbon dioxide is an important factor governing cloud pH, especially in remote environments. Equilibration of a pure water drop with current levels of atmospheric $CO_2$ at 298 K results in a droplet pH of approximately 5.6, a value often referred to as the pH of natural rain or cloud water. Cloud pH values above or below this value are often referred to as

alkaline or acidic, respectively.





Sulfuric and nitric acids frequently make significant contributions to cloud/fog drop pH while ammonia is typically the most abundant base. Sulfuric acid is taken up through particle scavenging, including scavenging of ammoniated sulfate particles, and is also formed *in situ* through aqueous phase oxidation of sulfur dioxide. Nitric acid is a highly soluble gas, in part because of its strong acidity which leads to nearly complete deprotonation in a cloud drop to form nitrate. Addition of nitrate to cloud

water also comes from scavenging of particles containing solids or dissolved nitrate salts. These include ammonium nitrate, but also calcium or sodium nitrate, which are frequently formed by reaction of nitric acid or its precursors with sea salt or soil dust particles (e.g., ten Brink, 1998; Lee et al., 2008). Cloud water ammonium is derived by uptake of gaseous ammonia, as well as from particles containing salts of ammonium with nitrate, sulfate, and organic acids. As a result of these various scavenging and oxidation pathways, it is common for cloud/fog composition to be dominated by concentrations of sulfate,

nitrate, and ammonium (e.g., Weathers et al., 1988; Collett et al., 2002).

A variety of weak organic acids and bases, including carboxylic/dicarboxylic acids and amines, can also influence the pH of fog or cloud drops. Carboxylic and dicarboxylic acids are frequently reported (e.g., Kawamura and Kaplan, 1984; Weathers et al., 1988; Munger et al., 1989; Facchini et al., 1992; Collett et al., 1999; van Pinxteren et al., 2005; Boris et al., 2016) as

contributors to cloud or fog acidity, even in remote environments, due to the abundance of these compounds, including formic acid, acetic acid, pyruvic acid, succinic acid, and oxalic acid, in the atmosphere. For these weak acids and bases, the extent of partitioning from the gas phase is a sensitive function of droplet pH. Partitioning of weak carboxylic acids, such as formic and acetic acids, into the aqueous phase is strongly favored at pH values above the acid's pKa value, due to the deprotonation of the acid in such high-pH solutions.

**5.2.2    Recent observations of cloud/fog pH and long-term trends**

Clouds and fogs have been observed to exhibit a wide range of pH values (Table S8). Typical values fall between pH 3 and 6. The most acidic observation reported was in an evaporating fog in Corona Del Mar in coastal Southern California, with a pH of 1.69 (Hileman, 1983). Other highly acidic pH values include 1.95 for a fog at Mt. Oyama in Japan (Mori et al., 1997), 1.94 for a fog in Duebendorf, Switzerland (Sigg et al., 1987), and 1.7 for a fog in Kahler Asten, Germany (Kroll and Winkler, 1988).

Such acidic values are typically associated with large inputs of sulfuric and nitric acids, although hydrochloric acid has also been an important source of acidity in some urban areas (e.g., the Duebendorf fog). High pH fogs or clouds have also been reported in situations with large inputs of ammonia or alkaline soil dust. For example, Collett et al. (1999) reported pH values up to 7.43 for radiation fogs in California's Central Valley, a region with high ammonia concentrations stemming from major agricultural activities. Wang et al. (2011) collected cloud water at Mt. Tai, China and observed cloud pH values during periods

of strong soil dust influence in the range of 6.5 – 6.7. Changing regional transport patterns and resulting variations in inputs of acids and bases to Mt. Tai clouds, however, resulted in a wide range of values between 2.56 and 7.64 overall at this site. pH as high as 7.76, in $Ca^{2+}$ rich advection fogs, has been observed at a roadside location near Sao Paulo, Brazil (Vasconcellos et al., 2018). Fog pH values above 7 have also been reported in polluted fogs in Kanpur, India (Kaul et al., 2011; maximum pH





8.07), in Ca$^{2+}$-rich fogs in Xishuangbanna, China (Zhu et al., 2000; maximum pH 9.15), in marine-influenced clouds at Puy de Dome, France (Deguillaume et a., 2013; maximum pH 7.6), and in other locations (see Table S8).

Figure 7 (and Table S8) depicts pH observations from locations around the globe including observations at continental and marine locations for fogs and clouds collected by airborne and ground-based sampling platforms. Panels represent different time periods, from pre-1985 to post 2005, to highlight how strong regional changes in anthropogenic emissions, especially sulfur and nitrogen oxides, are incorporated in clouds thus affecting pH. Measurements pre-1985 are mostly associated with studies in the United States, Europe, Japan, and Australia. More global interest and coverage was seen in the later 1980s and 90s, with several measurement sites active in east Asia, Africa, and South America. Since 2005, there has been continued interest in cloud and fog observations in some regions, including in China and India, two countries facing increasingly severe air quality challenges. Globally, observed pH values range from highly acidic to more alkaline. While measurement locations are not constant over time, there appears to be a decrease in incidence of more acid clouds and fogs in North America and Europe since the 1980s and early 1990s, while trends in the incidence of acid fogs and clouds in East Asia are less clear.

While the global scientific community lacks long-term monitoring programs for cloud/fog composition, there are a few locations around the world where such measurements have been made routinely, or at least periodically, over periods of a decade or more. Figure 8 shows temporal trends in cloud/fog pH from a number of sites in the United States and Japan. Fog pH values from radiation fogs in California's Central Valley show a significant increase from the 1980s to the current decade. Herckes et al. (2015) attributed the rapid pH rise in the early part of the record, particularly at sites in the southern part of the valley, to decreases in SO$_2$ emissions. A steady climb is also apparent in cloud pH values measured at Whiteface Mountain, located in upstate New York in the northeastern U.S., consistent with reductions in regional NO$_x$ and SO$_2$ emissions. Schwab et al. (2016) previously reported decreases in cloud water SO$_4^{2-}$, NO$_3^-$, NH$_4^+$, and H$^+$ concentrations at Whiteface Mountain of 3.8%, 3.7%, 2.8%, and 4.3% per year, respectively, over the period 1994-2013. Over this twenty-year period the cloud pH increased approximately 0.4 pH units per decade. Cloud pH values have been measured at remote locations in the Luquillo Mountains on the Caribbean island of Puerto Rico since 1967. Mean values reported in several studies up through 2012 (Lazrus et al., 1970; Weathers et al., 1988; Asbury et al., 1994; Gioda et al., 2009; Gioda et al., 2011; Gioda et al., 2013; Reyes-Rodríguez et al., 2009; Valle-Díaz et al., 2016) fall between 4.6 and 5.8, with no apparent trend. The pH values observed and the lack of a clear trend here are consistent with the fairly clean conditions in the region. Long-term records of cloud or fog composition are even rarer in Asia. Between the 1960s and 1990s in central Japan (Fig. 8), conditions are fairly acidic, with mean pH values mostly between 3 and 4. Measurements after 2000 suggest a possible increase in cloud pH in the region. Long-term (unpublished) cloud pH measurements exist for a few locations in Japan and Taiwan; future anticipated publication may shed more light on acidity trends in the region.



Figure 9 examines changes in cloud and fog pH measured from 1980 to present in Europe. By combining data from multiple locations, a more complete assessment of pH trends on the continent is possible. While there is considerable variability within individual record years and between years, the data overall suggest a trend toward increased pH, with values at the present time typically about one pH unit higher than in 1980. This increase, also seen in the U.S., is consistent with decreasing

European emissions of key acid precursors: $SO_2$ and $NO_x$.

Given the obvious connections between clouds and precipitation, similarities in the temporal trends of cloud pH with pH trends reported in precipitation are worth noting. Vet et al. (2014) analyze a large set of precipitation measurements from around the world, considering precipitation composition and its temporal changes by region, with the analyses focused especially on the

period 2000 – 2007. They report that changes in $SO_2$ and $NO_x$ emissions in many regions of the globe result in measurable changes in sulfate and nitrate wet deposition that in turn produce changes in pH and $H^+$ wet deposition. Vet et al. (2014) note that 75% of European sites and 85% of North American sites saw increases in precipitation pH over this time period. A more spatially heterogeneous pattern of changing precipitation pH is reported for Asia. Looking at a longer time period, Duan et al. (2016) report a decreasing average precipitation pH in China from 1999 – 2006 with pH increasing after 2006, a pattern that

is the inverse of temporal trends in China's $SO_2$ emissions.

### 5.2.3    pH variation across drops within a cloud/fog

Up to this point, a single pH value for a cloud or fog has been discussed. In reality, each droplet within a fog or cloud is likely to have a unique composition. Within a cloud, droplets have a range of sizes as they grow, following initial cloud drop activation on CCN, by condensation (water vapor depositing on cloud drops) and coalescence (droplets typically of different

sizes impacting and forming a single, larger drop). Cloud drops form when a critical supersaturation, associated with the critical (dry) diameter of the particle, is met following Köhler theory. Köhler theory indicates that larger CCNs activate at lower supersaturations and are therefore the first CCNs to be dissolved in droplets. A simple model representing the initial stages of condensational growth (Twohy et al., 1989) found that larger cloud drops activate on larger CCNs. Larger particles are typically mechanically generated and oftentimes are comprised of more alkaline components, including soil dust and sea salt. Smaller

particles, typically made up of sulfate, nitrate, ammonium, and organic species, tend to be more acidic (Hoag et al., 1999). Given that the composition of the CCNs varies with size, then the fog/cloud solute composition will vary with drop size as observed in a variety of clouds and fogs (Noone et al., 1988; Ogren et al., 1989; Munger, 1989; Bator and Collett, 1997; Laj et al., 1998; van Pinxteren et al., 2016; Moore et al., 2004; Guo et al., 2012a; Herckes et al., 2013).

Other cloud physical processes also affect the solute composition as a function of cloud drop size. Because of the larger surface area to volume ratio in smaller drops compared to larger cloud drops, water vapor condensation will favor more rapid growth (per unit volume) of small drops, quickly diluting their solute concentrations. Furthermore, droplets formed from smaller hygroscopic particles are much more diluted at the point of cloud droplet formation (CCN activation), compared to coarse

mode particles (e.g., Nenes and Seinfeld, 2003). Coalescence tends to occur among larger and intermediate cloud drops of sufficiently different sizes because of their different fall speeds. Thus, coalescence tends to mix the composition of the larger drops leaving the composition of very small drops less affected by this process. The same is true for mixed phase, ice-water, clouds, where falling ice crystals capture large cloud drops most effectively. Once drops or ice crystals are large enough to fall

out of the cloud (i.e. precipitation as rain or snow), removal from the atmosphere (by wet deposition) of larger drops and their dissolved solutes and trace gases will occur.

The rate of mass transfer of other trace components between the gas and aqueous (cloud/fog droplet) phases also depends upon the size of the droplet. The kinetic mass transfer coefficient often used to describe the mass transfer between the gas and

aqueous phases in cloud chemistry models incorporates a representation of gas phase diffusion and interfacial mass transfer limitations and illustrates the dependence of mass transfer on droplet size, with the overall transfer rate related to the inverse of the droplet radius (or inverse of the square of the radius) (Schwartz, 1986). This size dependence of the mass transfer coefficient can contribute to mass transfer occurring to and from droplets at variable rates across the droplet size spectrum, sometimes leading to slower aqueous concentration increases in large droplets from the uptake of soluble gas phase species

compared to smaller droplets (Ervens et al., 2003).

Measurements of pH in cloud water samples collected by size fractionating cloud water collectors (Collett et al., 1994) revealed that pH was lower in small drops compared to large drops for clouds and fogs sampled at various locations in the United States. The variation of pH across the droplet size spectrum has important implications for aqueous-phase chemistry, especially for

S(IV) oxidation to form sulfate (Seidl, 1989; Hegg and Larson, 1990; Pandis et al., 1990; Lin and Chameides, 1991; Roelofs, 1993; Fahey et al., 2005; Gurciullo and Pandis, 1997; Reilly et al., 2001; Tilgner et al., 2013; Hu et al., 2019; Rao and Collett, 1998), which increases the acidity of the drops. Bulk cloud $pH_F$ calculations (i.e. average characteristics, a common treatment in chemical transport models, Sect. 8.2) tend to underestimate the fraction of dissolved S(IV) in the form of $SO_3^{2-}$ (where S(IV) = $SO_2 \cdot H_2O$ + $HSO_3^-$ + $SO_3^{2-}$), which causes the underestimation of sulfate production rates (Fahey and Pandis, 2003; Hegg et

al., 1992; Hoag et al., 1999; Moore et al., 2004; Roelofs, 1993). This is particularly true for S(IV) oxidation by ozone ($O_3$), since the rate constant for $SO_3^{2-}$ + $O_3$ is several orders of magnitude larger than for $HSO_3^-$ + $O_3$. Barth (2006) found that pH variation across cloud drop sizes is also important for aqueous-phase formaldehyde oxidation forming formic acid, which, as a weak acid, can reduce the pH of the drops. Furthermore, Tilgner et al. (2013) demonstrate that a size-resolved multiphase chemistry treatment results in higher acidity production in smaller droplets leading to more acidic, smaller CCN particles after

cloud processing while larger particles tend to be less acidic.

Both the composition and pH variation across the droplet population lead to differences in reactivity for different size droplets. Not only does the composition of droplets vary across the size spectrum in terms of reactants/oxidants, but many reactions and effective Henry's Law coefficients are pH dependent (Sect. 3). Additionally, droplets of different size settle and deposit at

different rates. In fogs, where the net effect of processing can be a cleansing of the atmosphere, larger droplets deposit faster than smaller ones, so those species enriched in larger droplets will be removed from the atmosphere faster than those species enriched in smaller droplets (Collett et al., 2001; Collett et al., 2008; Fahey et al., 2005). Nevertheless, most chemistry transport models still use a bulk water composition (i.e. average characteristics) to compute aqueous phase chemistry. Parameterizations

informed by how bulk and size-resolved pH differ can be employed to better represent aqueous-phase oxidation within clouds (see Sect. 8.2).

### 5.2.4    Need for future monitoring of cloud pH

Although cloud and fog sampling is generally more challenging than aerosol collection, pH measurement of the collected cloud/fog water is simpler due to its much larger volume and much lower ionic strength. Over the past several decades, fogs

and clouds have been sampled and their pH determined in areas around the globe. Depending on inputs of key acids and bases, cloud/fog pH has been observed to range from below 2 to greater than 7. Programs designed to target reductions in acid rain have had direct impacts on cloud and fog pH. Analysis of pH observations over the past 25 – 30 years reveals that cloud/fog acidity in many regions has decreased as anthropogenic emissions of the important acid precursors, $SO_2$ and $NO_x$, have decreased. A continued rise in cloud/fog pH is likely in many regions with planned, future decreases in $NO_x$ and $SO_2$ emissions

and stable or increasing $NH_3$ emissions. Increases in cloud pH are expected to enhance the solubility of gas phase organic acids, potentially shortening their atmospheric lifetimes.

Much remains to be learned about factors controlling cloud/fog pH in the atmosphere and the influence of this acidity on aqueous phase chemistry, including the aqueous phase uptake and oxidation of soluble gases to form secondary inorganic or

organic aerosol. More detailed measurements of organic acids and bases, and their influence on cloud pH, will be increasingly important as sulfate and nitrate concentrations decline. Likewise, there is a need for more systematic monitoring of cloud and fog composition in key environments, as opposed to the more *ad hoc* past sampling approaches driven primarily by the objectives of process-based research. Because fogs and clouds are good integrators of atmospheric acids and bases in both the gas and particle phases, they may offer a convenient and practical basis for ongoing monitoring of atmospheric acidity. Future

monitoring strategies should consider long-term monitoring at surface sites as well as periodic measurements of cloud composition from aircraft in order to enhance our understanding of acidity at higher elevations in the troposphere. Future measurements should also better document heterogeneity of acidity across individual drops within a cloud or fog, for example looking at the size-dependence of drop pH.

### 6    Box-model guidance for the use of approximations and proxies of acidity for fine particles

This section applies the concepts introduced in previous sections regarding the definition of pH (Sect. 2.1), approximations of pH (Sect. 2.3), and proxies of acidity (Sect. 4). Specifically, E-AIM, AIOMFAC-GLE, MOSAIC, ISORROPIA II, and



EQUISOLV II are used to carry out an intercomparison of pH predictions, approximations, and/or proxies using idealized and ambient fine particle compositions. Observations of gas–liquid equilibrium of semi-volatile inorganic compounds were obtained from published studies from North America, Europe, and China representing what can be found in typical regional and global model studies.

## 6.1    Idealized scenarios

### 6.1.1    Description of systems

In this section, well-constrained acidity calculations were carried out by the models described in Sect. 2.6. The test cases involve the prediction of gas–liquid equilibrium of water and semi-volatile inorganic compounds as well as pH for a range of equilibrium RH. Three aerosol test systems are compared: (1) an ammonium- and sulfate-rich system, (2) a NaCl-rich, sea-salt-like aerosol system, and (3) a nitrate- and ammonium-rich, but relatively sulfate-poor system. For each system, moderately acidic and highly acidic conditions were investigated, while covering seven RH levels: 99 %, 90 %, 80 %, 70 %, 60 %, 50 %, and 40 %. All calculations were for a temperature of 25 °C. Molar input concentrations were chosen to represent realistic atmospheric conditions, for example using gas phase ammonia concentrations of $1.2 - 25$ ppb$_\text{v}$ typical for suburban to polluted air (Wang et al., 2015) and sulfate amounts resulting in ~ $3 - 8$ µg m$^{-3}$ inorganic aerosol mass concentration in the highly acidic cases. The input concentrations and conditions for the systems are summarized in Table 3. These input concentrations describe initial (non-equilibrium) total (gas + liquid) molar amounts per unit volume of air – except for water, which is constrained by the given RH. The thermodynamic models equilibrate the different dissolved species and volatile inorganic gases, including solving for the equilibrium dissociation degree of bisulfate (HSO$_4^-$) in the liquid aerosol phase, the ammonia–ammonium equilibrium, and the aerosol water content. Mean molal activity coefficients for (H$^+$, HSO$_4^-$) or (H$^+$, Cl$^-$) cation–anion pairs are used in sulfate-rich and sulfate-poor systems, respectively, to estimate pH using pH$_\pm$(H, X) (Eq. 7). The calculated pH for all systems are summarised in Tables S3 – S5.

*System 1: Water + (NH$_4$)$_2$SO$_4$ + H$_2$SO$_4$ + NH$_3$*

The first test system is an acidic aqueous ammonium + sulfate / bisulfate system. Input concentrations of the electrolytes include (NH$_4$)$_2$SO$_4$ (99.9 % by mass or 50 % by mass for moderately and highly acidic cases respectively) and H$_2$SO$_4$ with a separate gas phase input of NH$_3$ (mol m$^{-3}$ air), all of which are then subject to change within a thermodynamic equilibrium calculation. No solid–liquid equilibria were considered. The highest pH values predicted are ~ 4 for the slightly acidic case at 99 % RH, while the lowest pH values of ~ 0.53 to 0.88 were predicted for the highly acidic case at 40 % RH.

*System 2: Water + Na$_2$SO$_4$ + NaCl + H$_2$SO$_4$ + HCl*


The second system represents an acidified sea-salt-like aerosol solution, in which $Mg^{2+}$ was substituted by charge-equivalent amounts of $Na^+$. A highly acidic and a moderately acidic variant were created by specifying different amounts of sulfuric acid. The input for this system includes HCl, some of which will exist in the gas phase.

*System 3: Water + $(NH_4)_2SO_4$ + $H_2SO_4$ + $NH_3$ + $HNO_3$*

The third system represents an acidic, nitrate-rich and comparably sulfate-poor aerosol ($X_T > 2$ in Eq. 9). It involves the gas–liquid equilibration of the inorganic base $NH_3$ and the acid $HNO_3$, critical for establishing the equilibrium pH in the system. In the moderately acidic case, the highest pH values are ~ 2.5 at 99 % RH, while the lowest pH values were predicted for the highly acidic case at 40 % RH, with the pH ranging between ~ 0.91 and 1.60 depending on the model.

### 6.1.2    Comparison of pH predictions and approximate measures of pH

Results for systems 1, 2, and 3 are shown in Fig. 10 panels (a), (b), and (c) for moderately acidic and (d), (e), and (f) for highly acidic scenarios. The pH values predicted by those models accounting for the single-ion activity coefficient of $H^+$ (E-AIM and AIOMFAC-GLE, solid symbols in Fig. 10) differ only slightly from each other. For example, the E-AIM and AIOMFAC–

GLE calculations for system 1 yield pH differences of 0.03 to 0.2 pH units for RH between 99 % and 80 %, while differences in magnitude of 0.21 to 0.35 pH units result for RH between 70 % and 40 %. Differences are expected to be smaller at high RH, where high water contents result in relatively high dilution and model–model differences in activity coefficients become smaller (see Sect. 2.5 for challenges at high ionic strength). However, this system illustrates that even at 99 % RH the $H^+$ activity coefficients deviate from a value of one and are as small as 0.4. The pH predictions solely based on free $H^+$ ($pH_F$, Eq.

6) frequently deviate from pH by 1 unit for the idealized scenarios considered here (Fig. 11). Predictions based on total $H^+$ ($pH_T$, Eq. 8) can differ from pH by up to 2 units.

MOSAIC, EQUISOLV II, and ISORROPIA II use mean molal ion activities when computing the dissociation of bisulfate, the gas–liquid equilibrium of ammonia, and other equilibria (single-ion activity coefficients are not computed by these models).

Therefore, pH predictions with such models require an approximation, for example the application of the mean molal ion activity approach of Eq. (7) for $pH_\pm$. Although not a perfect approximation, $pH_\pm$ predictions can be very close to those carried out with the single-ion activity coefficient consideration. For example, for the moderately acidic case in system 1 (Fig. 10a, MOSAIC predictions differ from those by E-AIM by about 0.03 pH units at 99 % RH (0.07 pH units with respect to AIOMFAC–GLE), 0.3 pH units at 80 % RH and 0.46 pH units at 40 % RH (0.75 pH units with respect to AIOMFAC–GLE).

For system 1, the MOSAIC $pH_\pm$ value is generally lower than the pH from E-AIM and AIOMFAC–GLE. In the highly acidic system 1 case, the pH difference between MOSAIC and E-AIM is within $0.03 - 0.21$ units, except for a 0.33 pH unit difference at 40 % RH. The ISORROPIA II model shows the largest variation in predicted pH over the 99 % to 40 % RH range, especially for the highly acidic case (Fig. 10d). For reasons of enhanced computational efficiency, ISORROPIA II uses look-up tables to



determine the water content at a specified RH for a given aerosol system and is run with a higher tolerance level for numerical convergence than, for example, AIOMFAC-GLE. These efficiency adjustments may contribute to a notable difference in predicted water content and resulting $pH_\pm$, particularly at 99 % RH, compared to the predictions with more rigorous equilibrium solvers used by the other models.

Generally, the observed differences in pH predicted by the thermodynamic models occur due to a combination of reasons. These include: (1) differing predicted liquid water content at given equilibrium RH (water activity); (2) the predicted degree of bisulfate dissociation, which depends on the aqueous phase composition and the values of the predicted activity coefficients (for $H^+$, $SO_4^{2-}$, and $HSO_4^-$) involved in the equilibrium; (3) the gas–liquid partitioning of $NH_3$ (or other volatile components

for systems 2 & 3), and (4) the use of single-ion vs. mean molal ion activity coefficients in the calculation or approximation of pH. Reasons (1) – (3) affect each other directly, such that any inherent difference among the model equations, for example the temperature and ionic strength dependence of water and ion activity coefficients, will lead to a different equilibrium solution for the aqueous phase composition and pH. The interplay among composition-dependent activity coefficients and the gas–liquid or ion dissociation equilibria are non-linear and may amplify or dampen effects on predicted pH in a complex

manner. Therefore, given the type of test computations with gas–liquid equilibria considered here, differences among models on the order of 0.05 – 0.2 pH units (or even larger at very high ionic strengths resulting at moderate to low RH) are to be expected. Within this range, pinpointing which of the models is closest to the truth is not possible, but in general, pH from models that calculate single-ion activity coefficients (and hence $a_{H+}$) using a rigorous numerical approach are to be preferred over those that assume a unity $H^+$ activity coefficient or those that assume a mean activity coefficient. Figure 10 indicates that

the disagreement between model predictions typically increases with decreasing water activity (RH) for both moderately and highly acidic conditions.

For system 2 (sea-salt like), all models predict the highest acidities (lowest pH) at high RH, with the pH values increasing with decreasing RH for both the slightly and highly acidic calculation variants (Fig. 10 b,e). This is because most of the HCl in the

system is present in the gas phase, and this amount remains relatively constant over the whole RH range. Chloride ion activity in the aqueous phase rises as RH decreases and the aqueous solution becomes more concentrated. As a result, $H^+$ activity decreases with decreasing RH to compensate and so maintain equilibrium with the roughly constant partial pressure of HCl(g). This rise in pH may be unrealistic compared to typical ambient conditions due to the high HCl and absence of ammonia in this test case. The pH predictions by E-AIM model III and AIOMFAC–GLE agree well (absolute differences of 0.01 to 0.06 pH

units). The MOSAIC and ISORROPIA II predictions of pH in system 2 were carried out using the ions ($H^+$, $Cl^-$) for $pH_\pm$. For the moderately acidic conditions, the MOSAIC and ISORROPIA-derived $pH_\pm$ are in good agreement with E-AIM only at 99 % RH (0.02 units difference), while larger deviations of 0.15 to 0.75 pH units occur for 90 % to 40 % RH. In this moderately acidic case, MOSAIC, ISORROPIA II, and EQUISOLV II tend to systematically overpredict the pH value towards lower RH relative to the other models. In the highly acidic case, the MOSAIC–E-AIM deviation is between 0.07 and 0.43 pH units at



RH < 99 %. Such deviations are linked to large variations in the molality-based H$^+$ activity coefficients ranging from ~ 0.72 at 99 % RH to > 30 at 40 % RH (see AIOMFAC–GLE values in Table S4), which lead to larger errors when mean molal activity coefficients are used to obtain pH$_\pm$. This important influence of the H$^+$ activity coefficient or its approximation via $\gamma_\pm$(H$^+$, Cl$^-$) is exemplified by comparison of the ISORROPIA II predictions with E-AIM and the other models. The very high

activity coefficients of H$^+$ and Cl$^-$ predicted by E-AIM, which get larger as RH decreases, result in only very low molalities of H$^+$ left in the aqueous aerosol. ISORROPIA II yields a mean activity coefficient of (H$^+$, Cl$^-$) that is very low compared to that of the other models, and varies little with RH, which means that the predicted HCl concentration in the aerosol is substantially higher. This results in the lower, and rather invariant, predicted pH$_\pm$ by ISORROPIA II for RH < 80 %. This example further indicates that assuming activity coefficients of unity in the computation of pH$_F$ based on free H$^+$ molality (or for pH$_T$) can lead

to errors in this approximation of actual pH values in concentrated solutions (see also Sect. 6.2).

The pH predictions by E-AIM model III and AIOMFAC–GLE agree relatively well for system 3, which contains mainly ammonium nitrate (Fig. 10 c,f), especially in the moderately acidic case. There, pH differences are 0.02 units at 99 % RH and about 0.10 – 0.12 units between 90 % and 40 % RH, with AIOMFAC–GLE predicting the slightly lower pH. In the highly

acidic case, the differences are similarly low above 70 % RH, while they are ~ 0.18 to 0.25 pH units between 70 % and 40 % RH. The deviations between the E-AIM pH and MOSAIC, ISORROPIA II, or EQUISOLV II pH$_\pm$ are clearly larger than those between AIOMFAC–GLE and E-AIM. Even at high RH (> 80 %), the acidity estimates from ISORROPIA II and MOSAIC can differ from each other by almost 1 pH unit (Fig. 10f). Furthermore, the models using mean molal activity coefficients disagree from E-AIM and AIOMFAC-GLE at the highest RH where relatively good agreement is expected due to more dilute

conditions. There are several reasons mentioned above that may be responsible for these deviations. The activity coefficient value (reason (4) above) contributes to the difference between MOSAIC/ISORROPIA II/EQUISOLV II and the E-AIM and AIOMFAC–GLE models, because the mean molal ion activity coefficient used as a substitute for $\gamma_{H^+}$ in first three of these models can either over- or underpredict the single-ion $\gamma_{H^+}$ depending on the solution composition. The differences in pH predictions by ISORROPIA II compared to those by AIOMFAC-GLE and E-AIM at 99 % and 90 % RH (Fig. 10 f) are mainly

because ISORROPIA II yields free H$^+$ molalities ($m_{H^+}$) that are similar at the two RH levels, whereas for E-AIM they differ by a factor of four (higher RH, lower molality). This difference seems to be related to variation in the predicted equilibrium gas–aerosol partitioning of total H$^+$: at 99 % RH the cumulative particle-phase mass concentrations of H$^+$ + HSO$_4^-$ (~ total particle-phase H$^+$) per unit volume of air predicted by E-AIM and ISORROPIA II are similar, but at 90 % RH ISORROPIA II predicts a factor of 6 less total H$^+$ than E-AIM. At similar aerosol liquid water content, this yields a lower $m_{H^+}$ and higher pH than expected at 90 %. Elucidating detailed differences in acidity predictions between the thermodynamic models for nitrate-

containing systems like system 3 should be considered in future work.



Figure 11 compares the different pH estimation options proposed in Sect. 2.3 for use by models that do not predict the single-ion activity coefficient of $H^+$. All calculations (for the systems shown in Fig. 10 and discussed above) were carried out using AIOMFAC-GLE for consistency, and E-AIM is expected to yield similar results. The pH approximations based on total or free $H^+$ molality imply the assumption of an $H^+$ activity coefficient of unity. The comparison in Fig. 11 shows that these two options tend to show larger deviations from the molality-based pH predicted by AIOMFAC–GLE compared to the use of single ion activity coefficients. Both $pH_T$ and $pH_F$ approximate pH within about 0.5 pH units under highly dilute conditions with pH greater than about 3. However, $pH_T$ becomes a poorer approximation of pH when pH values decrease below 3, mainly due to the increasing concentrations of $HSO_4^-$. In the case of the three systems compared here, $pH_F$ is overall a better estimate for pH than $pH_T$, such that use of total $H^+$ is not recommended for atmospheric aerosols. The suitability of $pH_F$ as an approximation may be influenced by the specific system being tested (e.g., RH condition, composition) and needs of the application.

Overall, the AIOMFAC–GLE model results suggest that $pH_\pm$ (with $(H^+, HSO_4^-)$ or $(H^+, Cl^-)$ as the ions for the computation of $\gamma_{\pm,HX}^{(m)}$ used in Eq. 7) is better than $pH_F$ or $pH_T$ in approximating pH.. Computation of the mean molal activity coefficient based on $(2\ H^+, SO_4^{2-})$ leads to a better $pH_\pm$ approximation only in the moderately acidic case of system 1 (Fig. 11a) at RH < 90 %, while it is worse than using $(H^+, HSO_4^-)$ in the highly acidic case of system 1 (open symbols Fig. 11a). Therefore, the use of a 1:1 electrolyte for $pH_\pm$ is recommended. This mean molal activity coefficient approach is recommended when ISORROPIA II, EQUISOLV II, or MOSAIC are used as thermodynamic models for the calculation of aerosol properties.

## 6.2 Ambient scenarios

### 6.2.1 Description of datasets and calculations

Datasets were selected to cover a broad range of acidity, temperature, RH, and species present that drive aerosol pH (Table 4). In addition to the major species $NH_4^+/NH_3$, $SO_4^{2-}$, and $NO_3^-/HNO_3$, the dataset also contained variable concentrations of $Cl^-/HCl$ and the non-volatile species $Na^+$, $Ca^{2+}$, $K^+$ and $Mg^{2+}$ (not shown in the table). A total of more than 7700 data points were available for evaluation (Nenes et al., 2019), with ~7200 data points having relative humidities above 35 % and spanning a temperature range from 252 to 305 K. Given that the MOSAIC box model required – in its current implementation – a manual setup of each input condition, a few data points were selected from the total available in each dataset to compare against the corresponding predictions from the other models. The points were selected to span the range of RH and sulfate amounts encountered in the datasets. Four data points per study location were selected, giving 20 total simulations from MOSAIC to compare against. The MOSAIC inputs are summarized in Supplementary Table S9 with results shown in Fig. S6.





Before thermodynamic calculations are carried out with E-AIM, AIOMFAC-GLE, MOSAIC, and ISORROPIA II, each data point was evaluated to ensure that the resulting thermodynamic solution was atmospherically relevant – i.e., with an alkalinity that does not exceed that of carbonate aerosol (e.g., CaCO₃). Specifically, the charge-equivalent amount of cations is not allowed to exceed the abundance of anions which would result in considerable amounts of hydroxyl ion. In the case of E-AIM

and AIOMFAC-GLE, the composition data were pre-processed and evaluated before input, while MOSAIC and ISORROPIA II evaluate the data automatically and issue error messages or apply adjustments to the input. The input composition data for each model consists of total amounts of $Na^+$, $K^+$, $Mg^{2+}$, $Ca^{2+}$, $TNH_4$, $TCl$, and $TNO_3$, in moles per unit volume of air. The prefix T emphasizes the fact that the final three of the amounts are totals of: $NH_4^+{}_{(aq)}$, $NH_{3(aq)}$, and $NH_{3(g)}$; $Cl^-{}_{(aq)}$ and $HCl_{(g)}$; and $NO_3^-{}_{(aq)}$ and $HNO_{3(g)}$. The amount of $H^+$ needed to achieve charge balance is calculated from:

10       $$Z = [TNH_4] + [Na^+] + [K^+] + 2[Mg^{2+}] + 2[Ca^{2+}] - [TCl] - [TNO_3] - 2[TSO_4], \qquad (19)$$

This equation differs from the strong acidity charge balance proxy introduced in Sect. 4 since it considers the total (gas + particle) amounts of semi-volatile acids and bases rather than exclusively the particle phase as in Eq. (13). If the value of Z is zero, then the system is charge balanced with all $TNH_4$ present as $NH_4^+$ (during a model calculation $NH_3$ can still partition into the gas phase, but this will occur by dissociation of $NH_4^+$). If the value of Z is greater than zero, there is an excess of cations,

and a Z amount of $TNH_4$ is assumed to exist as $NH_3$, and the $NH_4^+$ ion in the system is reduced to $[TNH_4 - Z]$. If the value of Z is less than zero, there is an excess of anions even when all $TNH_4$ is present as $NH_4^+$. In this case, an amount of $H^+$ equal to $-Z$ is added to the system. For E-AIM and AIOMFAC, the calculation of Z and adjustments specified above yield the starting point for the calculation. The amounts of $NH_4^+{}_{(aq)}$, $NH_{3(aq)}$, $NH_{3(g)}$, $HSO_4^-{}_{(aq)}$, $OH^-{}_{(aq)}$, $HCl_{(g)}$ and $HNO_{3(g)}$ in the system at the specified RH and temperature are determined by solving the relevant equilibrium equations. In MOSAIC, the excess $Cl^-$ and

$NO_3^-$ anions are transferred to the gas phase as HCl and HNO₃, in that order, while any excess $SO_4^{2-}$ in the particle phase is balanced by adding $H^+$ to the system. The adjusted gas- and particle-phase concentrations are then used as the initial conditions for further dynamic gas–particle partitioning. In the case of ISORROPIA II, the aerosol is required to be more acidic than aqueous CaCO₃ at given RH, so Z should be less or equal to zero.

The presence of non-volatile cations is handled slightly differently by the models. When calcium is present in ISORROPIA II, the code first forms CaSO₄ as a precipitate (Fountoukis and Nenes, 2007). If there is any remaining Ca and its mole-equivalent exceeds those of $SO_4^{2-}$, $NO_3^-$, and $Cl^-$ combined, an error message is noted and the code assumes that the excess $Ca^{2+}$ is in the form of CaCO₃ and the pH of dissolved CaCO₃ is prescribed at the given RH (see Sect. 7.1 for a discussion of carbonate chemistry and pH). If all Ca precipitates out as CaSO₄, then the ISORROPIA II code examines if the mole-equivalents of $Na^+$,

$K^+$, and $Mg^{2+}$ exceeds that of the $NO_3^-$, $Cl^-$, and remaining $SO_4^{2-}$ combined. If that is the case, an error message is issued, and the excess cations are ignored. Otherwise, the code then uses the inputs of $Na^+$, free $Ca^{2+}$, free $SO_4^{2-}$, etc. to calculate the pH, ALWC, and semi-volatile partitioning of $TNH_4$, $TCl$, and $TNO_3$. A similar approach is taken in MOSAIC, which assumes that the maximum possible amount of CaSO₄ precipitates out over the full RH range, and any excess Ca after forming Ca(NO₃)₂ and CaCl₂ is assumed to be in the form of CaCO₃. MOSAIC does not explicitly treat $K^+$ and $Mg^{2+}$, which are instead represented





by equivalent moles of Na$^+$. Both E-AIM and AIOMFAC also assume that the maximum possible amount of CaSO$_4$ precipitates out and is not considered in the gas–particle partitioning calculations for RH < 98 %. Furthermore, E-AIM does not consider Ca$^{2+}$, K$^+$, and Mg$^{2+}$ in the calculations, but instead uses a charge-equivalent amount of Na$^+$. For AIOMFAC-GLE model input, the electroneutral set of ions is mapped to a set of representative electrolyte components. To facilitate

intercomparison among the models over a wide range in RH, aside from the consideration of the precipitation of solid CaSO$_4$, the models were run using the assumption of the aerosol phase being present as an aqueous electrolyte solution, potentially supersaturated with respect to certain crystalline salts (also referred to as metastable mode in ISORROPIA). Since this assumption becomes invalid at low RH, the statistical evaluation of model–model differences and pH approximations was restricted to the RH range above 35 %, while model calculations were carried out with the supersaturated solution assumption

including data points at lower RH.

During the calculation of the equilibrium composition and corresponding aerosol pH by ISORROPIA II here, all non-volatile cations are converted into their mole-equivalent sodium concentrations. Also, data where non-volatile cation concentrations exceed what is required to neutralize the amount of anions (sulfate, nitrate, and chloride) present are not considered. All models

were allowed to predict partitioning according to their equations and property databases, therefore differences in pH and activity coefficients are a convolution of all differences in the underlying thermodynamic treatment (equilibrium constants, numerical solver tolerance thresholds, calculated activity coefficients, and aerosol water content). A comprehensive accounting of the effects of these differences will be the focus of future work – and here we present only the differences in pH between models and their different implementations of pH approximations.

**6.2.2    pH and its approximations**

The pH values predicted by AIOMFAC-GLE (Fig. 12a) and E-AIM (Fig. 12b) for the combined datasets show both the wide range of pH calculated for each of the datasets as well as variability in the results from the two models. E-AIM and AIOMFAC-GLE agree in their trends but differences increase as RH decreases (as the aqueous aerosols become more concentrated). Differences between the models are usually within 0.5 pH units. The most acidic systems are SOAS (Centreville, AL) in the

southeastern U.S. (pH range: ~ -2 to 2) and WINTER for measurements aloft in the northeast U.S. (pH range ~ -4 to 2), which in part is related to the very low NH$_3$ concentrations and lack of non-volatile cations. The extremely high acidity branch of WINTER data is related to measurements carried out aloft, where temperatures as low as 252 K were encountered. The low humidities in that environment decrease aerosol water to very low levels (Guo et al., 2016). The CalNex dataset is characterized by intermediate pH values, ranging between 0 and 2.5, mostly driven by higher NH$_3$ levels and presence of NVCs. The Tianjin

and Cabauw datasets are characterized by the largest concentration of NH$_3$ and NVCs, and for this reason have the highest pH, reaching a value of 5.





In order to understand the uncertainty introduced by using $pH_F$ or $pH_\pm$ instead of model-predicted pH, model results are examined for each campaign separately. Figure 13 presents the differences between pH predictions and approximations for the Cabauw dataset. Calculations of pH with AIOMFAC-GLE (left column) using the various approximations ($pH_F$, $pH_\pm$) has notably different structure than that using E-AIM (right column). In both models, the difference between $pH_\pm$ and pH rarely

exceed 0.5 pH units, especially for RH above 60 %; $pH_F$ is characterized by larger differences, but still mostly within one pH unit – and reflects the effect of the $\log_{10}(\gamma_{H+})$ contribution which is largest at the lowest RH. These results are consistent with prior studies that assume that the activity coefficient of $H^+$ is equal to unity for the purpose of pH estimation (but not for solving the thermodynamic equilibria) (Song at al., 2018).

Results in Fig. 13 suggest that using a $H^+$–$X^-$ ion pair and applying an activity coefficient in the calculation of $pH_\pm$ gives less scatter and absolute bias across the dataset than using $pH_F$ within a given model framework (E-AIM or AIOMFAC-GLE). The ion pair that leads to the best $pH_\pm$ estimate varies between AIOMFAC-GLE and E-AIM with $H^+$–$Cl^-$ and $H^+$–$HSO_4^-$ showing overall the most promise. The $H^+$–$NO_3^-$ pair tends to exhibit a large scatter and systematic bias in the case of E-AIM, while it shows the least scatter and bias among all ion pairs in the case of AIOMFAC-GLE. Repeating this exercise for all the other

datasets (Supplementary Figures S2 – S5) partly supports these observations – but the pattern and magnitude of the differences (approximate pH minus pH) vary according to the aerosol compositions characteristic of each dataset. For the WINTER data in particular (Fig. S5), biases much larger than 0.5 pH units can be seen at lower humidity (which is especially notable for E-AIM). These deviations can be attributed to the value of the activity coefficient of $H^+$, which becomes very large for the ultra-high ionic strengths characteristic of the WINTER aerosols at intermediate and low RH. The $H^+$ activity coefficient, which

exceeds 10 and can reach up to 100, tends to decrease the pH between 1 and 2 units beyond what is expected from $pH_F$ (Fig. S5 b). The results from AIOMFAC-GLE and E-AIM in Figure 13 and supplementary Figs. S2 – S5 show that the calculated values of the pH approximations differ between the models in quite complex ways, largely reflecting the different treatments of the activity coefficients. These are reflected both in the value of $\gamma_{H+}$, and secondary effects on liquid water uptake, ion dissociation and semi-volatile partitioning.

Both ISORROPIA II and MOSAIC nominally output $pH_F$ (and can be modified to output $pH_\pm$); using approximations ($pH_F$ or $pH_\pm$) in place of pH introduces uncertainty. ISORROPIA-predicted pH approximations as a function of relative humidity compared to AIOMFAC-GLE (left column) and E-AIM (right column, Fig. 14) show that the deviation between $pH_F$ and pH increases as the humidity decreases, with the largest deviations occurring for the extremely acidic aerosol dataset of WINTER.

However, for most cases, relative humidities above 60 % are correlated with a deviation from pH that is less than a unit (smaller differences are seen for AIOMFAC-GLE than E-AIM). Comparisons of MOSAIC calculations, against the predictions from ISORROPIA II for the 20 selected cases (Table S9) indicated the two models produce $pH_\pm$ ($H^+$, $NO_3^-$), $pH_\pm$ ($H^+$, $Cl^-$), and $pH_F$ metrics that are highly correlated ($r^2 \geq 0.96$) with minimal offset (regression slope within 0.11 pH units of 1:1 line) between



the models (Fig. S6). Using the $H^+$–$NO_3^-$ ion pair to express $pH_\pm$ from ISORROPIA provides the closest agreement with pH if AIOMFAC-GLE is used as a reference.

The pH errors between ISORROPIA II and AIOMFAC-GLE/E-AIM for all the datasets combined are summarized in Table

5. Using $pH_\pm$ ($H^+$, $NO_3^-$) as a pH approximation shows the lowest RMSE and mean bias error in the case of AIOMFAC-GLE predictions when considering all field data sets, followed by $pH_\pm$ ($H^+$, $Cl^-$) as the next best approximation. However, when considering E-AIM, the evaluation of all data sets shows that $pH_\pm$ ($H^+$, $Cl^-$) and $pH_\pm$ ($H^+$, $HSO_4^-$) are favored over $pH_\pm$ ($H^+$, $NO_3^-$), as $pH_\pm$ ($H^+$, $NO_3^-$) shows an RMSE of ~ 1 for the WINTER data, which was characterized by the lowest pH values. The comparison between thermodynamic models for the performance of pH proxies include a convolution of numerous errors in

the cases of ISORROPIA II, MOSAIC, and EQUISOLV II; therefore, they cannot be used to determine a priori which choice of anion is best for use in $pH_\pm$ (H, X). Some of the $pH_\pm$ (H, X) variants also show a larger dependence on RH than others, with the largest deviations from pH typically found towards the problematic region of lower RH (< 50 %); see Fig. 13. Based on the combined evaluations of pH approximations by E-AIM and AIOMFAC-GLE against their own pH predictions (no model–model bias incurred), $pH_\pm$ ($H^+$, $Cl^-$) has the best agreement for the wide pH range examined, although any of the pH± variants

work sufficiently well, especially at RH > 60 %.

### 6.2.3    Comparison of proxies to aerosol pH

Several studies have compared certain proxies of acidity (Sect. 4, see also Table 2) to thermodynamic model predictions of $pH_F$ (Guo et al., 2015; Guo et al., 2016; Hennigan et al., 2015; Lawal et al., 2018; Murphy et al., 2017; Winkler, 1986).

Predictions of $pH_F$ using the semi-volatile partitioning approach (Eq. 18) were evaluated in Mexico City (Hennigan et al., 2015), but more commonly, due to the lack of direct aerosol $pH_F$ measurements for comparison, semi-volatile species partitioning is often used as a critical check of thermodynamic equilibrium model assumptions and predictive skill (Guo et al., 2016; Guo et al., 2017a; Guo et al., 2015; Nah et al., 2018). In Fig. 15, comparisons from the literature are extended to include more locations, representing diverse chemical regimes, source influences, and meteorological conditions. Proxies introduced

in Sect. 4 are compared to ISORROPIA II-predicted values of aerosol $pH_F$ using gas + aerosol inputs for four locations (southeast U.S. in summer, California in summer, northeast U.S. in winter, and Tianjin in China in summer; Table 4).

Figure 15 shows that the cation/anion equivalent ratio is fundamentally limited as a proxy for aerosol $pH_F$. The assumption applied throughout the literature is that a cation deficit (anion equivalents > cation equivalents excluding $H^+$; ratio < 1; see

Sect. 4 and Table 2) indicates acidic particles, an anion deficit (ratio > 1) corresponds to alkaline particles. Consequently, a molar equivalent ratio near unity represents near-neutral conditions. Figure 15 shows clearly that these interpretations of the molar ratio are not valid. For a given cation/anion equivalent ratio, predicted $pH_F$ values vary by 3 – 4 $pH_F$ units. All of the data with cation/anion equivalent ratios near unity are predicted to be quite acidic, with $pH_F$ < 3 (and often < 1). The behavior





in Fig. 15 is consistent with observations at locations in Canada (Murphy et al., 2017) and more broadly across the U.S. (Lawal et al., 2018). Even the aerosol predicted by the cation/anion equivalent ratio to be alkaline are actually quite acidic, with $pH_F$ < 3 for almost all of the data where cation/anion > 1. Even if $pH_F$ underestimates pH by 2 units (the maximum underestimation in Fig. 14 a,b) particles would still generally be considered acidic. Common simplifying assumptions associated with the molar ratio method that were discussed in Sect. 4.1 (e.g., considering only $NH_4^+$–$NO_3^-$-$SO_4^{2-}$ or $NH_4^+$–$SO_4^{2-}$) were shown by Guo et al. (2018) to be especially problematic in estimating $pH_F$. Taken together, these results support prior recommendations against use of equivalent ratios as surrogates for particle acidity (Guo et al., 2018b; Hennigan et al., 2015; Lawal et al., 2018; Shi et al., 2017).

Estimates of particle acidity based on an ion charge balance are similarly problematic (Fig. 15a). A charge balance of zero, which corresponds to a cation/anion equivalent ratio of unity, wrongly implies nearly neutral aerosols according to this proxy. Excess cations (negative charge balance, Eq. 13, Fig. 15a), which corresponds to cation/anion equivalents ratios > 1, wrongly implies alkaline conditions. Figure 15 agrees with prior recommendations against using the charge balance as a proxy for particle acidity (Guo et al., 2015; Hennigan et al., 2015; Murphy et al., 2017; Winkler, 1986). The equivalent ratio and charge balance methods both suffer from the same deficiencies, which include: sensitivity to limitations in the precision and accuracy of measurements, not accounting for the buffering effects of many species or the modulating effects of aerosol water, and the non-ideal nature of concentrated aqueous particles, which necessitates the computation of species activity coefficients. As noted above, some usages of strong acidity, a once commonly used parameter to access aerosol acidity health impacts is essentially an ion balance and suffers from similar limitations. A further limitation of the charge balance proxy is the use of an extensive quantity ($H^+_{air}$) to represent an intensive property (pH) of an aerosol distribution, which points to a major design flaw of that approach.

To our knowledge, Fig. 15c represents the first quantitative comparison between GR (the gas ratio proxy, Table 2) and predictions of aerosol $pH_F$. Based on the thermodynamics of gas–particle partitioning, the GR (and adjGR) relationship to pH follows a sigmoidal curve that similarly defines the partitioning of semi-volatile species sensitive to pH (e.g., see Fig. 5 in Pinder et al., 2008a for an illustration of how nitrate PM is a function of GR and adjGR). For the northeast U.S. and California data, the GR follows this sigmoidal behaviour and is strongly correlated with predicted $pH_F$. Increasing GR corresponds to increasing pH, although the slope and intercept of the two data sets differ substantially since they lie on different areas of the curve. In the southeast U.S., the GR results show much larger absolute values than the other locations, since $HNO_3$ measurements were unavailable and the aerosol nitrate values were used as input for $TNO_3$ (Guo et al., 2015). The GR in Tianjin shows no relationship with $pH_F$ (slope = 0.03, $R^2$ = 0.01), even though the data included complete aerosol and gas phase measurements. Although the GR may be highly correlated with $pH_F$ in some environments, it is not advisable to use the GR as a $pH_F$ proxy given the variability observed in Fig. 15. For example, at a given GR, the $pH_F$ range spans ~1 – 4 pH units while the coefficient of determination ranges from 0.01 to 0.75 across the four locations. This suggests that *a posteriori*





knowledge of the $pH_F$–GR relationship is required to use the GR as a proxy for pH. The GR requires aerosol inorganic composition and measurements of both gas-phase $NH_3$ and $HNO_3$. Therefore, with such a data set, $pH_F$ (or pH) can be predicted directly with one of the thermodynamic equilibrium models, which is the recommended approach.

A fundamental limiting factor in using the GR as a proxy for pH is its assumptions about free ammonia. The GR method assumes that under ammonia-poor conditions, where $TNH_4$ is less than $2\times TSO_4$, the aerosol is acidic and $TNH_4$ will partition predominantly to the aerosol phase (Seinfeld and Pandis, 2016). Similarly, the method assumes that ammonia-rich conditions, which exist when $TNH_4$ is greater than $2\times TSO_4$, correspond to largely neutralized aerosols and significant gas-phase $NH_3$ (Seinfeld and Pandis, 2016). While ammonia-rich particles are less acidic in terms of charge balance (Eq. 13) than ammonia

poor particles (all else being equal), a plot of pH vs. GR (Fig. S1) shows that even in ammonia-rich conditions with a high GR, particles do not approach pH near 7. The aerosol is strongly acidic (pH < 1) under ammonia-poor conditions, and a small but significant fraction of $TNH_4$ can exist in the gas phase even though the pH is low. The fraction of $TNH_4$ in the gas phase ($\varepsilon_{NH3}$) approaches 0.1 while the GR < 0, (corresponding to $TNH_4 < 2\times TSO_4$). Likewise, the aerosol remains strongly acidic even under ammonia-rich conditions, where $TNH_4$ exceeds the amount required to neutralize all of the $TSO_4$ (GR > 0). Even when

the amount of $TNH_4$ greatly exceeds all available $TNO_3$ and $TSO_4$, the aerosol remains strongly acidic (approaching a predicted pH of 3.7 as the GR approaches 50 in Fig. S1 example). For such high gas ratios the majority of $TNH_4$ (~0.95) resides in the gas phase. This phenomenon is somewhat counterintuitive: it seems logical that gas-phase ammonia would react completely with acids as strong as $HNO_3$ and $H_2SO_4$ until they were fully neutralized. However, the volatility of $NH_3$ is an important factor that balances the extent to which it reacts with acidic components in the aerosol phase. This explains the insensitivity of

aerosol pH in the southeast U.S., even though sulfate levels are also decreasing while ammonia has remained steady or even increased (Weber et al., 2016).

Aerosol pH calculations based on partitioning of $HNO_3$ and $NH_3$ between the gas and aerosol phases show mixed results when compared to predictions by thermodynamic equilibrium models (Fig. 15d-f). In the southeast U.S., the ISORROPIA and $NH_3$

partitioning-derived $pH_F$ values are moderately correlated, with nearly all values within 1 $pH_F$ unit of the 1:1 line (Fig. 15f). In this case, the pH calculated from $NH_3$ partitioning was systematically lower than the thermodynamic model predictions, a result that was also observed for predictions in Mexico City (Hennigan et al., 2015). In California, the $pH_F$ calculations from $NH_3$ and $HNO_3$ partitioning generally did not agree with the thermodynamic model predictions. The model-predicted $pH_F$ was higher (avg. $pH_F = 2.67$) than the calculation from $NH_3$ (avg. $pH_F = 1.64$) and similar to the one based on $HNO_3$ partitioning

(avg. $pH_F = 2.45$). Although the $pH_F$ calculations from $NH_3$ and $HNO_3$ partitioning lie in the same general area of the graph in Fig. 15, they were inversely correlated with each other (r = -0.76, not shown), an observation that requires further investigation and likely future studies to reconcile. Keene et al. (2004) also observed disagreement between the $pH_F$ calculations from $NH_3$ and $HNO_3$ partitioning. In the northeast U.S., the phase partitioning of $HNO_3$ gave mixed results, as well. At times, the predicted and calculated $pH_F$ values agreed well, while at other times there were differences of ~2 – 3 $pH_F$ units. The greatest





discrepancies were observed at the lowest aerosol liquid water contents (mass fraction basis, Fig. 15e), a relationship also identified by Guo et al. (2016) and consistent with the idea that activity becomes harder to predict at lower water content (Sect. 6.1, Fig. 10). Potential problems with $pH_F$ calculated from semi-volatile species partitioning have been discussed (Keene et al., 1998; Keene and Savoie, 1998; Young et al., 2013). The approach requires measurements of at least one semi-volatile gas-phase species and the aerosol inorganic composition, which are input into a thermodynamic model to get the ALWC, a required component to calculate condensed-phase activities in Eq. 16 and needed to use typical Henry's law coefficients in Eq. 17 and 18. Therefore, given the need for all these inputs, thermodynamic models should be used to directly predict pH or one of its approximations.

## 6.3 Recommendations on the calculation of pH by approximation and proxy

Where single-ion activity predictions are not available, the comparison based on the ambient datasets used here suggests that the best pH approximation is obtained by using Eq. 7 for $pH_\pm$. However, identifying a universal $H^+$–anion pair that best reproduces pH appears to be model-dependent with only $pH_\pm$ ($H^+$, $NO_3^-$) having the potential to be a worse estimate of pH than $pH_F$ (in the case of E-AIM). Although, on average, all of the approximate measures of pH compare similarly against pH from AIOMFAC-GLE and E-AIM, there is a strong dependence of the bias on RH that is mitigated through the use of $pH_\pm$ (as opposed to $pH_F$), and for this reason it is the recommended approach when ISORROPIA, MOSAIC, EQUISOLV II or similar models are used for calculations of pH in the future. Low RH also coincides with time periods where models (both box and chemical transport models) face challenges in accurately predicting gas–particle partitioning (e.g., Guo et al., 2016; Kelly et al., 2018) thus motivating a need to properly characterize acidity under those conditions.

Based on the analyses and discussion presented in this section and Sect. 4, it is strongly recommended that proxies are avoided in the analysis of particle acidity. Some of the proxies correlate with $pH_F$, even strongly at times, although this varies greatly with ambient conditions (T, RH), composition, and concentration. This leads to large inconsistencies across locations, and even within a given observational data set. Often, the proxies are not able to qualitatively distinguish acidic from neutral particles or to capture qualitative trends in acidity (e.g., pH increases or decreases with a given indicator). A detailed comparison with thermodynamic equilibrium model predictions constrained with aerosol and gas inputs is required to identify the periods and locations where a proxy may perform adequately; defeating the purpose of using the proxy. With the open access and web-based availability of validated aerosol thermodynamic equilibrium models (Sect. 2.6), scientists are encouraged to use one or more of these tools in future studies of particle acidity.





## 7 Role of particle size, composition, and mass transfer kinetics in pH heterogeneity

Traditionally, fine particle pH is calculated assuming equilibrium and a uniform distribution of species across all particles (e.g., Sect. 5.1 and 6). Here, the role of differences in particle size, mass transfer, and composition (including presence of organic species) in driving pH in a population of particles is highlighted.

### 5    7.1    Role of particle size and composition

The pH of aerosols varies with particle size because of the differences in the chemical composition, hygroscopicity, and gas–particle equilibration time scales between fine and coarse particles. Fine-mode aerosols are produced by new particle formation and growth but are also directly emitted from anthropogenic as well as natural sources (dust and sea salt). At least some of the chemistry that initiates new particle formation, and thus drives low pH for the smallest sizes, involves sulfuric acid and acid-

base reactions (Kulmala et al., 2004). Anthropogenically-derived fine mode aerosols are typically composed of inorganic salts, organic species, and black carbon and are generally acidic. Fine-mode pH is sensitive to the relative amounts of non-volatile cations (if any), sulfate, nitrate, and ammonium present in the particle phase (Fig. 2) and continuously responds to the changing concentrations of their gas-phase counterparts—$H_2SO_4(g)$, $HNO_3(g)$, and $NH_3(g)$ —as well as the ambient RH and temperature.

In contrast, coarse mode aerosols mainly consist of sea-salt and dust particles directly emitted to the atmosphere as a result of wind stress on the surface of the oceans and arid land, respectively. Sea salt and dust contain significant amounts of non-volatile cations such as $Na^+$, $Ca^{2+}$, $Mg^{2+}$, and $K^+$, whereas the dominant cation in fine mode particles is typically semi-volatile ammonium. Dust, and more generally non-volatile cations, can also originate from mechanical wear or disturbances associated

with anthropogenic activity such as road, residential, and commercial construction as well as brake wear and road salt application (Philip et al., 2017; Lough et al., 2005; Kolesar et al., 2018). Both fossil fuel and biomass combustion also emit non-volatile cations in the $PM_{2.5}$ size range (Reff et al., 2009). Whether or not non-volatile cations influence pH depends on their mixing state with deliquesced particles. In other words, particles of the same size but different compositions should be treated as external mixtures when calculating their pH. For example, equilibrium model analysis of bulk ambient aerosol

observations by Guo et al. (2016) indicate that the refractory ions were externally mixed from $PM_1$ because including those ions caused deviation between the predicted and measured nitrate partitioning. A small fraction of non-volatile aerosol components is sometimes present in the fine mode and tends to reduce acidity. For example, $K^+$ associated with biomass burning has been shown to cause higher pH compared to cases with very low $K^+$ levels (Bougiatioti et al., 2016).

Sea-salt and dust are naturally basic or alkaline, as they contain carbonates. The pH of ocean water ($\approx 8$) is relatively uniform and sets the $pH_F$ for unprocessed sea-salt emissions (Keene et al., 1998). The pH of fresh airborne dust is more difficult to assess due to the high degree of heterogeneity in composition and its hygroscopicity; however, it is very likely that ambient





dust is not acidic (has pH > 7). Sea salt and dust aerosol can initially maintain high pH (above 5 and close to 7) due to the presence of carbonate ($CO_3^{2-}/HCO_3^-$). However, the uptake of acid gases such as $SO_2(g)$, $HCl(g)$, $HNO_3(g)$ and $H_2SO_4(g)$ result in a chemical reaction (Usher et al., 2003) such as:

$$CO_3^{2-}(aq) + 2H^+(aq) \longrightarrow CO_2(g) + H_2O(l) \tag{R4}$$

Reaction R4 consumes $H^+$ produced from the uptake of acid gases (e.g., $HNO_3(g) \longrightarrow H^+(aq) + NO_3^-(aq)$, Reaction R1 and R2 combined), allowing the aerosol to maintain its high pH until the carbonate has been depleted via conversion to $CO_2(g)$. Once the carbonate has been depleted (a process not treated by current equilibrium models, Sect. 2.6), the dust and sea-salt aerosol can become acidified by continued uptake of acid gases. Observations of aged sea salt and dust show an internal mixture with sulfate, nitrate, and chloride due to such reactions (Fairlie et al., 2010; Kirpes et al., 2018; Tobo et al., 2010). Freshly emitted

sea-salt aerosol is in the liquid state while Ca-rich dust particles are emitted as solids. Consequently, acidification of sea salt aerosol is thought to proceed more efficiently due to relatively high mass accommodation coefficients (about 0.1 or higher) for condensing acids on liquid particles compared to solid dust aerosol with much lower uptake coefficients ranging between $10^{-4}$ and $10^{-3}$ (Alexander et al., 2005; Fairlie et al., 2010). The increase of aerosol water with increasing RH and the solubilization of gaseous HCl that is present in the marine boundary layer (due to acid displacement reactions) has also been

suggested as the reason for increasing acidity of sea-salt aerosols with increasing RH and altitude in the marine environment (von Glasow and Sander, 2001). Further acidification of sea salt aerosols occurs via displacement of $Cl^-$ as $HCl(g)$ due to reactions such as (Mcinnes et al., 1994; Zhao and Gao, 2008):

$$HNO_3(g) + Cl^-(aq) \longrightarrow HCl(g) + NO_3^-(aq) \tag{R5}$$

$$H_2SO_4(g) + 2Cl^-(aq) \longrightarrow 2HCl(g) + SO_4^{2-}(aq) \tag{R6}$$

Although these reactions do not directly produce additional $H^+$ ions, the resulting $H^+$ molal concentration increases due to a decrease in the overall aerosol water content in particles containing $NaNO_3$ and $Na_2SO_4$, which are less hygroscopic than $NaCl$.

Overall, atmospheric particle pH is size dependent and generally higher for coarse mode particles due to variations in inorganic composition with particle size. Differences as large 4 pH units have been reported between fine and coarse particles (Fang et

al., 2017; Young et al., 2013). Bulk $PM_1$ and $PM_{2.5}$ acidity is more similar than fine vs coarse mode acidity ($pH_F$ within $1 - 2$ units, e.g., Bougiatioti et al., 2016; Guo et al., 2017b), but submicron (diameter < 1 μm) particles still show higher acidity than bulk $PM_{2.5}$. The reason for this is the strong enrichment of aerosol with NVCs from dust and sea salt at the larger sizes (even in the fine mode) and role of sulfate in new particle formation and surface-area driven condensation at the small sizes (Fig. 2). Significant pH changes can occur in the 1 to 2.5 μm size range (Fang et al., 2017; Ding et al., 2019). The size dependent pH is

also seen for sea salt aerosol (Fridlind and Jacobson, 2000) as well as in urban aerosols in China (Ding et al., 2019) where the fine mode is consistently 2-3 pH units lower than the coarse mode. The implications of this acidity gradient are considerable, for metal solubility and their impacts on public health and ecosystem productivity, as well as chemistry and semi-volatile partitioning of pH-sensitive species.



## 7.2 Role of mass transfer

Acidity is dependent on particle composition, and particle composition can be affected by mass transfer rates that vary by particle size. For fine mode particles, the characteristic time for particle growth or shrinkage from one equilibrium state to another after changes in RH is short enough (< 1 s) to justify the assumption of thermodynamic equilibrium with respect to water uptake (Pilinis et al., 1989). In comparison, equilibration of semi-volatile components ($HNO_3$, HCl, and $NH_3$) with the fine mode ranges from 20 min or less (Guo et al., 2018b) up to 10 hours (Meng and Seinfeld 1996; Fridlind and Jacobson, 2000). In the case of coarse mode aerosols or large accumulation mode aerosols, mass transfer rates for semi-volatile components can lead to equilibration time scales of several hours. Hanisch and Crowley (2001), for example, found vapors of $HNO_3$ reach equilibrium through uptake by sea spray aerosols of 1-3 μm diameter within 3-10 h. In another study of remote marine aerosols, equilibrium in the coarse sea salt mode is reached quickly for $NH_3$, but $HNO_3$ and HCl require much longer times, of the order of 10 – 300 hours (Fridlind and Jacobson, 2000). In this case, relatively small amounts of $TNH_4$ partition to the coarse sea salt particles compared to much larger amounts of $HNO_3$ needed to displace HCl to reach equilibrium. These time scales are comparable or can even exceed the lifetime of the particles, implying that some particles can be removed by deposition before equilibrium is reached (Fridlind and Jacobson, 2000). In a subsequent theoretical study, Jacobson (2005a) found that under at least some conditions equilibrium can be reached within less than 1 h by large particles (< 6 μm) and within 15 min by particles < 3 μm, while in several other cases coarse particles took longer to reach equilibrium. Thus, aerosols of different sizes within the fine and coarse modes may not always be in mutual equilibrium due to mass transport limitations, and equilibrium alone may not uniquely determine the distribution of condensed semi-volatile gases across the particles of different sizes (Wexler and Seinfeld, 1990, 1992).

Given the above, both mass transport and thermodynamics must be considered to accurately predict the distribution of semi-volatile gases and the associated aerosol pH across the entire aerosol size spectrum. However, simulating mass transfer and thermodynamics for the size- and composition-distributed aerosol is computationally challenging due to numerical stiffness. There are two main sources of numerical stiffness. The first source arises from the large differences in the mass transfer time scales for particles of different sizes. Additional stiffness and non-linearity are introduced by $H^+$ ions in partially and fully deliquesced aerosols. In such cases, the $H^+$ ion molal hydrogen ion concentration ($m_{H^+}$) plays a crucial role in the determination of equilibrium aerosol phase state as well as in the determination of equilibrium gas-phase concentrations of $HNO_3$, HCl, and $NH_3$ at the particle surface for computing their driving forces for mass transfer. The characteristic time scale for $H^+$ ions is quite short relative to other species, especially under "acid-neutral" or "sulfate-poor" conditions, where the pseudosteady-state concentrations of $H^+$ ions are two or more orders of magnitude smaller than the sum of all other cations (Sun and Wexler, 1998). Since semi-volatile species in different particles of different sizes are coupled via the gas phase, the numerical solver for mass transfer would have to take time steps on the order of the shortest timescale to ensure accuracy for all the species across the entire aerosol size distribution. Such small time-steps are computationally prohibitive for common chemical





transport model applications. Several attempts have been made over the past 20 years to reduce the stiffness of the system of nonlinear ordinary differential equations that describe the multicomponent, size-distributed mass transfer problem so that it could be efficiently solved (Capaldo et al., 2000; Hu et al., 2008; Jacobson, 1997; Jacobson, 2002, 2005a; Jacobson et al., 1996; Pilinis et al., 2000; Sun and Wexler, 1998; Zaveri et al., 2008; Zhang and Wexler, 2006).

Here, we illustrate the time-evolution of size-distributed pH using the sectional MOSAIC box model with 60 size bins for a scenario (test case 14 in Zaveri et al., 2008) in which fine mode aerosol composed of $(NH_4)_2SO_4$ and coarse mode aerosol composed of NaCl were exposed to appreciable gas-phase concentrations of $H_2SO_4$ (1 ppbv), $HNO_3$ (15 ppbv), HCl (1 ppbv), and $NH_3$ (10 ppbv) at 85% RH and 298.15 K temperature (Fig. 16). While the fine mode rapidly absorbs significant amounts

of these gases within the first few minutes of the simulation, it takes nearly 10 h for the aerosol composition, and hence the pH, to become uniform across the bins of different sizes. Furthermore, the displacement of HCl from the coarse mode due to $HNO_3$ absorption occurs slowly over this time, although significant differences in the pH can be seen across the coarse mode size bins even after 10 h. In conclusion, it is important to treat dynamic mass transfer to accurately simulate size-distributed pH and composition of aerosols. Although challenging, fully dynamic and hybrid (i.e., a combination of equilibrium for fine

mode and dynamic for coarse mode) numerical methods have been implemented in 3D chemical transport models (Fast et al., 2006; Jacobson et al., 2007; Zhang et al., 2010).

### 7.3 Role of organic–inorganic interactions

Aerosol particles are rarely composed of a completely distinct organic-free aqueous inorganic phase and electrolyte-free organic phase – an assumption often made in air quality models for reasons of simplicity. Instead, mixed particles exist

consisting of a complex mixture of organic compounds, inorganic ions, and water that may be separated into multiple liquid/solid phases (Bertram et al., 2011; Hallquist et al., 2009; Maria et al., 2004; Murphy et al., 2006; Pöhlker et al., 2012; Song et al., 2012; Zuend et al., 2010). The role of organic–inorganic interactions on the acidity of liquid/amorphous aerosol phases has been addressed in only a few studies and represents an area of research where further efforts are needed. Particle phase acidity could be affected in multiple ways by organic–inorganic interactions: directly by means of non-ideal mixing

effects on the activity coefficient of $H^+$ (and all other species) in a liquid phase of given composition; indirectly via the effect of organics on composition and the equilibrium gas–particle partitioning of water and other semi-volatile components (including $NH_3$, inorganic and organic acids), as well as the potential for LLPS; and directly by dissociating organic acids that contribute dissolved $H^+$ or amines that associate with $H^+$.

A phase-separated particle typically consists of a rather hydrophobic organic-rich phase and an aqueous electrolyte-rich (salt/ion rich) phase (You et al. (2014) and references therein) (Fig. 17). Note that water and inorganic ions, including $H^+$, can exist in the organic-rich phase of a liquid–liquid phase separated system (Pye et al., 2018; Zuend and Seinfeld, 2012). Both the detection of LLPS and pH in ambient particles as well as micron-sized droplets in laboratory experiments is a difficult





technical challenge (Wei et al., 2018). To our knowledge, no online measurement techniques applicable to field sampling exist for that purpose (see Sect. 5.1 for aerosol pH measurement challenges). The current state of knowledge is therefore limited to relatively simple laboratory systems and theoretical considerations. Dallemagne et al. (2016) used a model system in the form of a super-micron-sized ternary aqueous poly(ethylene glycol)-400 (PEG-400) + ammonium sulfate droplet. They studied this

system in an RH and temperature-controlled cell with confocal microscopy in the presence of a pH-sensitive fluorescent dye to determine the pH value at different locations in the liquid drop. They report a small, yet distinct change in pH due to the phase transition from a single to two liquid phases for this system when RH decreases: pH = 3.8 ± 0.1 in a single mixed phase at > 90 % RH, while the organic-rich shell phase in a LLPS state exhibited pH = 4.2 ± 0.2 at 80 % RH to pH = 4.1 ± 0.1 at 65 % RH; the pH in the sulfate-rich phase was not determined during LLPS. The pH value of the organic-rich phase was

similar to that of a corresponding salt-free aqueous PEG-400 solution measured using a standard pH probe. Since changes in RH lead to changes in particle water content, here causing the LLPS, the degree to which such changes affected the measured pH in the Dallemagne et al. (2016) study remain unclear.

Losey et al. (2016) controlled the pH in aqueous solution droplets consisting of 3-methylglutaric acid, ammonium sulfate and

sodium hydroxide. They found that changes in pH and the degree of methylglutaric acid dissociation (deprotonation), affect the separation RH (SRH), the onset of LLPS during dehumidification. The SRH was ~79 % for pH = 3.65, ~70 % for pH = 5.17, and ~64 % for pH = 6.45. The RH at which the two liquid phases merge into a homogeneous single phase was observed around 80 % RH in this system, approximately independent of pH – indicating that a hysteresis between SRH and merging RH occurs for pH close to neutral, but not at lower pH. While this study did not attempt to measure the pH in distinct liquid

phases, it indicates that the established pH, resulting from interactions between inorganic electrolytes and organic acids, affects the LLPS behavior. Losey et al. (2018) further explored similar systems at higher acidity in the presence of sulfuric acid (varying the ammonium-to-sulfate ratio from 2.0 to 1.5 to 1.0). They report that all observed RH levels of phase transitions were affected by the pH established with sulfuric acid. The SRH consistently decreased with increasing amounts of sulfuric acid (toward lower pH); e.g., for 3-methylglutaric acid + ammonium sulfate + sulfuric acid from SRH of ~80 % at pH = 2.68

to SRH of ~30 % at pH = 0.34. Similar lowering of SRH with increasing acidity was also found for a non-acidic organic mixture component (1,2,6-hexanediol). Furthermore, at high acidity (here pH lower than 0.5), several of the studied systems did not show any LLPS down to very low RH. While the quantitative phase transition behaviour depends on the organic component, these experiments by Losey et al. (2018) imply that organic–inorganic interactions can have an impact on mutual solubility and phase transitions, in those cases with increasing mutual solubility towards higher acidity.

While a LLPS will impact the acidity in coexisting liquid phases, the extent to which the pH values will typically differ between the phases – and, related to that, the molar concentrations of hydronium ions and ionic strength – remains an open question. Theoretical considerations aid in constraining the range of expectations in this case. Thermodynamic equilibrium between two liquid phases, each of neutral electric charge, implies that the electrochemical potential of H$^+$ ions is equivalent in both phases



(see Sect. 2). Therefore, the activity-based pH in coexisting phases is expected to be similar, but not necessarily of the exact same value. Computations with the AIOMFAC-based liquid–liquid equilibrium model confirm for case studies that the pH in two liquid phases is of the same order of magnitude, often with a difference of less than 0.2 pH units (Pye et al., 2018). However, $H^+$ molalities (or concentrations) in the two coexisting phases of atmospheric aerosols are predicted to be very

different, often by up to several orders of magnitude; hence, it is important to calculate pH based on $H^+$ activity, not simply concentration (see also Sect. 6.3 recommendations for approximating pH). In Pye et al. (2018), several thermodynamic models were applied to predict the partitioning of ammonia, water, and organic compounds between the gas and particle phases for conditions in the southeastern U.S. during summer 2013. AIOMFAC-based coupled liquid–liquid and gas–particle partitioning computations within that study predicted partial to complete miscibility among organic and inorganic aerosol components,

depending on RH. The AIOMFAC-based model predicted an increase in the concentration of gas-phase ammonia ($NH_3$) alongside a decrease in acidity when partial miscibility of organics was accounted for. In comparison to calculations with complete phase separation between organic and inorganic ions enforced, the interactions of inorganic ions with organic compounds (in mixed phases) were predicted to promote an enhanced association of $H^+$ and $SO_4^{2-}$ into $HSO_4^-$, resulting in a slightly higher pH (0.1 pH units median increase), since the bisulfate ion is predicted to be more miscible with organic

compounds than equivalent amounts of $H^+$ and $SO_4^{2-}$ (Pye et al., 2018). This indicates a pH buffering effect of the degree of bisulfate dissociation; however, additional complexity in understanding the main drivers of such pH changes arises from simultaneous changes in the equilibrium gas–particle partitioning of water, organics, and ammonia.

The impact of amines and organic acids on $H^+$ is usually neglected in efforts to model pH. Amines may contribute to aerosol

alkalinity – especially given their potentially strong proton affinity (Dall'Osto et al., 2019), but they must be in sufficient quantities to compete with $NH_3$ and other cations. Although not strong sources of protons or cations, these alkaline and acidic organics may still be considered together with other water-soluble organic compounds (WSOCs) in the particulate phase in terms of their ability to influence the aerosol water content. The uptake of water due to organic components is often used to correct the solvent volume and $pH_F$ derived based on the inorganic aerosol composition (e.g., Guo et al., 2015; Bougiatioti et

al., 2016). This implies that aerosol pH is reversibly influenced by the amount of water (driven by RH and composition) associated with the aerosol particles, which has been shown to drive some of the diurnal variability of pH (Guo et al., 2015).

For systems where a single mixed aerosol phase is assumed, current work indicates dissociating organic acids do not strongly affect pH and the limited studies to date suggest that inorganic species drive pH (Battaglia et al., 2019; Song et al., 2018;

Vasilakos et al., 2018). For the southeastern U.S., pH changes predicted by E-AIM were generally limited to < 0.2 pH units in response to dramatic increases in oxalic acid (Vasilakos et al., 2018). Similarly, E-AIM predicted that increases in oxalic acid concentrations resulted in < 0.1 pH unit changes for polluted Beijing conditions (Song et al., 2018). This is notable since the predicted pH in Beijing (neglecting organics) was consistently above the first acid dissociation constant ($pK_{a1}$) value for oxalic acid, conditions where pH is predicted to be most sensitive to organic acids (Nah et al., 2018). Nah et al. (2018) showed that

for aerosol $pH_F$ varying between 0.9 and 3.8, the inorganic-only predicted $pH_F$ was sufficient to define an effective sigmoid curve for oxalic acid, one of the most abundant of organic acids with a $pK_a$ that is well within this range. Neglecting the effects of oxalate on pH by Nah et al. (2018) did not seem to affect the quality of the partitioning. Battaglia et al., (2019) extended these prior studies to include additional organic acids (oxalic, glutaric, and malonic acids) as well as three non-acid organics (levoglucosan, tetrahydrofuran, and 1-pentanol) mixed with inorganics representative of Beijing winter haze and eastern U.S. summertime compositions. The changes in pH relative to the inorganic-only system were predicted by AIOMFAC to be quite small, generally < 0.2 pH units, when a single aerosol phase was present (Battaglia, Jr., et al., 2019). The response of pH to the same organics at lower RH (< 70%) or under LLPS conditions was not characterized.

While current work suggests organic–inorganic interactions only slightly affect the pH, they can drive both LLPS and other phase transitions. Based on case studies (Pye et al., 2018; Battaglia et al., 2019), the interactions between water and ions are likely the main determinants of the resulting pH value. However, considering the complexity and variability of realistic aerosol compositions, the extent to which organic–inorganic interactions moderate the pH in liquid phases has not yet been studied in depth.

## 8     Regional and global model representations and usage of $pH_F$

Chemical transport models and climate models are the ultimate integrators of knowledge that link emissions to the endpoints of public health, climate, and deposition. Aerosol acidity, however, is almost never considered or reported in these large frameworks (although there are exceptions, e.g., TM4-ECPL reported model-predicted $pH_F$ for clouds and particles; Fig. S2 of Myriokefalitakis et al., 2015), so potentially large differences in acidity may be a driver of bias that has been unidentified to date. In the following section, major features of a set of models (Community Multiscale Air Quality modeling system, CMAQ; Goddard Earth Observing System with Chemistry model, GEOS-Chem; TM4-ECPL; and Weather Research and Forecasting coupled with chemistry WRF-Chem) are summarized in terms of fine aerosol $pH_F$ predictions (Sect. 8.1). The cloud $pH_F$ from a subset of the CTMs listed above and the Community Atmosphere Model with Chemistry (CAM-Chem; Lamarque et al., 2012; Tilmes et al., 2015), a component of the NCAR Community Earth System Model (CESM), are also included (Sect. 8.2).

### 8.1    Aerosol $pH_F$

All three-dimensional CTMs presented here use thermodynamic models to predict aerosol composition and thus $PM_{2.5}$ predictions are sensitive to $pH_F$. Thermodynamic models for the inorganic system were initially implemented to predict the gas–particle partitioning of semi-volatiles including nitric acid and ammonia due to their importance in forming fine particulate matter, but later studies have leveraged the predicted acidity for acid-mediated reactions. In TM4-ECPL, the pH of clouds and aerosol water affects the equilibria and thus chemistry of organic acids as well as the partitioning of reactive nitrogen and the





solubilization of the trace elements iron and phosphorus. TM4-ECPL explicitly accounts for interconversion of Fe (II) and Fe (III) and formation of oxalate (the partitioning of which is also pH-sensitive, e.g., Nah et al., 2018) that acts as a ligand and contributes to secondary organic aerosol. This chemistry has been used to understand changes in oceanic deposition of Fe and P from preindustrial, present-day, and future atmospheres (Myriokefalitakis et al., 2015; Myriokefalitkis et al., 2016) as well

as with regional focus on the Mediterranean region (Kanakidou et al., 2019). The CMAQ v5.1+ (Pye et al., 2013) and GEOS-Chem v11-02+ (Marais et al., 2016a) models use particle acidity, although in slightly different forms, to mediate the uptake of isoprene epoxydiols and resulting production of secondary organic aerosol in PM$_{2.5}$. For purposes of acid-catalyzed particle-phase reactions, GEOS-Chem uses ISORROPIA II-predicted pH$_F$ (Marais et al., 2016a) while CMAQ v5.1 and later consider the entire internally mixed fine-mode particle phase abundance in calculating the concentration of H$^+$ (Pye et al., 2013). In

CMAQ, organic constituents act to dilute H$^+$ (increase pH$_F$ when the solvent includes organics) relative to an externally-mixed or phase-separated assumption (Schmedding et al., 2019). This leads to a moderate correlation between acidity (expressed as 10$^{-pHF}$) and isoprene-derived organic aerosol constituents (r$^2$=0.3-0.5) (Budisulistiorini et al., 2017) for the SE US, in contrast to acidity pH$_F$ estimates under an externally-mixed or inorganic-only solvent assumption that show no significant correlation with isoprene SOA (Budisulistiorini et al., 2015). The WRF-Chem model, configured with MOZART chemistry and MOSAIC

aerosols with the MESA thermodynamic model, uses particle acidity to calculate SOA production from glyoxal (Knote et al., 2014). Even though aqueous production of sulfate in clouds is mediated by cloud pH, heterogeneous sulfate production on aqueous aerosol (via pathways in Fig. 3) is generally not considered in models, but future efforts may include these pathways due to model underestimates of sulfate in regions like Beijing, China (e.g., Shao et al., 2019; Cheng et al., 2016) and Fairbanks, Alaska (Molders and Leelasakultum, 2012).

Chemical transport models use a variety of thermodynamic box models depending on their needs for accuracy and efficiency or treatment of specific systems and processes. The MESA thermodynamic model is used in CTMs configured with the MOSAIC aerosol model (e.g. WRF-Chem; Fast et al., 2006). ISORROPIA II is employed in several CTMs including GEOS-Chem (v8-03-01 and later), the CMAQ (v5.0 and later) modeling system, NASA GISS, WRF Polyphemus 1.6, the Tracer

Model (TM) v4,5-ECPL family of models (Appel et al., 2013; Metzger et al., 2018; Myriokefalitakis et al., 2011; Pye et al., 2009), PM-CAMx (both regular and UF versions), and some versions of WRF-Chem (e.g., Zhang et al., 2013). GEOS-Chem, CMAQ, and TM4-ECPL assume the fine particles are in metastable equilibrium with the gas phase and employ the forward (i.e., gas and aerosol precursors as input) calculation mode of ISORROPIA II to partition semi-volatiles, calculate liquid water content, and pH$_F$. While stable vs metastable assumptions strongly affect the amount of liquid water content and may influence

the resulting composition of the aqueous phase, Song et al. (2018) found that calculations assuming stable and metastable state yield similar results in terms of pH$_F$ when the aerosol is deliquesced for conditions in China. The generality of this finding, especially when the complex phase diagram associated with eutectics of multiple salts is fully considered, remains to be determined.



The pH of the coarse mode is treated to varying degrees in models. TM4-ECPL (Myriokefalitakis et al., 2015) applies the equilibrium assumption to internally mixed sulfate, nitrate, ammonium, sea-salt and dust aerosols in the coarse mode after equilibrating the fine mode aerosol. CMAQ, starting with v4.7, uses a hybrid approach to mass transfer (Kelly et al., 2010) where the internally mixed Aitken and accumulation modes are in equilibrium with the gas phase, and mass transfer with the

coarse mode is treated dynamically using the difference between the ambient and equilibrium vapor pressure of semi-volatiles (computed with ISORROPIA II in reverse mode in CMAQ v5.0 and later) as a driving force for condensation/evaporation (Capaldo et al., 2000). This driving force, however, is not allowed to exceed the gas-to-particle diffusional limit prescribed in CMAQ which would result in numerical instability when the aerosol pH is mildly acidic to alkaline (see Sect. 7.2 for additional discussion) (Pilinis et al., 2000). While ISORROPIA is not recommended for estimating $pH_F$ in field or laboratory applications

when only particle composition is available (reverse mode, open system, see discussion in Hennigan et al., 2015 and Song et al., 2018), the reverse mode can be used for a driving force in a chemical transport model since CTMs represent a closed system, species concentrations are not subject to measurement error, and the driving force can be capped at the diffusion limitation. For coarse particles in CMAQ, $H^+$ determined via charge balance (assuming all particulate sulfur is in the form of sulfate, Eq. 13) is output for diagnostic purposes but is not used within the model. GEOS-Chem does not perform

thermodynamic calculations for coarse particles. However, it does keep track of coarse-mode sea-salt and dust alkalinity, which is relevant for calculating heterogeneous reactions on coarse particles. For example, heterogeneous $S(IV)+O_3$ only happens in the model when the sea-salt and dust aerosol is still alkaline (Alexander et al., 2005). Heterogeneous reactions between hypohalous acids and halide ions (e.g., HOBr+Br-) on sea salt aerosol are acid catalyzed, so these reactions are only allowed to occur in the model after the sea-salt alkalinity has been titrated (titrated coarse-mode pH is assumed to be equal to 5)

(Sherwen et al., 2016). The Weather Research and Forecasting model (Powers et al., 2017; Skamarock et al., 2008) coupled with chemistry (WRF-Chem version 3.9.1; Fast et al., 2006; Grell et al., 2005) has four aerosol configurations, including a bulk aerosol scheme, two modal aerosol schemes, and a sectional aerosol scheme (8 or 4 bin). Figures 19 and 21 use the 4-bin sectional aerosol scheme (bins 0.039–0.156, 0.156–0.625, 0.625–2.5, 2.5–10 μm in diameter) with MOSAIC coupled to MOZART (Model for Ozone and Related Chemical Tracers, version 4) gas-phase chemistry and cloud water chemistry (Knote

et al., 2015; Zaveri et al., 2008). Within MOSAIC, the multi-component equilibrium solver for aerosols (MESA, Zaveri et al., 2005a) solves inorganic aerosol thermodynamics for each aerosol size bin (see also Sect. 2.6.3 for a description of MOSAIC).

Even models using the same thermodynamic algorithms can produce different $pH_F$ estimates since CTMs differ in their assumptions regarding equilibrium, mixing state, emission speciation, composition distribution across size, and chemical

constituents important for driving pH. Especially notable are differences that occur with respect to the presence, abundance, and mixing state of non-volatile cations, the presence of which tends to increase $pH_F$ – and may further elevate $pH_F$ by co-condensation of nitrate and its associated water uptake (e.g., Guo et al., 2018a). In TM4-ECPL, $SO_4^{2-}$, $NH_3$, $NH_4^+$, $HNO_3$ and $NO_3^-$ are explicitly treated. Additional cations in TM4-ECPL (for fine-submicron- and coarse modes) are specified based on the composition of mineral dust and sea salt. However, TM4-ECPL does not include Ca and Mg from dust in the fine mode



calculations, as also proposed by Ito and Feng (2010), since TM4-ECPL considers all submicron dust and sulfate aerosol to be externally mixed in the atmosphere. The opposite assumption is made for coarse particles in TM4-ECPL since sulfate and nitrate are produced by heterogeneous processes on coarse particles (leading to an internal mixture) and coarse particle lifetime is short (decreasing the likelihood of distinct source plumes interacting). GEOS-Chem, employing a bulk scheme for fine

aerosol, considers $SO_4^{2-}$, $NH_3$(g), $NH_4^+$, $HNO_3$(g), $NO_3^-$, $Na^+$, $Ca^{2+}$, $Mg^{2+}$, and $Cl^-$ from fine-mode sea-salt aerosol. Cations from dust are not included in GEOS-Chem fine aerosol $pH_F$ calculations by default (version 12.0.0), but $Ca^{2+}$ and $Mg^{2+}$ from fine-mode dust aerosol (assuming 3% and 0.6% by mass dissolution, respectively; Fairlie et al., 2010) were added to simulations in this work (see Fig. S7). CMAQ v5.0 and later, with Aitken and accumulation modes for fine aerosol, considers non-volatile cations from sea-salt, wildfires, wind-blown dust, and anthropogenic sources such as fugitive road dust,

agricultural soils, and coal combustion as described by the EPA National Emission Inventory (NEI) and SPECIATE database (Reff et al., 2009). WRF-Chem configured as MOZART-MOSAIC represents the major aerosol species including sulfate, $MSA^-$, $NO_3^-$, $Cl^-$, $CO_3^{2-}$, $NH_4^+$, $Na^+$, $Ca^{2+}$ in the charge balance for $H^+$ (other inorganic species, primarily dust particles, are considered inert, Sect. 2.6.3). Since HCl is not present in MOZART gas-phase chemistry, displacement of $Cl^-$ from sea salt aerosols cannot be represented.

Chemical transport model predictions of particle acidity in the literature as well as in this paper (Fig. 18-19) are expressed as $pH_F$ and assume molarity- and molality-based concentrations lead to equivalent $pH_F$ with water as the solvent (Sect. 2.2 and Jia et al., 2018). CTMs, particularly those that use ISORROPIA or assume externally mixed inorganic and organic particles, assume the solvent for $H^+$ is water associated with inorganic electrolytes. In WRF-Chem with MOSAIC, species that do not

contribute directly to the ion balance (e.g., organics and inert mass) can absorb water, and thereby indirectly influence the $pH_F$ via solvent abundance. TM4-ECPL and CMAQ v5.3 can calculate $pH_F$ including solvent water associated with both inorganic and organic constituents, but in this paper, only water associated with electrolytes is considered in $pH_F$.

The limited literature to date evaluating CTM-predicted $pH_F$ indicates agreement between models and observationally-

constrained estimates within 1 $pH_F$ unit or better (summarized in Table S10). Observationally-constrained $pH_F$ from the eastern U.S. in summer at the surface (0.9 ± 0.6, Guo et al., 2015), summer aloft (1.1 ± 0.4, Xu et al., 2016) and winter aloft (0.8 ± 1.0, Guo et al., 2016) all indicate strongly acidic particles and are in good agreement with GEOS-Chem aloft predicted $pH_F$ of 1.3 during the SEAC4RS 2013 (Marais et al., 2016a) and WINTER 2015 (Shah et al., 2018) campaigns. CMAQ agreement with observations in the eastern U.S. are sensitive to assumptions regarding non-volatile cations with surface-level predictions

of $pH_F$ showing good agreement with observations in the work of Vasilakos et al. (2018) ($pH_F$ = 0.82) and when non-volatile cations were excluded in the work of Pye et al. (2018) ($pH_F$ = 0.9 ± 0.9). For Centreville, Alabama in the work by Vasilakos et al. (2018), CMAQ predicted excessively acidic aerosol during the day and similar to or higher than observationally-constrained $pH_F$ estimates by 1 unit at night. Reductions in non-volatile cations, which may be overpredicted due to errors in nocturnal mixing (Appel et al., 2013), reduced the nocturnal $pH_F$ in CMAQ making it more consistent with observations





(Vasilakos et al., 2018). $pH_F$ evaluation in other locations is more limited. Guo et al. (2017b) indicate a $pH_F$ for wintertime Beijing of 4.2, consistent with GEOS-Chem simulations of Beijing for Autumn/winter ($pH_F$=4.3, range: 3.6 to 5.0; Shao et al., 2019) and CMAQ ($pH_F$ of 4.5 ± 0.8 for Beijing February 2016, this work). Shao et al. (2019) found that including $Ca^{2+}$, $K^+$, and $Mg^{2+}$ from dust in the aerosol $pH_F$ calculations had a small effect on predicted aerosol $pH_F$ (increase of 0.1) in Beijing in autumn and winter, consistent with Guo et al. (2017b).

An evaluation of CTM-predicted $pH_F$ can be leveraged to understand the responsiveness of a model to changes in atmospheric composition. For example, Vasilakos et al. (2018) show that modeling $pH_F$ correctly in CMAQ is critical to accurately partition nitrate between the gas and aerosol-phase and thus capture trends in $PM_{2.5}$ nitrate as sulfate is reduced in the United States. Shah et al. (2018) provide insight into the effectiveness of past and future emission reductions by tracking $pH_F$ predicted by GEOS-Chem. Shah et al. (2018) predict that $pH_F$ for winter in the eastern U.S. increases from 0.39 to 1.7 between 2007 and 2023 using GEOS-Chem. As a result, nitrate aerosol concentrations are predicted to decrease less than the reductions of $NO_x$ emissions and total nitrate would imply. Since regulatory guidance for model application encourages the use of relative response factors (RRFs) by $PM_{2.5}$ component (EPA, 2018), a $pH_F$ evaluation can be particularly useful since a bias in $pH_F$ can result in a bias in gas–particle partitioning sensitivity. However, absolute abundances are also important in model applications, and thus a $pH_F$ evaluation complements evaluation against speciated $PM_{2.5}$ measurements from networks and intensive campaigns.

Figures 18 and 19 show results from four CTMs to give a sense of whether common spatial features exist among models predicting fine particle $pH_F$. Differences in fine particle $pH_F$ between models are likely caused by differences in model resolution as well as emission and meteorology scenarios as the ISORROPIA II and MESA thermodynamic models produce similar values (Sect. 6). $pH_F$ is strongly influenced by non-volatile cations in broad regions affected by dust and sea spray. Some of the least acidic fine aerosols ($4 < pH_F < 7$) are predicted in sea spray rich regions and where strong westerlies occur over the southern oceans just north of Antarctica. These $pH_F$ values are consistent with those of fresh sea spray shortly (minutes) after emission (pH ~5, Keene et al., 1998). Other areas over the ocean, for example, the northern Pacific south of Alaska, show more acidic aerosol, likely influenced by sulfur from shipping in combination with lower concentrations of sea-spray cations due to colder water temperatures and relatively low wind speed. For all models that include high latitudes in the domain, extremely acidic particles are predicted in the Arctic and over Greenland ($pH_F < 1$). No estimates of Arctic aerosol $pH_F$ are available in the literature, but several studies have inferred low amounts of ammonium and high acidity from proxies (Fisher et al., 2011; Croft et al., 2016), yet sources of $NH_3$ from seabirds may neutralize particle acidity in the Arctic (Wentworth et al., 2016). Other work has noted that sulfur dioxide is able to escape scavenging more effectively during lifting than ammonia or ammonium sulfate (Park et al., 2004) thus providing for long-range transport of acidity over alkalinity. The decrease in $pH_F$ (increase in acidity) due to scavenging also appears in the westerly outflow from China (Fig. 18), consistent





with the higher fraction of soluble iron found in particles collected in the region (Li et al., 2017). A similar pattern is also found in the easterly outflow region from central America (Fig. 18), although observational confirmation is still needed.

Fine mode $pH_F$ downwind of deserts varies by model but is between 4 and 6 for dust-dominant conditions in CMAQ and
GEOS-Chem. TM4-ECPL does not includes Ca and Mg from dust in the fine mode while it considers those NVCs in the coarse mode calculations of aerosol pH. This assumption leads to an aerosol $pH_F$ in TM4 over the Sahara of ~2-3 for the fine mode and ~6-7 for the coarse mode (Fig. 18c,d) further implying that dust cations, if present and internally mixed in the fine mode, can affect aerosol by ~4 $pH_F$ units (see also Fig. S7 for GEOS-Chem fine aerosol pH predictions with and without dust NVC). $pH_F$ ~6-7 fine aerosols are predicted in GEOS-Chem over the Sahara and Atlantic outflow of dust, most notably in the winter
and spring and to a lesser degree in the summer and fall. $SO_2$ emissions are non-zero in Saudi Arabia (Krotkov et al., 2016) leading to lower $pH_F$ for the middle East compared to other desert regions such as the Sahara.

Anthropogenically-dominated locations, such as Europe, Asia, and the United States show different aerosol $pH_F$ values but are universally predicted to be acidic. All the models in Fig. 18 show a gradient in $pH_F$ over Europe with locations in the northern
part of western Europe (near Germany) showing higher predicted fine aerosol pH ($pH_F$ ~2-3) compared to the Mediterranean Sea where $pH_F$ values can be less than 1 and approach 0. This gradient is consistent with enhanced ammonia in northern Europe (Clarisse et al., 2009) and the limited European pH data, which includes observationally-constrained estimates for Cabauw in the Netherlands ($pH_F$ ~3.6; Guo et al., 2018b), for aerosol extracts from Germany (pH ~1-2; Scheinhardt et al., 2013), and for Finokalia, Crete ($pH_F = 1.25 \pm 1.14$ excluding water associated with organics; Bougiatioti et al., 2016). Predicted aerosol $pH_F$
indicates moderate acidity ($pH_F$ ~3-4) for locations such as Beijing, China and northern India. The eastern U.S. is one of the more acidic anthropogenically dominated locations with $0 < pH_F < 4$ (Fig. 19), consistent with or slightly higher than observationally constrained estimates (Sect. 5.1.4). The fine horizontal resolution in WRF-Chem and CMAQ continental U.S. simulations captures localized increases in $pH_F$ due to ammonia from agricultural activity in eastern North Carolina, the Great Plains, Idaho's Snake River Valley, and California's San Joaquin Valley (Fig. 19a,d).

The MOSAIC aerosol model in WRF-Chem provides information on how predicted aerosol $pH_F$ varies among different-size aerosols with different composition (Fig. S8). The $PM_{2.5}$ aerosol $pH_F$ (aerosol water weighted average for bins 0.039–0.156, 0.156–0.625, and 0.625–2.5 μm in diameter, Fig. 19d) shows higher values in regions where NaCl aerosol dominates (off coast of California, over the Great Salt Lake), moderate $pH_F$ values over the Great Plains and off the East Coast, low $pH_F$ in the Ohio
River Valley (due to large $SO_x$ emissions and sulfate formation), and the lowest $pH_F$ in the Southwest U.S. (where aerosol water is low). CMAQ (Fig. 19a) predicts similar spatial trends over the U.S. for the fine aerosol $pH_F$ (Aitken + accumulation modes), but generally predicts less spatial heterogeneity with more acidic particles (by ~1-2 $pH_F$ units) over the midwest, and slightly less acidic particles (by ~1 $pH_F$) in the southwest. GEOS-Chem and WRF-Chem exhibit differences over Nevada of 5 $pH_F$ units during summer. WRF-Chem indicates more acidic particles in the submicron range compared to $PM_{2.5}$. The most



notable differences between $PM_{2.5}$ (liquid water weighted sum over bins 1-3) and submicron $pH_F$ (liquid water weighted sum over bins 1-2) in WRF-Chem occur over the oceans, Gulf of Mexico, and Gulf of California where predicted $pH_F > 5.6$ for the larger fine (0.6 to 2.5 μm) bin. Similar values are not seen in the smaller aerosol bins except for one plume of $pH_F > 5.6$ (in bin 2, 0.156–0.625 μm). The two smallest aerosol size bins have very similar pH values over the continent, while the largest fine

bin (bin 3) has similar pH values over the continent well inland from the coast and higher pH values near the coasts over land. Similarly TM4-ECPL shows differences of about 4 $pH_F$ units between fine (submicron) and coarse mode particles from Arizona and Montana (Fig. 18c,d). Since different sources contribute differently across the size range, heterogeneity in size resolved $pH_F$ predictions also implies mixing state assumptions in bulk schemes affect $pH_F$ estimates and a single $pH_F$ value across a broad size range does not capture the range of states present in the atmosphere (see also Sect. 7 for a discussion on

mixing state).

Coarse mode aerosol $pH_F$ in the WRF-Chem MOSAIC (aerosol size bin 4, 2.5–10 μm size range, Fig. 19e) is generally higher than $PM_{2.5}$ values (Fig. 19d), especially over the oceans where NaCl dominates (note that WRF-Chem v4.1 and earlier does not include HCl thereby producing higher aerosol $pH_F$ over oceans than expected). In the more arid regions of the southwest

U.S. and northern Mexico, coarse mode aerosol $pH_F$ is quite acidic, while elsewhere over the conterminous U.S. coarse mode aerosol $pH_F$ ranges from 1 to 6, with low values over the Ohio River Valley. Coarse mode aerosol $pH_F$ in TM4-ECPL (Fig. 18d) shows values near 4-6 over the central-western U.S. and Canada and $pH_F < 2$ in the eastern U.S. Coarse mode aerosol $pH_F$ has similarly low values in other anthropogenic regions (part of Europe, India, and East Asia) as well as southern Africa, Indonesia, and most of South America. Over the oceans and remote regions, coarse mode aerosol $pH_F$ predicted by TM4-ECPL

has a value of 6 or greater.

## 8.2   Cloud pH

Compared to aerosol pH, cloud pH calculations have a longer history in CTMs and climate models, given that sulfate is the dominant secondary pollutant in fine particulate matter that is produced primarily in cloud droplets. Recent work has shown

that when cloud pH increases from a combination of $SO_x$ emission controls and increasing $NH_3$ from intensified agriculture, the efficiency at which $SO_2$ converts to sulfate via in-cloud $O_3$ oxidation can increase thus reducing the effectiveness of $SO_x$ controls close to emission regions (Paulot et al., 2017).

Compared to CTMs, climate models are more limited in their treatment of semi-volatile inorganic species (e.g., particulate

nitrate may not be considered as in CAM5; Liu et al., 2012), but generally include sea salt and dust (even if assumed inert) as well as sulfate. Modeling studies of acid rain with regional models (Carmichael and Peters, 1986; Chang et al., 1987; Venkatram et al., 1988) and sulfate production in climate models (Barth et al., 2000; Feichter et al., 1996; Koch et al., 1999) include pH since pH dictates the rate of aqueous reactions that convert $SO_2$ to sulfate (Sect. 3). Early global model studies





(Barth et al., 2000; Feichter et al., 1996; Park et al., 2004) either prescribed cloud pH, or diagnosed pH from the concentration of cloud-water S(IV) / S(VI) and assumed ammonium-to-sulfate ratio. Even recent studies (e.g., Turnock et al., 2019) may prescribe cloud pH to simulate sulfate production. In most CTMs, the calculation of cloud $pH_F$ is more comprehensive. Five such models are described here and $pH_F$ estimates are presented in Fig. 20-21. Chemical composition tracked for cloud

chemistry can be the same as or different from that used in aerosol chemistry.

CESM2.0, including CAM6-chem (released in 2018, Fig. 20a, 21a), includes an updated tropospheric chemistry mechanism (MOZART-T1) and represents aerosols using a Modal Aerosol Model (MAM; Liu et al., 2012; Liu et al., 2016) with 4 lognormal modes and including the species sulfate, ammonium, primary and secondary organic matter, black carbon, soil dust,

and sea salt. MAM considers the thermodynamic partitioning of $H_2SO_4$ (gas) and $NH_3$. The MAM scheme does include cloud chemistry that represents S(IV) oxidation by ozone and hydrogen peroxide to form sulfate and non-reactive uptake of $HNO_3$ and $NH_3$. The pH is estimated using an iterative method to solve the electroneutrality equation using $OH^-$, $HCO_3^-$, $NO_3^-$, $HSO_3^-$, $SO_3^{2-}$, $SO_4^{2-}$, and $NH_4^+$. The $pH_F$ is determined at each chemistry time step and grid point where liquid cloud water exists.

In CMAQ, there are two varieties of the cloud chemistry module: the default cloud chemistry routine that assumes instantaneous equilibrium to describe the distribution of species between gas/aqueous/ionic forms and the routines that include kinetic mass transfer (KMT) considerations. CMAQ's default cloud chemistry module is based on the work of Walcek and Taylor (1986). $pH_F$ (Fig. 20b, 21b) is estimated throughout the course of the chemistry calculations by solving the system of nonlinear algebraic equations resulting from electroneutrality and ionic/Henry's Law equilibrium assumptions. Activity

coefficients, estimated with the Davies equation, are applied to ionic species in solution. For the standard chemical mechanism (i.e., five S(IV) oxidation reactions and two SOA reactions), the following species are considered in the ion balance and ionic strength calculations: $H^+$, $OH^-$, $HSO_3^-$, $SO_3^{2-}$, $HSO_4^-$, $SO_4^{2-}$, $HCO_3^-$, $CO_3^{2-}$, $HCO_2^-$, $NH_4^+$, $NO_3^-$, $Cl^-$, $Ca^{2+}$, $Na^+$, $K^+$, $Mg^{2+}$. $Fe^{3+}$ and $Mn^{2+}$, potentially important players in catalyzing aqueous S oxidation, are included in the ionic strength calculation but do not impact droplet $pH_F$ as they are assumed to be associated with generic anions, $A^-$ and $B^-$. $pH_F$ evolves as S(IV) is oxidized

to S(VI) and additional species are scavenged from interstitial aerosol, allowing species to redistribute between phases and different (non)ionic forms for the duration of cloud processing. In CMAQ's KMT family of cloud chemistry modules, individual species/ions are tracked, including $[H^+]$, and evolve dynamically, using forward and reverse reactions to represent ionic equilibria (Fahey et al., 2017). Initial $pH_F$ is estimated from known concentrations of activated aerosol species (i.e., all accumulation and coarse mode species), and electroneutrality.

Bulk cloud $pH_F$ calculations were first implemented into GEOS-Chem as described in Alexander et al. (2012). Prior to this, cloud pH was assumed to equal 4.5 in GEOS-Chem. In GEOS-Chem (version 12.0.0 with MERRA-2 reanalysis) (Fig. 20c, 21c), bulk cloud $pH_F$ is calculated using local concentrations of $SO_4^{2-}$, $SO_2(g)$, $NH_3(g)$, $NH_4^+$, $HNO_3(g)$, $NO_3^-$, and $CO_2(g)$. The cloud water $pH_F$ calculation utilizes the electroneutrality equation and the following forms of dissolved species (in moles





L⁻¹): $SO_4^{2-}$, $OH^-$, $HCO_3^-$, $CO_3^{2-}$, $HSO_3^-$, $SO_3^{2-}$, $NO_3^-$, $NH_4^+$. The concentration of $HSO_4^-$ is assumed negligible, which is valid given that most $pH_F$ values > 3 where the second dissociation of sulfuric acid is virtually complete. The model assumes a cloud mass scavenging efficiency of 0.7 for $SO_4^{2-}$, $NO_3^-$, and $NH_4^+$ aerosol based on observations (Hegg and Hobbs, 1986; Hegg et al., 1984; Schumann, 1991; Sellegri et al., 2003). The concentrations of all species but $SO_4^{2-}$ in the electroneutrality equation

are calculated based on cloud liquid water content, temperature, each species' effective Henry's law constants. For example, $NO_3^-$ is calculated as follows:

$$[NO_3^-] = \left(\frac{K_H K_a}{[H^+]}\right) p_{HNO_3} \tag{20}$$

where activities are approximated as aqueous concentrations, $K_H$ is the Henry's law constant (Reaction R1) for $HNO_3$, $K_a$ is the dissociation constant (Reaction R2) for $HNO_3$, and $p_{HNO_3}$ is the partial pressure of $HNO_3$(g). The resulting cubic equation

is solved numerically. GEOS-Chem does not account for the effect of organic acids or cations originating from sea salt or dust (e.g., $Na^+$, $Ca^{2+}$) on cloud water $pH_F$. HCl is also not part of the cloud $pH_F$ calculation because HCl is not yet a transported species in the standard version of the model (version 12.0.0). This can easily be implemented into the cloud $pH_F$ calculation when the chlorine chemistry in GEOS-Chem is updated (Wang et al., 2019b).

Cloud $pH_F$ in GEOS-Chem (Fig. 20c, 21c), as well as other models, is utilized for the calculation of sulfate production rates from in-cloud oxidation of $SO_2$(g). Bulk schemes, such as those described above, tend to underestimate sulfate production compared to calculations accounting for heterogeneity in pH with drop size (see Sect. 5.2.3 for a discussion of drivers of heterogeneity). To account for this bias, GEOS-Chem utilizes parameterizations developed by Fahey and Pandis (2001) and Yuen et al. (1996). The Fahey and Pandis (2001) parameterization is a decision algorithm that determines whether or not cloud

droplet heterogeneity will impact sulfate production rates. The impact of cloud droplet heterogeneity on sulfate production rates tends to be most prevalent in the presence of alkaline aerosols such as sea salt (Alexander et al., 2012; Fahey and Pandis, 2001). If cloud water is acidic enough, heterogeneity will not matter. The Fahey and Pandis (2001) algorithm considered this effect and identifies a condition where bulk cloud $pH_F$ will underestimate sulfate production rates. GEOS-Chem corrects for this low bias in the sulfate production rate utilizing the Yuen et al. (1996) parameterization, which was developed by comparing

calculated sulfate production rates from a bulk cloud model with a cloud resolving model that accounts for cloud droplet size heterogeneity. Since the Yuen et al. (1996) parameterization was developed for warm clouds, its use is restricted to temperatures above 268 K in GEOS-Chem. Additionally, the Yuen et al. (1996) parametrization is only used in GEOS-Chem over the oceans, because the parameterization considers alkalinity typical of sea salt aerosols. The impact of cloud droplet heterogeneity on sulfate production rates was implemented into GEOS-Chem by Alexander et al. (2012).

In TM4-ECPL (Fig. 20d, 21d), in-cloud $pH_F$ is controlled by strong acids ($SO_4^{2-}$, methanesulfonate, $HNO_3$, $NO_3^-$), bases (ammonium ion, $NH_4^+$), as well as by the dissociations of hydrated $CO_2$, $SO_2$, $NH_3$ and of oxalic acid (Myriokefalitakis et al., 2011). Cloud droplet heterogeneity and dust and sea salt aerosol components are not considered for cloud $pH_F$ calculations.



The cloud chemistry configured with the WRF-Chem v3.9.1 with MOZART gas chemistry and 4-bin MOSAIC aerosol scheme is a bulk cloud water approach that is subsequently partitioned into the 4 cloud water bins (which connect to the 4 aerosol size bins). The Fahey and Pandis (2001) aqueous chemistry scheme is implemented and calculates sulfate formation as well as

formaldehyde oxidation and non-reactive uptake of nitric acid, hydrochloric acid, ammonia, and other trace gases. The $pH_F$ is found using a bisection method to solve the electroneutrality equation, which includes the following species: $OH^-$, $HCO_3^-$, $CO_3^{2-}$, $CO_3^-$, $HSO_3^-$, $SO_3^{2-}$, $HSO_4^-$, $SO_4^{2-}$, $SO_4^-$, $SO_5^-$, $HSO_5^-$, $HOCH_2SO_3^-$, $^-OCH_2SO_3^-$, $NO_2^-$, $NO_3^-$, $HO_2^-$, $O_2^-$, $HCOO^-$, $Cl^-$, $Cl_2^-$, $ClOH^-$, and $NH_4^+$. While trace metal ion chemistry is included in the aqueous-phase formation of sulfate, these metals are not part of the $pH_F$ calculation. The $pH_F$ is determined at each chemistry time step and grid point where liquid cloud water exists

(Fig. 21e).

Model predicted cloud-droplet $pH_F$ (Fig. 20-21) reflects atmospheric sources of inorganic species, similar to fine mode aerosol $pH_F$, but is further modulated by the presence of clouds and abundance of condensed water. Since cloud droplets are more dilute than particles, $pH_F$ is generally higher than for fine aerosol. The southern ocean clouds have $pH_F$ 4.5-6 with TM4-ECPL

showing more acidic cloud droplets ($pH_F$ = 4.5) in the southern oceans compared to GEOS-Chem and CAM-Chem ($pH_F$ ~5-6) due to lack of sea spray and dust aerosol components in TM4-ECPL cloud $pH_F$ calculations. Cloud droplet $pH_F$ is often greater over oceans than continents at the same latitude. GEOS-Chem and CAM-Chem show slightly different north-south trends in cloud pH over the Southern Ocean with GEOS-Chem indicating clouds decrease in acidity from Antarctica to the equator and CAM-Chem indicating increasing acidity. One cloudwater pH measurement gives a value about 5 for off coast of

Chile/Peru (Fig. 7), but more measurements are needed, particularly in the southern hemisphere. Note that CAM-Chem, GEOS-Chem, TM4-ECPL, and WRF-Chem do not include dust cations in the cloud pH calculations (CMAQ does include fine and coarse dust) and deserts and their downwind areas such as the Sahara and western U.S. (e.g., Hand et al., 2017) show diversity on the order of 3-4 $pH_F$ units among the models. Measured cloud pH over northern Africa is 6-7 (Fig. 7), consistent with CMAQ. TM4-ECPL, which does not consider dust cations in cloud pH calculations, predicts $pH_F$ < 4 cloud droplets in arid

regions such as over the Sahara, southwest U.S., and inland Asia where liquid cloud water may be very low. Dust regions also coincide with limited cloud coverage so aqueous chemistry is less important. Several models (CMAQ, CAM-Chem, GEOS-Chem, WRF-Chem, TM4-ECPL) correctly capture locally enhanced acidity for the cloud droplets in the vicinity of the Ohio River Valley in the eastern United States as well as in upstate New York at Whiteface Mountain where pH is 4.5-5.0 since 2010 (Table S11). The $pH_F$ gradient from northern to southern Europe is reversed for cloud water compared to aerosols with

Germany and Poland showing more acidic cloud droplets than over Italy and Spain (GEOS-Chem, CMAQ, TM4-ECPL). For both clouds and particles, aerosol $pH_F$ is higher in northern China (e.g., Beijing) compared to southern China (e.g., the Pearl River Delta) except in TM4-ECPL. CAM-Chem, CMAQ, GEOS-Chem, and WRF-Chem do not predict present-day average cloud droplet $pH_F$ below 3 (which is not strictly the lower limit in observed cloud pH, Fig. 7). For select locations and models (Table S11), predicted cloud pH was generally within 2 pH units of observations and often showed better agreement.





### 8.3 Recommendations for improving models

Evaluation of CTM predictions of fine aerosol $pH_F$ in literature (Table S10) suggest reasonable agreement between models and observations ($pH_F$ within 1 unit for fine aerosols). However, observed estimates of $pH_F$ (Sect. 5.1) are extremely limited in location and do not fully cover the diversity of environments and values covered by CTM predictions. Furthermore, the models that have been most evaluated (e.g., CMAQ, GEOS-Chem) tend to use a relatively complete set of inorganic species and advanced thermodynamic routines such as ISORROPIA II but this may not reflect the entire CTM or climate model community (e.g., CAM-Chem). CTM predictions here indicate that assumptions regarding non-volatile cations, from both dust and sea salt, play a large role in CTM predictions of $pH_F$. Prior to the inclusion of Ca, K, and Mg ions from dust in GEOS-Chem aerosol $pH_F$ calculations (this work, Fig. S7), CMAQ and GEOS-Chem showed large $pH_F$ differences (multiple $pH_F$ units) in dust outflow regions. Similarly, cloud droplet $pH_F$ predictions varied by up to 3-4 $pH_F$ units between models in non-volatile cation-rich environments. Despite cloud droplets generally occupying a smaller range in $pH_F$ than particles (Fig. 2), models examined here provide no indication that cloud droplet $pH_F$ is predicted more consistently across models than aerosol $pH_F$. Spatial and temporal variability in clouds, which are challenging to predict and represented differently across models, could contribute to some of this model variability. (Note that for 10 of the 11 cases where select models were compared to observations, models tended to systematically underestimate of overestimate rather than bracket the observed cloud $pH_F$ value, Table S11). Model-to-model differences in many locations amount to multiple $pH_F$ units and observational constraints would be needed to evaluate models. In addition to measuring inorganic aerosol constituents and gas-phase semi-volatiles to perform thermodynamic calculations, particle mixing state, and cloud properties may also need to be characterized with measurements. Remote locations (including over oceans, most of the southern hemisphere, and likely aloft) are locations with diverse CTM predictions of cloud and particle $pH_F$ and are particularly lacking observational constraints of $pH_F$.

## 9 Conclusions

Aerosol and cloud acidity are key drivers of atmospheric chemistry and processes that link emissions to impacts on air quality, human health, ecosystems, and climate. Despite their importance, limited information exists on the spatiotemporal distribution of atmospheric acidity, its drivers, and its influences. For aerosol acidity, only recently have data become available that can be used for model evaluation and improvement. This review aims to provide a comprehensive overview of the state of knowledge of atmospheric acidity, considering particulate matter as well as clouds and fogs. Apart from a review of the published literature, the study also includes a rigorous set of definitions for acidity, the methods used to measure and infer the in-situ levels of acidity in each condensed-phase type, and a synthesis and critical evaluation of current estimates. Across the review, the following major messages emerge (see the sections listed for more discussion):



- The various pH definitions in use for characterizing aerosol and cloud-water acidity differ in important ways from each other and from the definition of pH by IUPAC, which is based on the negative base-10 logarithm of the molal activity of $H^+$. A nomenclature is provided for the community to document how different studies calculate and express aerosol acidity. The use of the definition of pH by IUPAC (Eq. 1), involving the activity coefficient of $H^+$, is recommended to best and consistently quantify the pH. (Sect. 2.)

- While thermodynamic partitioning and ionic equilibria are the dominant factors that drive aerosol pH levels, models are frequently lacking $H^+$ that is kinetically generated as a result of transient gas- and liquid-phase chemical reactions. The representation of kinetic processes is necessary to determine sulfate levels important for driving pH. The consequences of varying acidity for organic particle and cloud chemistry have only started to be investigated. (Sect. 3 and companion paper *in prep*.)

- Methods for measuring cloud-water pH are relatively established, but methods for measuring aerosol pH remain challenging. Measuring aerosol pH is difficult due to the extremely high ionic strengths that are typically found in aqueous aerosol, the low amounts of mass, and the extreme sensitivity to environmental perturbations, as well as the chemical heterogeneity found in particles across size, location, and time. Methods for determining the pH of bulk aerosol samples and individual particles continue to be developed and will address an important measurement gap that still exists for determining the acidity of aerosol. Particularly important is the application of such methods to understand the pH environment from particle to particle and within particles, especially under conditions where the aerosol is not at equilibrium or not internally mixed. (Sect. 2, 5, 7.)

- None of the observationally-based aerosol acidity proxies in use today are suitable as a universal indicator of pH. Under certain conditions (strongly acidic conditions), certain proxies may be of limited use and when combined with gas-phase measurements exhibit some correlation with pH. However, the uncertainty of these proxies remains very large, and even the best ones require verification with models. The best estimates of particle pH are obtained from thermodynamic model calculations when gas–particle partitioning observations are available for evaluation as well as for constraining the calculations. These estimates generally require a thermodynamic equilibrium assumption, which is reasonable for submicron aerosol. Direct measurement of $NH_3$ is extremely important, since combined with aerosol data, it provides a constraint on model-derived acidity estimates and a metric for evaluation. Other semi-volatile inorganic acidic gases (like $HNO_3$ and $HCl$) also provide constraints on acidity but are subject to higher uncertainty due to interaction with the coarse mode. (Sect. 4 and elsewhere.)

- Different box model-based estimates of pH using the same inputs differ on average by 0.3 pH units (but can vary up to one pH unit, increasing with decreasing RH) depending on the model framework used and the approach for estimating the $H^+$ activity coefficient. When single-ion activity coefficients are unavailable, an approximation based





on the mean molal activity coefficient of a relevant ion pair (e.g., $\gamma_\pm(H^+,Cl^-)$ yielding a $pH_\pm$) can reduce the bias in acidity by up to 0.43 pH units for atmospherically relevant conditions. The ion pair that leads to the best single-ion activity coefficient for $H^+$ may be model dependent; for example, $\gamma_\pm(H^+,NO_3^-)$ yields the best $pH_\pm$ from ISORROPIA II. (Sect. 6.)

•   The limited observationally-constrained pH estimates to date establish that acidic aerosol is ubiquitous and can be extremely acidic (pH as low as -1 averaged over long timescales and episodically even lower). Aerosol pH depends on the size, composition, and mixing state of particles. Fine mode aerosols are often dominated by ammonium, sulfate, nitrate, and organics and are systematically more acidic (up to 5 pH units) than coarse mode aerosols, which are rich in non-volatile cations originating from sea salt and dust. Most observationally-constrained estimates of particle

acidity to date are the approximation, $pH_F$. Since the accuracy of $pH_F$ as a measure of pH depends on RH and composition (Sect. 6), characterization of ambient pH is incomplete. (Sect. 5.1.)

     •   Although aerosols and clouds both tend to be acidic, the response of acidity to changes in precursor emissions is distinctly different in the two media. Published studies suggest that reductions in sulfur dioxide and nitrogen oxide emissions across the U.S. and Canada have had little impact on aerosol pH, and pH is relatively insensitive to ammonia

changes. Conversely, clouds and fogs exhibit a broad pH range that is quite sensitive to the relative abundance of sulfuric and nitric acids and ammonia with multiple locations showing increases in cloud pH as anthropogenic emissions are controlled. This is a direct consequence of the liquid water content and other aerosol species being in equilibrium with the ambient relative humidity – while in clouds all the species can vary independently of each other. (Sect. 5.)

•   Large-scale model variation in predicted $pH_F$, up to 5 pH units in specific locations, is likely not driven by the thermodynamic representations in models, but by the composition that feeds the thermodynamic calculations (especially the emission and microphysical interactions of non-volatile cations with other aerosol components). For locations with observationally-constrained pH estimates, agreement between models and observations can be within one pH unit. In addition, the global acidity distribution in models and observations can be surprisingly similar (Figure

2). Cloud pH does not seem to be better constrained than aerosol pH, suggesting that there is considerable work to be done refining simulations to reach agreement with observational values and trends. Spatial gradients in CTM pH predictions (that do not coincide with availability of measurements) suggest regions where future measurements should be made. The level of agreement required between models and observations depends on the target of a specific assessment (e.g., PM sensitivity to emissions, nutrient deposition, metals solubility). Therefore, model frameworks

should evaluate their endpoint of interest (e.g., nutrient deposition, $PM_{2.5}$ concentration) and consider how an error in predicted pH could lead to a bias. The error in pH may be important for some applications, but not others. (Sect. 8.)





Perhaps one of the more important outcomes of this review is the recognition that cloud and aerosol pH emerge as an important property for influencing a wide range of CTM predictions, and therefore improvements to how aerosol and cloud pH are represented in CTMs could potentially enhance policy and programs informed by these models. pH determines the innate response of a model to emission changes, can provide insights that established approaches (evaluation of gas/aerosol composition) are not able to provide, and determines the chemical regime for PM formation, nutrient deposition, and soluble metals. Including and reporting pH (or an approximation thereof) in future studies will increase the understanding of the effects of emissions, human activity, and climate change on society and the Earth System as a whole.

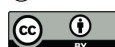

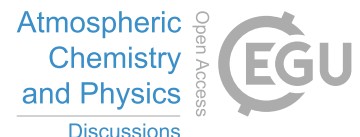
## Appendix A: Nomenclature

| Symbol | Description |
|---|---|
| $a_{H^+}$ | The activity of hydrogen ions in aqueous solution on molality basis |
| $a_{H^+}^{(c)}$ | The activity of $H^+$ ions on a molarity basis |
| $a_{H^+}^{(x)}$ | The activity of $H^+$ ions on a mole fraction basis |
| AMS | Aerosol mass spectrometer |
| $c^{\ominus}$ | The standard state (unit) molarity |
| $c_{H^+}$ | The molarity or "molar concentration" of hydrogen ions in an aqueous solution (also written using square brackets as $[H^+]$) |
| CTM | Chemical transport model |
| FR | Flex Ratio, identifies the $NH_3$ emissions level at which the nitrate concentration switches from $NH_3$-insensitive (or negative sensitivity) to positive $NH_3$ sensitivity |
| $f_{H^+}^*$ | The (rational) activity coefficient based on the mole fraction concentration scale |
| $H^+_{air}$ | Concentration of aerosol $H^+$ per volume of air (e.g., moles per $m^{-3}$ of air) |
| $K_w$ | Activity-based equilibrium constant for the dissociation of water into $H^+$ and $OH^-$ |
| LLPS | Liquid–liquid phase separation |
| KMT | Kinetic mass transfer |
| $m^{\ominus}$ | The standard state (unit) molality |
| $m_{H^+}$ | Molality of $H^+$ (mol kg$^{-1}$) |
| $M_w$ | Molar mass of water: 0.018015 kg mol$^{-1}$ |
| $n_i$ | Number (e.g., moles) of species $i$ |
| PAHs | polycyclic aromatic hydrocarbons |
| PFASs | polyfluoroalkyl substances |
| PFSAs | perfluroroalkane sulfonic acid |
| PFCAs | Perfluoroalkyl carboxylic acids |
| pH | Hydrogen ion potential with activity coefficient and concentration expressed on a molality concentration scale (see Table 1) |
| $pH_c$ | pH on a concentration (molarity) basis |
| $pH_x$ | pH on a mole fraction basis |
| $pH_T$ | "Total" pH based on the molality of sulfate and bisulfate ions (see Table 1) |
| $pH_F$ | "Free ion" approximation of pH obtained when the activity coefficient of $H^+$ is unity (see Table 1) |
| $pH_{\pm}$ | Approximation of pH using the mean molal ion activity coefficient of an $H^+$ and anion X pair (see Table 1) |
| $pK_w$ | $-\log_{10}$ of $K_w$ (see Bandura and Lvov (2005) for tabulation of values) |



| | |
|---|---|
| PM$_{2.5}$ | Particulate mass with an aerodynamic equivalent diameter below 2.5 μm |
| RH | Relative humidity |
| TMI | Transition Metal Ions |
| TCl | Total chloride (sum of gas-phase hydrochloric acid and aerosol chloride) |
| TNO$_3$ | Total nitrate (sum of gas-phase nitric acid and particulate nitrate) |
| TNH$_4$ | Total ammonia (sum of gas-phase ammonia and particulate ammonium) |
| VOCs | Volatile organic compounds |
| WSOC | Water-soluble organic compounds |
| $x_{H^+}$ | The mole fraction of H$^+$ in the solution |
| $\gamma_{H^+}$ | The molal activity coefficient |
| $\gamma_{H^+}^{(c)}$ | The molarity-based activity coefficient |
| $\gamma_{\pm,HX}$ | Single ion activity coefficient (for monovalent acid HX) |
| $\rho_0$ | The density of the reference solvent (water) |



*Code and data availability.*
E-AIM can be run at http://www.aim.env.uea.ac.uk/aim/.

AIOMFAC can be run at https://aiomfac.lab.mcgill.ca for single liquid phases; code is available at

https://github.com/andizuend/AIOMFAC.

MOSAIC is available upon request from its author (Rahul Zaveri).

ISORROPIAII is available at http://isorropia.epfl.ch (access to source code requires login provided by AN upon request).

EQUISOLV II output was obtained from its creator, Mark Jacobson (jacobson@stanford.edu).

CAM6-Chem code is available as part of CESM available at http://github.com/ESCOMP/cesm.

CMAQ v5.2 and v5.3 code is available at https://github.com/USEPA/CMAQ and at doi:10.5281/zenodo.107987.

GEOS-Chem code is available at https://github.com/geoschem.

TM4-ECPL code is available from its authors (Stelios Myriokefalitakis and Maria Kanakidou) upon request

Instructions for obtaining WRF-Chem are available at http://www2.mmm.ucar.edu/wrf/users/download/get_source.html.

Box model inputs used in Sect. 6 and observed cloud and fine aerosol pH estimates from literature will be deposited in

electronic tabular format in the Environmental Protection Agency Science Hub repository upon final publication

(https://catalog.data.gov/harvest/about/epasciencehub, doi:10.23719/1504059).

*Supplement.* The supplement related to this article is available online and includes additional documentation for definitions of

pH, methods used to estimate sulfur production as a function of pH (Fig. 3), and details regarding the proxy evaluation (Fig.

15). In addition, figures further exploring the gas ratio, suitability of pH approximations, an ISORROPIA–MOSIAC

intercomparison, and additional CTM predictions are shown. Data used as box model input, to create spatial maps of particle

and cloud pH, and for CTM-observation comparisons of pH are available in the supplement.

*Author contributions.* HOTP provided overall project coordination including preparation and finalization of synthesized drafts.

HOTP and AN designed the overall scope of this study. AN coordinated the supplement. JW and JK led Sect. 1 on the

importance of acidity with contributions from AN. AZ led Sect. 2 on definitions of pH. VFM led Sect. 3 on the role of kinetics

and mechanisms of pH. CH led Sect. 4 on proxies of pH. MK and AN led Sect. 5.1 on observations of atmospheric particle

pH. JC led Sect. 5.2 on observations of cloud water pH. AN led Sect. 6 on the box model intercomparison. Section 6 contains

significant portions of text originally created by the Sect. 2 (idealized scenario calculations) and Sect. 4 (proxy calculations)

teams. RAZ led Sect. 7 on the role of particle size and mixing state. HOTP led Sect. 8 on large-scale model predictions of pH.

Major messages (Sect. 9) written by AN, were created at a workshop organized by HOTP and hosted at EPA in Research

Triangle Park. Authors prepared text, figures, and tables in collaboration.

*Competing interests.* The authors declare that they have no conflict of interest.



*Disclaimer.* The U.S. Environmental Protection Agency through its Office of Research and Development collaborated in the research described here. The research has been subjected to Agency administrative review and approved for publication but may not necessarily reflect official Agency policy. The views expressed in this article are those of the authors and do not

5   necessarily represent the views or policies of the U.S. Environmental Protection Agency.

*Acknowledgements.* We thank the EPA for funding and hosting the workshop *The State of Acidity in the Atmosphere: Particles and Clouds* and Ken Elstein, Brooke Hemming, and Randa Boykin for their assistance during the workshop. We are grateful to Mark Z. Jacobson for participating in the model intercomparison (section 6) and for providing the results of EQUISOLV II

10  calculations. We thank Barron Henderson for assistance with CMAQ plots, Homaira Sharif for assistance with reference formatting in the main text, and Bo Xu for help assembling the appendix. We thank Barron Henderson and Sharon Phillips for their technical leadership on EPA contracts resulting in CMAQv5.2 output for hemispheric and continental US simulations, respectively. AN was supported by the project PyroTRACH (ERC-2016-COG) funded by H2020-EU.1.1. – Excellent Science – European Research Council (ERC), project ID 726165. The work by JC and ITK was supported by grant number NSF-AGS-

15  1650786. CJH acknowledges support from the National Science Foundation through project CHE-1454763. NR acknowledges support from NSF AGS-1254428. AZ acknowledges support by the Natural Sciences and Engineering Research Council of Canada (NSERC, RGPIN/04315-2014). TW acknowledges support by the Hong Kong Research Grants Council (T24-504/17-N). RAZ acknowledges support from the Office of Science of the U.S. Department of Energy as part of the Atmospheric System Research program (DE-AC05-76RL01830).





**Figures**



**Figure 1.** Sources and receptors of aerosol and cloud droplet acidity. Major primary sources and occurrence in the atmosphere are identified in bold red text: sea salt, dust, biomass burning (sources); and aerosols, fog droplets, cloud droplets, precipitation (occurrence). Key aerosol processes are indicated by arrows and grey text: nucleation/growth, light scattering, CCN and IN activation, and gas–particle partitioning. The aerosol sinks (wet, dry and occult deposition) are indicated by blue lines and text. The effects that aerosols have in the atmosphere, and on terrestrial and marine ecosystems and human health, are highlighted in yellow. Approximate pH ranges of aqueous aerosols and droplets, seawater, and terrestrial surface waters are also given.





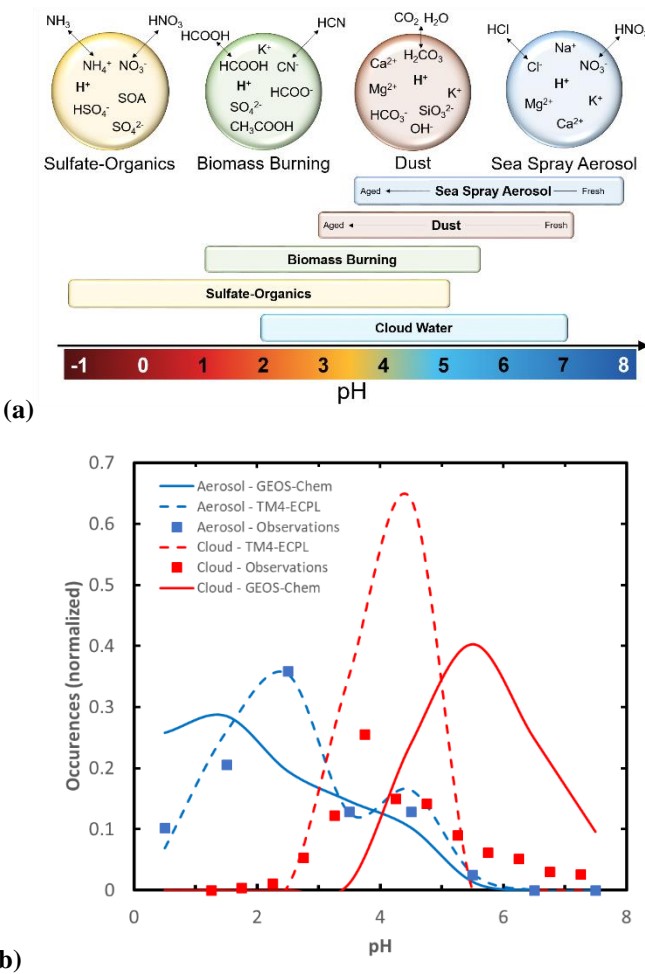

**Figure 2.** Characteristics of (a) aerosol and cloud pH drivers and (b) fine-mode aerosol (ground level) and cloudwater pH
5    (column average weighted by liquid water content) from observations (Sect. 5) and global simulations (Sect. 8).





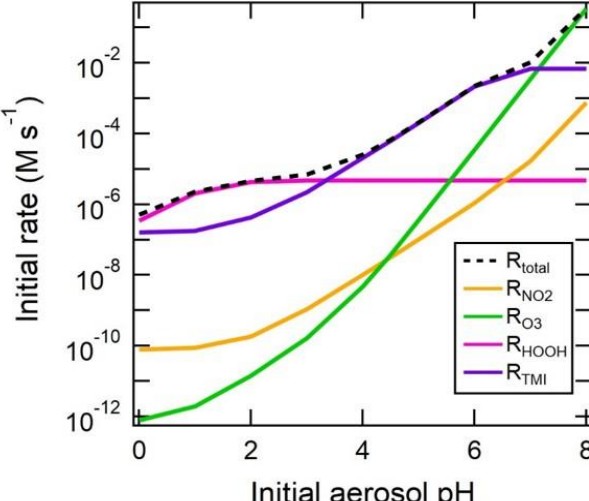

**Figure 3.** The rates of four in-particle sulfate production pathways (oxidation by $O_3$, $NO_3$, and HOOH and catalyzed by transition metal ions (TMI) Fe(III) and Mn(II)) as a function of initial aerosol pH, for Beijing winter haze conditions. See Supporting Information Sect. S2 for more details.





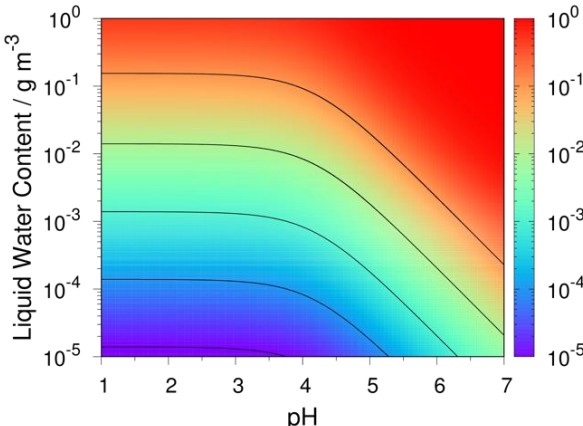

**Figure 4.** Calculated dissolved aqueous-phase fraction of benzoic acid as a function of liquid water content and aqueous phase acidity. The black lines are the isolines of the aqueous fractions of $10^{-i}$ (i = 1,...,6).



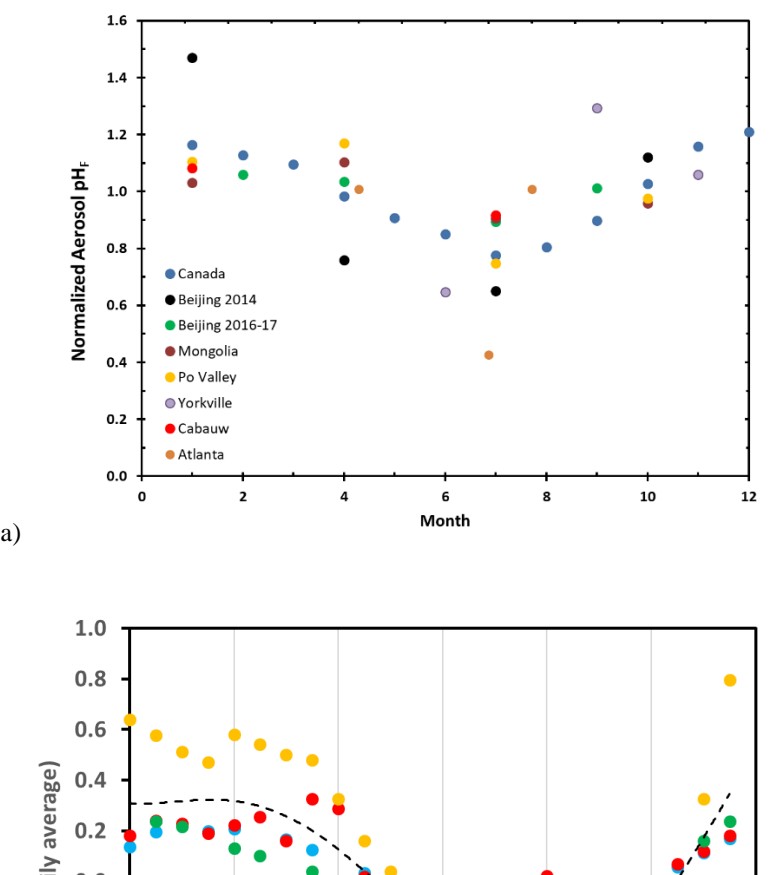

**Figure 5.** (a) Seasonality of fine aerosol acidity ($pH_F$ normalized to the annual average) from the literature. Annual average pH for Canada (Tao and Murphy, 2019b), Beijing 2014 (Tan et al., 2018), Beijing 2016-2017 (Ding et al., 2019), Inner Mongolia (Wang et al., 2019a), Po Valley (Squizzato et al., 2013), Cabauw (Guo et al., 2018b), Atlanta (Guo et al., 2015), and Yorkville (Guo et al., 2015; Nah et al., 2018) is 2.5, 2.8, 4.3, 5.5, 3.1, 3.6, 1.3, 1.7, respectively. (b) Diurnal cycle of aerosol $pH_F$ at four select sites (see Sect. 5 and Table S6 for a description of data), expressed as the departure from its daily mean value (3.7, 0.6, 2.1, 3.2, for Cabauw (Guo et al., 2018b), SOAS-Centreville (Guo et al., 2015), CalNex-Pasadena (Guo et al., 2017b), and Tianjin (Shi et al., 2017), respectively). Estimates are $pH_F$ except for Tao and Murphy (2019b) who estimate pH.



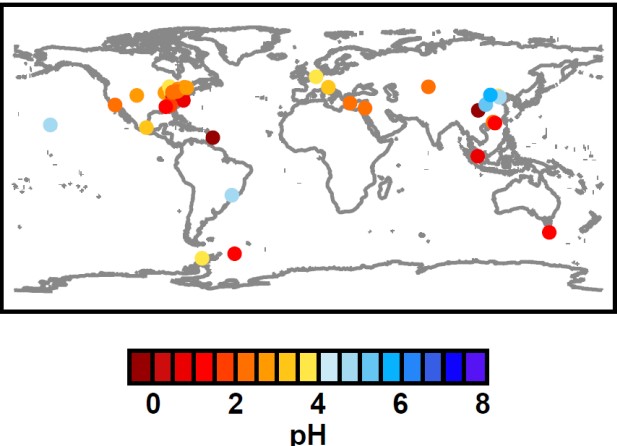

**Figure 6.** Observationally estimated ground-level fine aerosol pH. Estimates are primarily based on observationally-constrained thermodynamic equilibrium model predictions reported as $pH_F$. Values correspond to present-day conditions.

5   Measurement locations, measurement time periods, reported pH values, and citations are listed in Table S6. Values shown in the Figure will be available in tabular format via data.gov (see data availability statement).





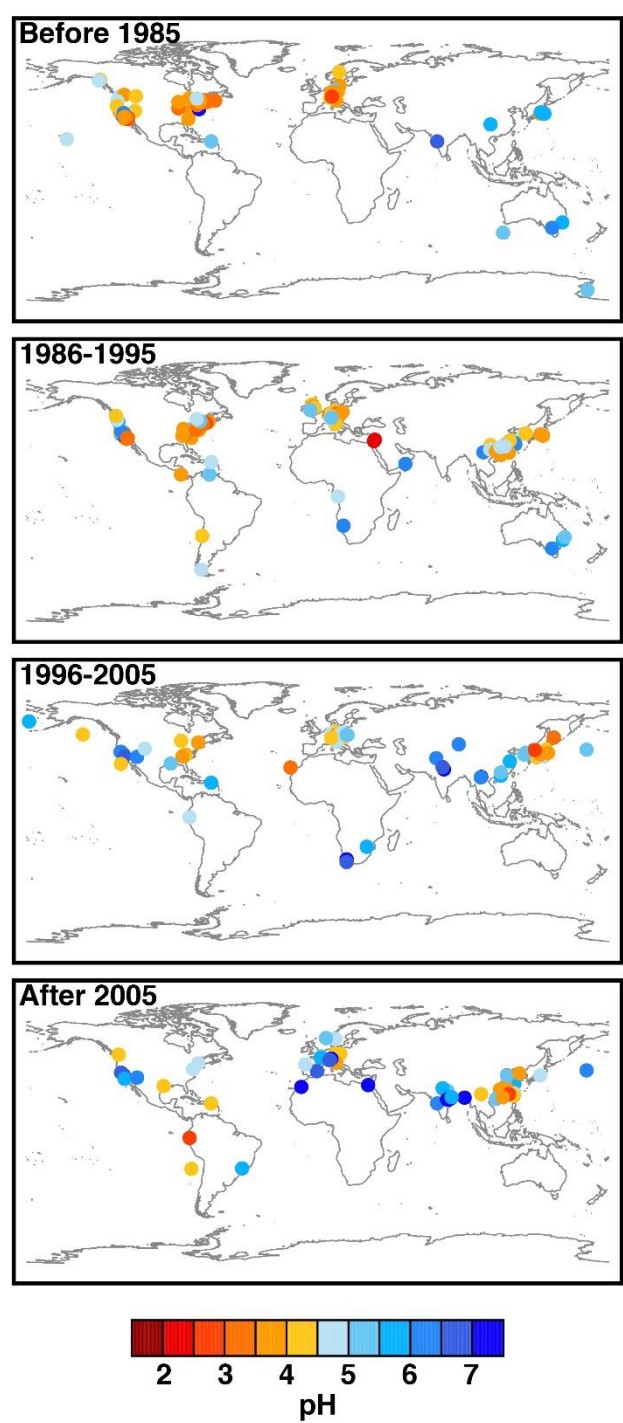

**Figure 7.** Values of pH in fog and cloud water samples collected around the globe. The measurements are divided into 4 time periods to provide better comparison across an era of changing anthropogenic emissions. Plotted points represent reported





mean pH values. In cases where mean pH was not reported or calculable from reported data, either the reported median pH or the average of reported minimum and maximum H$^+$ concentrations converted to pH are plotted. Measurement locations, measurement time periods, reported pH values, and citations are listed in Table S8. Values shown will be available in tabular format via data.gov (see data availability statement).





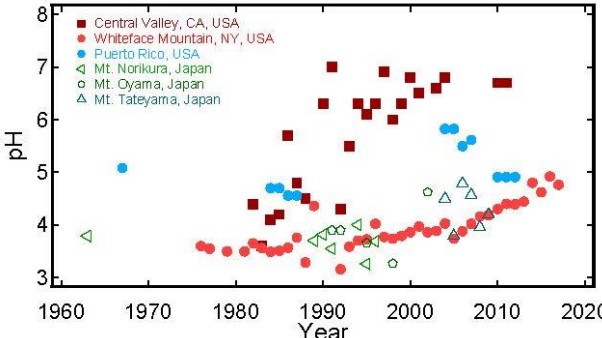

**Figure 8.** Mean cloud/fog pH values reported for individual measurement locations and years, from the 1960s to present. Sites are included from the western and northeastern United States, from the Caribbean island of Puerto Rico, and from locations in central Japan. See text and Table S8 for references.



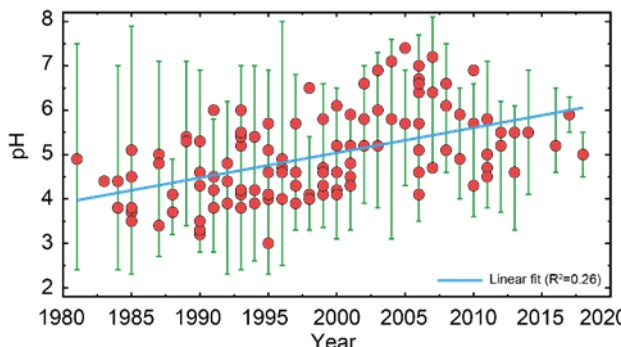

**Figure 9.** Trend in measured pH values of fogs and clouds sampled in Europe from the 1980s to present. See Table S8 for a list of sites and references. The dots represent the average or median values measured at the site and the green interval bars the range (minimum, maximum) in the reported data distributions. The trend line indicates an increase of approximately 0.56 pH units per decade.

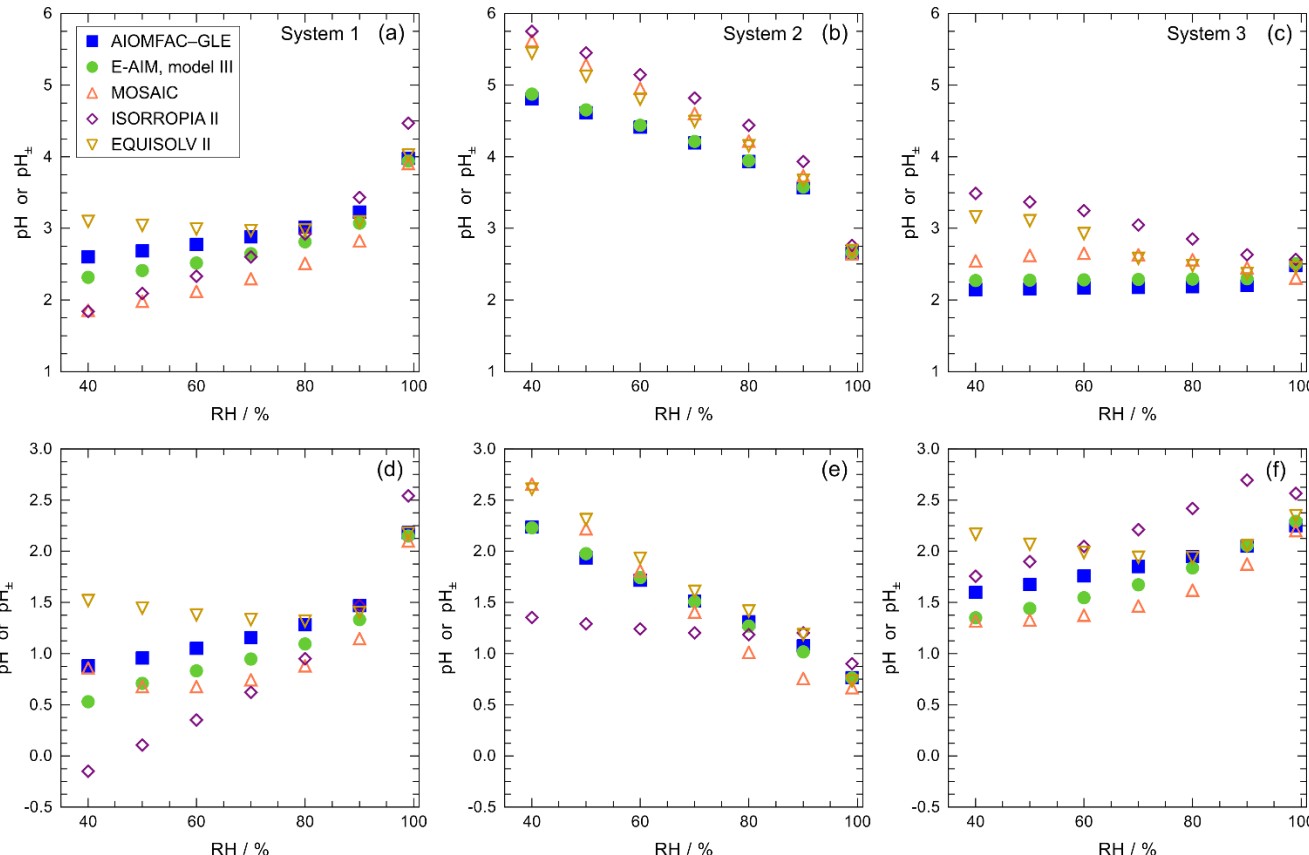

**Figure 10.** Comparison of pH predictions by six aerosol thermodynamics models for seven RH levels, including gas–liquid partitioning of volatile components (see details in Table 3). Upper panels show moderately acidic cases, lower panels highly acidic cases (with different *y*-axis scaling). Systems are described in Sect. 6.1: (a, d) System 1: sulfate-rich aqueous $(NH_4)_2SO_4$ + $H_2SO_4$ + $NH_3$; (b, e) system 2: acidified sea-salt-like aqueous aerosol $(Na_2SO_4$ + $NaCl$ + $H_2SO_4$ + $HCl$); (c, f) system 3: nitrate-rich aqueous $(NH_4)_2SO_4$ + $H_2SO_4$ + $NH_3$ + $HNO_3$. Models E-AIM and AIOMFAC (solid symbols, green and blue respectively) predict pH based on the single-ion activity coefficient of $H^+$, the other models (open symbols, MOSAIC in upward orange triangle, ISORROPIA II in purple diamond, EQUISOLV II in downward yellow triangle) approximate pH by a version of $pH_\pm$; the specific mean ion activity coefficients used for $pH_\pm$ by those models are listed in supplemental Tables S3 – S5 for each system.




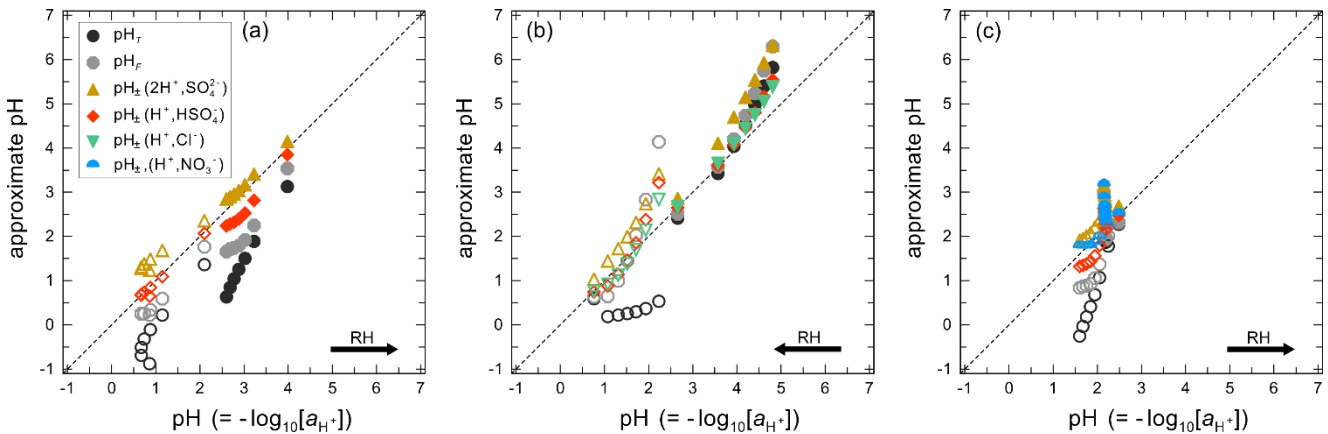

**Figure 11.** Comparison of different approximate pH values *vs.* the molal pH by system (panels a, b, c for system 1, 2 ,3) introduced in Sect. 6.1 and also shown in Fig. 10, all calculated by AIOMFAC–GLE. Solid symbols show the moderately acidic cases and open symbols the highly acidic cases. pH approximations are based on total $H^+$ ($pH_T$), free $H^+$ ($pH_F$), and $pH_\pm$ (defined in Table 1). For the $pH_\pm$ variants, the $H^+$ activity coefficient was approximated by the mean molal activity coefficient of $H^+$ combined with either $SO_4^{2-}$, $HSO_4^-$, $Cl^-$ or $NO_3^-$ ($Cl^-$ only for system 2; $NO_3^-$ only for system 3). Arrows on the lower right indicate the relationship between increasing RH and pH for each system; the highest RH shown is 99 % in all cases. Colors indicate different pH approximations ($pH_T$ vs $pH_F$ vs $pH_\pm$) including activity coefficient approximations based on different ion pairs (for $pH_\pm$).



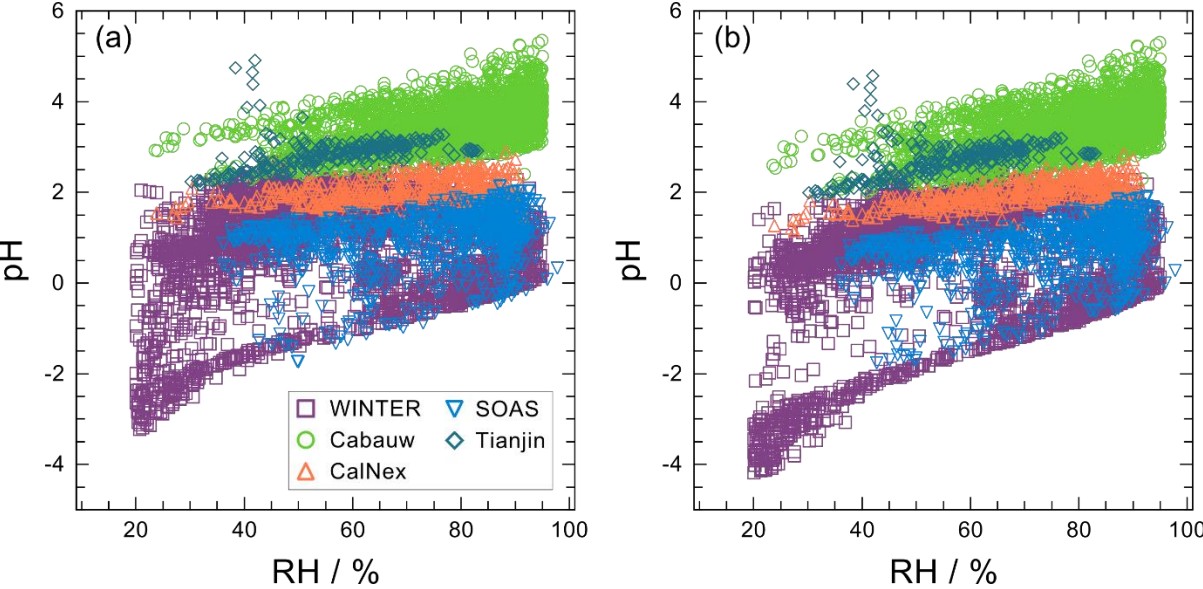

**Figure 12.** A comparison of calculated pH using (a) AIOMFAC-GLE, and (b) E-AIM as a function of RH for all field campaign
datasets examined (SOAS Centreville, Cabauw, CalNex, Tianjin and WINTER, see Sect. 6.2 and Table 4 for a description of
the data sets).





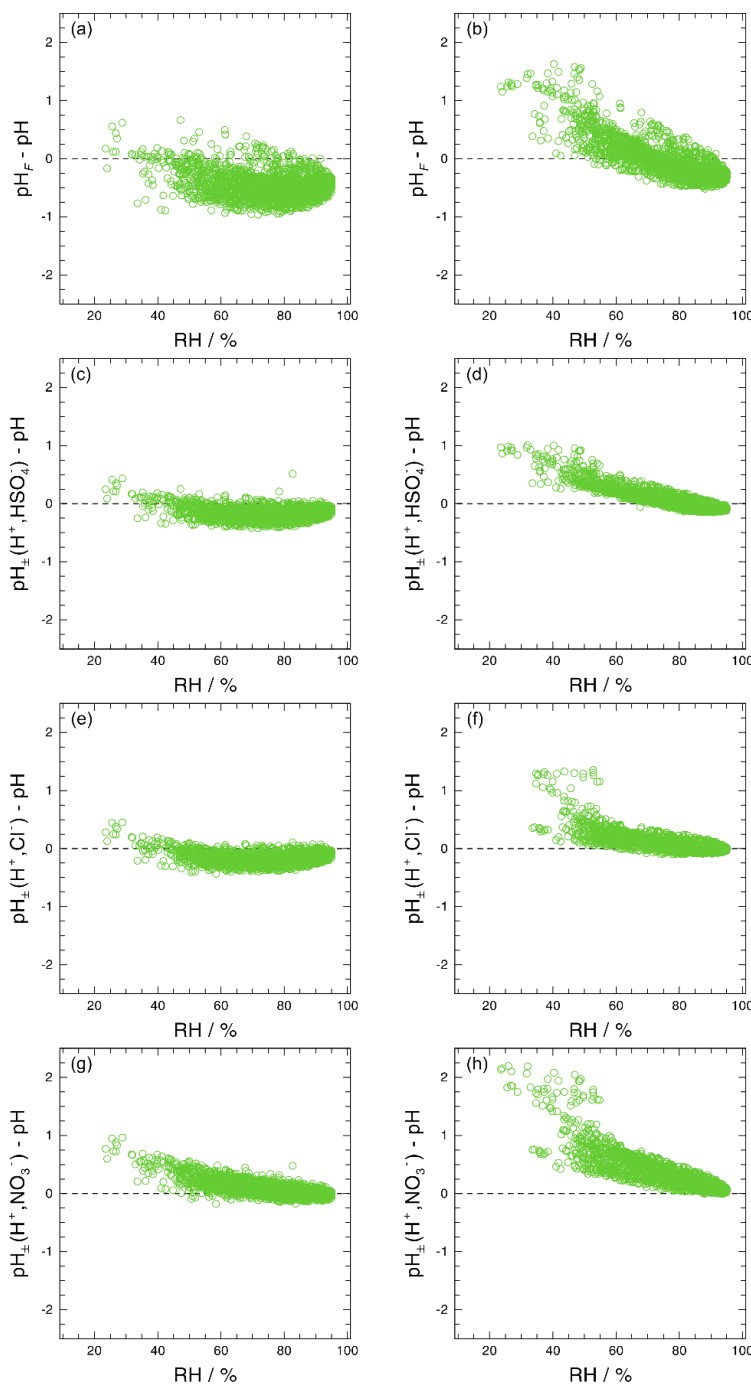

**Figure 13.** Comparison of different approximate metrics of pH, using the Cabauw data and calculated by AIOMFAC-GLE (left column), and E-AIM (right column). The results are shown as differences from pH as follows: (a, b) $pH_F$ - pH; (c, d) $pH_{\pm}$ $(H^+, HSO_4^-)$ - pH; (e, f) $pH_{\pm}(H^+, Cl^-)$ - pH; (g, h) $pH_{\pm}(H^+, NO_3^-)$ - pH.



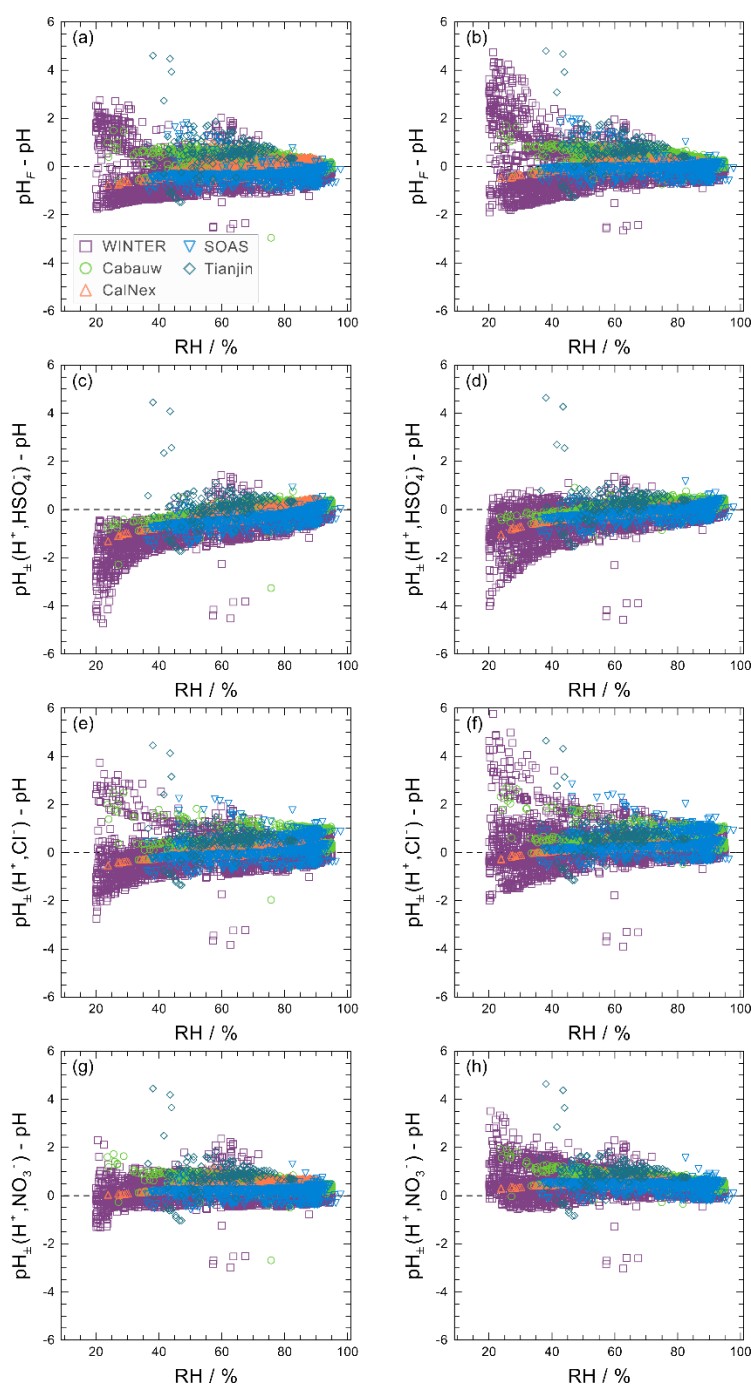

**Figure 14.** ISORROPIA-calculated pH using the different metrics, and its difference against pH values calculated with AIOMFAC-GLE (left column) and, E-AIM (right column). Data shown for all the field campaign observations considered in this study (Table 4).









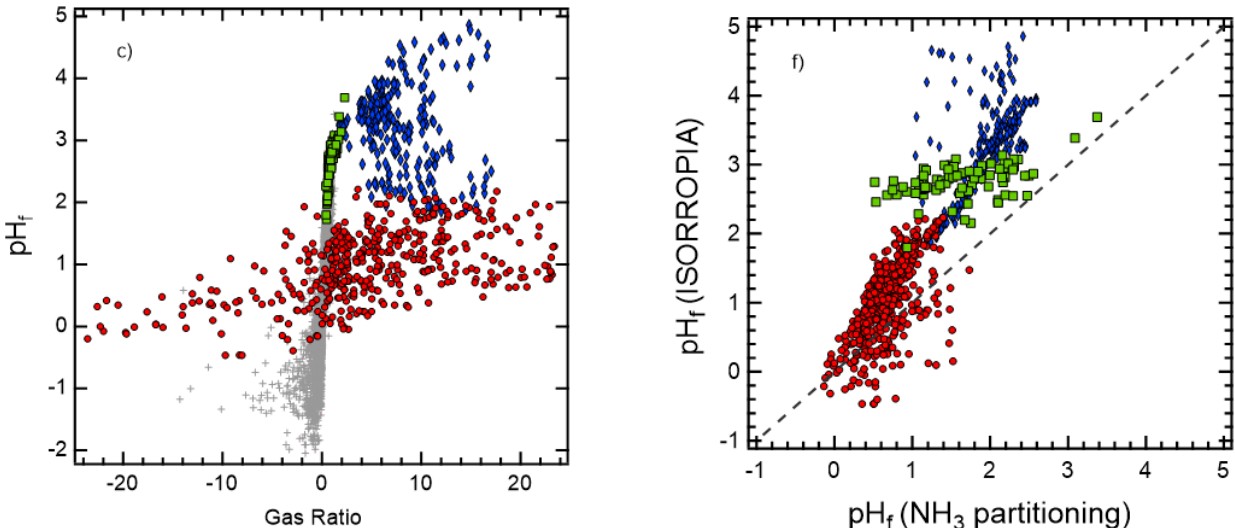

**Figure 15.** Comparison of pH$_F$ calculated by ISORROPIA with proxies: (a) the charge balance, (b) cation/anion molar equivalent ratio, (c) gas ratio (GR), (d, e) HNO$_3$ gas–particle partitioning (Eq. 17), and (f) NH$_3$ gas–particle partitioning (Eq. 18). The ambient data sets are described in detail in Table 4. The proxy methods were calculated as in Table 2, unless otherwise noted. Note the convention for the charge balance calculation in (a) results in a positive value when there is a cation deficit. The dashed lines in d-f represent the 1:1 line, shown for visual effect. The legend in panel (a) applies to all panels except (e), which only shows data for the WINTER data set, colored according to the aerosol water mass fraction (relative to the total predicted aerosol mass).





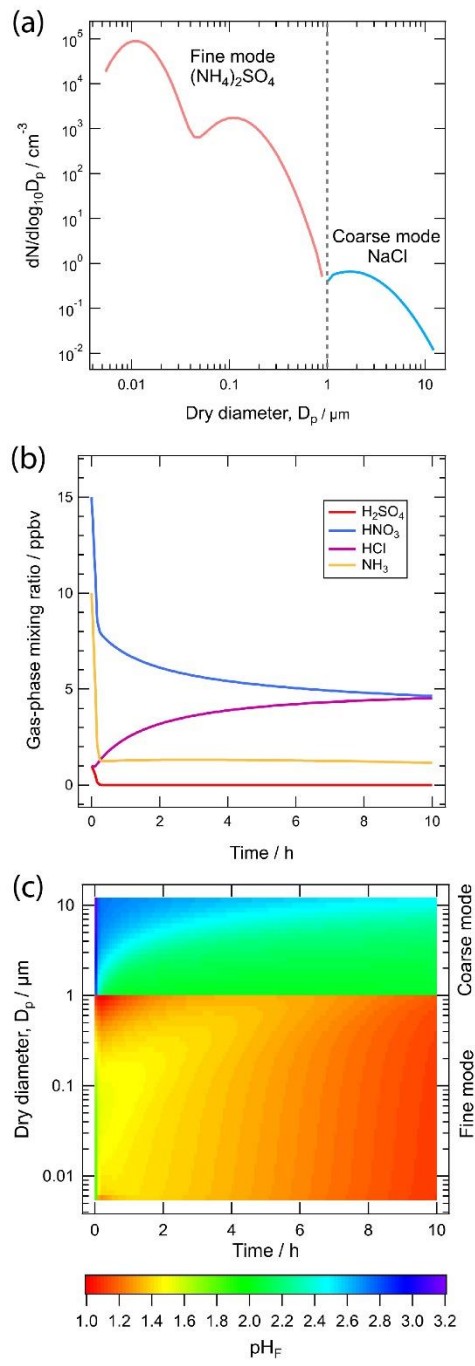

**Figure 16.** Simulated evolution of size-distributed aerosol pH using the sectional MOSAIC aerosol box model: (a) Initial aerosol number size distributions for the fine and coarse modes, (b) Time evolution of gas-phase species, and (c) Time evolution of aerosol pH as a function of size (bins labeled based on what was initially in the fine and coarse modes).





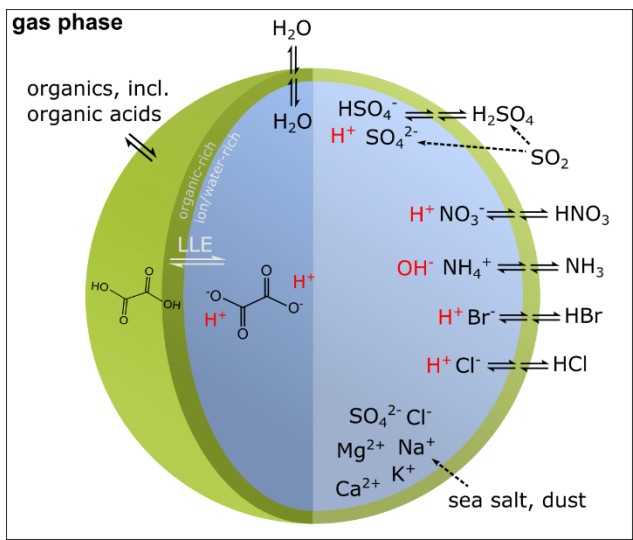

**Figure 17.** Sketch of a multicomponent aerosol particle consisting of an organic-rich and an aqueous inorganic-ion rich phase

5   in LLPS; here adopting a core–shell morphology. Shown are equilibria of water, organic compounds, as well as inorganic acids and bases contributing to the overall particulate matter mass, ionic strength and the pH. The dynamically established pH exerts control on the gas–particle partitioning of semi-volatile acids and bases, such as $HNO_3$, $HCl$ and $NH_3$. All species may partition into all phases; however, inorganic ions tend to favour the aqueous phase, while organic compounds of moderate to low polarity will predominantly partition to an organic-rich phase.





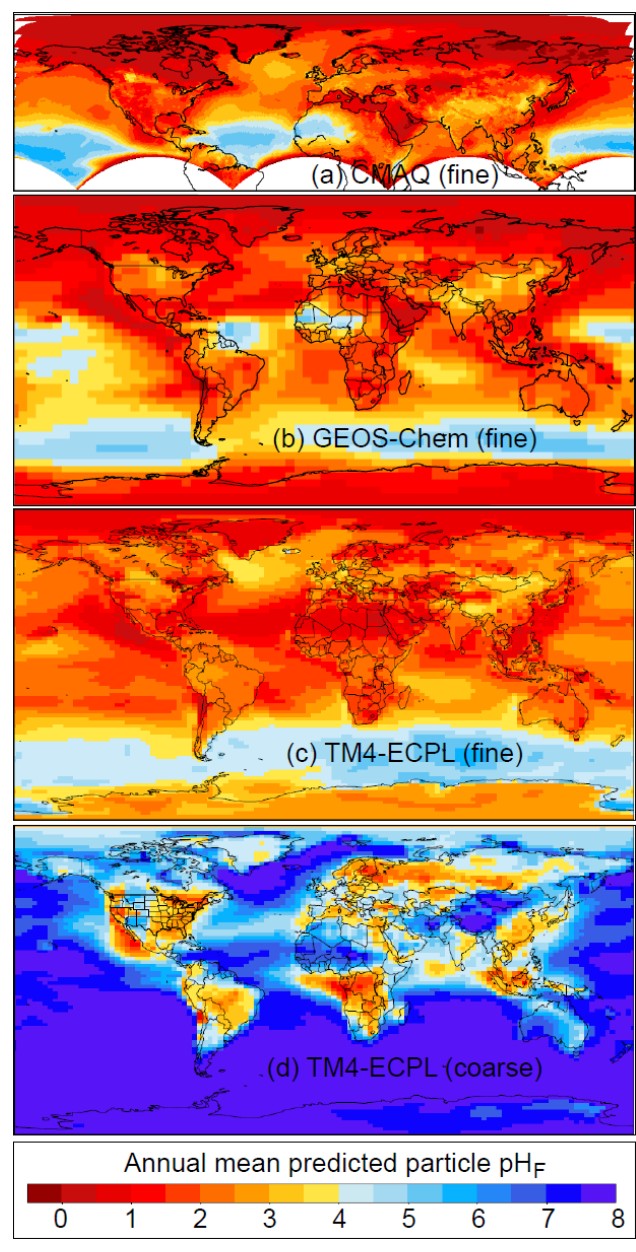

**Figure 18.** Particle $pH_F$ at the surface predicted by (a) CMAQv5.2 for Aitken + accumulation modes (2016 northern hemisphere annual average), (b) GEOS-Chem for bulk fine aerosol (2015 annual average, version 12.0.0 with additional dust cations), (c) TM4-ECPL for $PM_1$ (2009 annual average), and (d) TM4-ECPL for coarse aerosol (2009 annual average). Values averaged over aerosol liquid water content greater than 0.01 μg m$^{-3}$. The solvent for H$^+$ is water associated with inorganic electrolytes.





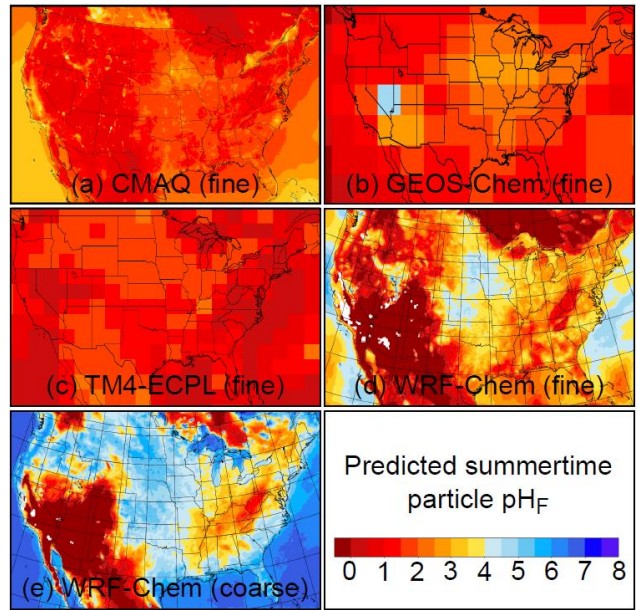

**Figure 19.** Particle pH$_F$ predicted at the surface for June over the contiguous United States by (a) CMAQv5.2 for Aitken +
accumulation modes (2016), (b) GEOS-Chem for bulk fine aerosol (2015), (c) TM4-ECPL for PM$_1$ (2009), (d) WRF-Chem
v3.9.1 with MOSAIC for PM$_{2.5}$ aerosols (June 1-14, 2013, liquid water weighted average), and (3) WRF-Chem for coarse
aerosol (2.5 to 10 μm). Values averaged over aerosol liquid water content greater than 0.01 μg m$^{-3}$. The solvent for H$^+$ is water
associated with inorganic electrolytes in a-c and total aerosol water (including water associated with organics and OIN) in d-
e.





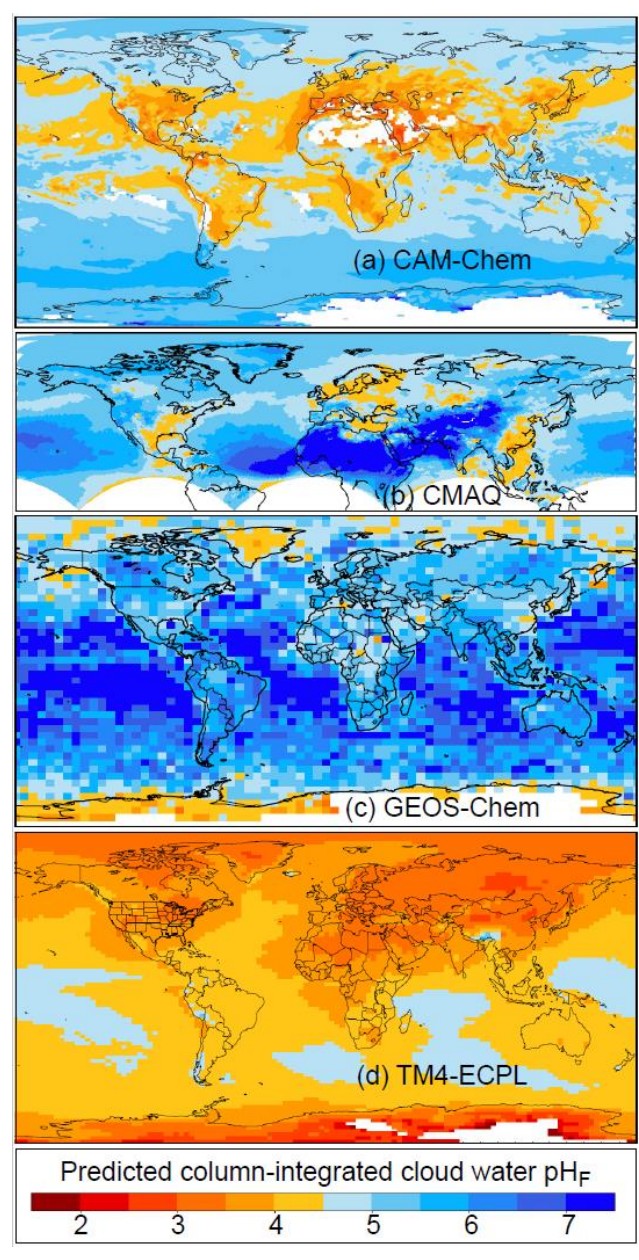

**Figure 20.** Liquid-water-weighted vertical column integrated cloud water $pH_F$ predicted by: (a) CAM6-Chem (convective clouds excluded, June 2015), (b) CMAQv5.3 (resolved clouds only, 2016 annual average), (c) GEOS-Chem (2015 annual average), and (d) TM4-ECPL (2009 annual average). Note different color scale compared to particle predictions. White indicates no cloud water.





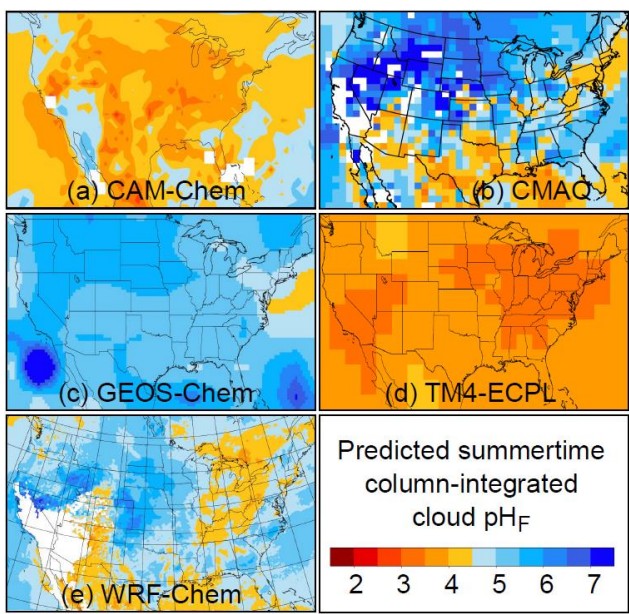

**Figure 21.** Liquid-water-weighted vertical column June average cloud water $pH_F$ over the contiguous United States predicted by: (a) CAM6-Chem (convective clouds excluded, 2015), (b) CMAQv5.3 (resolved clouds only, 2016), (c) GEOS-Chem (2015), (d) TM4-ECPL (2009), and (e) WRF-Chem (2013). White indicates no cloud water.



**Tables**

**Table 1. Definitions and notation for pH and various molality-based pH approximations. All activity coefficients are on a molality basis, relative to a reference state of infinite dilution in pure water. Note, for mathematical rigor (omitted above for simplicity), all expressions in the $\log_{10}$ need to be normalized by unit molality $m^{\ominus}$.[b]**

| Symbol and defining expression | Remarks |
|---|---|
| $\mathrm{pH} = -\log_{10}(a_{\mathrm{H^+}}) = -\log_{10}\left(m_{\mathrm{H^+}}\gamma_{\mathrm{H^+}}\right)$ | recommended IUPAC definition of pH |
| $\mathrm{pH_F} = -\log_{10}(m_{\mathrm{H^+}})$ | *free* H$^+$ approximation of pH |
| $\mathrm{pH_{\pm}(H,X)} = -\log_{10}\left(m_{\mathrm{H^+}}\gamma_{\pm,\mathrm{HX}}\right)$ | approximation of pH based on mean molal ion activity coefficient [a] of H$^+$ and anion X, with X = Cl$^-$, NO$_3^-$ or HSO$_4^-$ (choice to be specified in parenthesis or as subscript) |
| $\mathrm{pH_T} = -\log_{10}(m_{\mathrm{H^+}} + m_{\mathrm{HSO4^-}})$ | *total* H$^+$ approximation of pH |

[a] For 1:1 electrolytes, $\gamma_{\pm,\mathrm{HX}} = \sqrt[2]{\gamma_{\mathrm{H^+}} \cdot \gamma_{\mathrm{X}}}$ . The difference between $\mathrm{pH_{\pm}(H,X)}$ and pH is related to the activity coefficient ratio, $\mathrm{pH_{\pm}(H,X)} - \mathrm{pH} = \frac{1}{2}\log_{10}\left(\frac{\gamma_{\mathrm{H^+}}}{\gamma_{\mathrm{X^-}}}\right)$ .

[b] With explicit normalization, $\mathrm{pH_F} = -\log_{10}\left(\frac{m_{\mathrm{H^+}}}{m^{\ominus}}\right)$ .



**Table 2. Definitions of proxy methods. For all quantities the units are moles of chemical species per unit volume of air (e.g., mol m-3). The quantity $[TSO_4]$ is total particulate sulfate, or the sum of $[HSO_4^-] + [SO_4^{2-}]$.**

| Proxy | Definition |
|---|---|
| Cation/anion equivalent ratio | $\text{Cation}/\text{Anion} = \dfrac{[NH_4^+] + [Na^+] + [K^+] + 2[Ca^{2+}] + 2[Mg^{2+}]}{2[TSO_4] + [NO_3^-] + [Cl^-]}$ |
| Degree of sulfate neutralization | $DSN = ([NH_4^+] - [NO_3^-])/[TSO_4]$ |
| Degree of neutralization | $DON = [NH_4^+]/(2[TSO_4] + [NO_3^-])$ |
| TNH₄: TSO₄ | $TNH_4 : TSO_4 = \dfrac{([NH_3] + [NH_4^+])}{[TSO_4]}$ |
| Strong acidity (charge balance) | $H_{air,cb}^+ = 2[TSO_4] + [NO_3^-] + [Cl^-] - ([NH_4^+] + [Na^+] + [K^+] + 2[Ca^{2+}] + 2[Mg^{2+}])$ |
| Gas ratio (GR) | $GR = ([NH_3] + [NH_4^+] - 2[TSO_4])/([HNO_3] + [NO_3^-])$ |
| Adjusted gas ratio (adjGR) | $adjGR = ([NH_3] + [NO_3^-])/([HNO_3] + [NO_3^-])$ |





**Table 3. Input concentrations (total gas+aerosol) for each of the highly and moderately acidic cases of inorganic systems 1 – 3. Compositions are specified in terms of moles of electrolyte and gas-phase species per m³ of air. Model calculations were carried out for seven equilibrium RH from 99% to 40%, at 298.15 K and 101.325 kPa pressure.**

| System no., case | $n_{(NH4)2SO4}$ (mol m⁻³) | $n_{H2SO4}$ (mol m⁻³) | $n_{NH3}$ (mol m⁻³) | $n_{HNO3}$ (mol m⁻³) | $n_{Na2SO4}$ (mol m⁻³) | $n_{NaCl}$ (mol m⁻³) | $n_{HCl}$ (mol m⁻³) |
|---|---|---|---|---|---|---|---|
| 1, moderately acidic | 2.50000E-08 | 3.37165E-11 | 1.00E-06 | 0 | 0 | 0 | 0 |
| 1, highly acidic | 1.50000E-08 | 2.02097E-08 | 5.00E-08 | 0 | 0 | 0 | 0 |
| 2, moderately acidic | 0 | 1.00000E-10 | 0 | 0 | 5.61260E-09 | 1.04451E-07 | 1.00E-08 |
| 2, highly acidic | 0 | 5.00000E-08 | 0 | 0 | 5.61260E-09 | 1.04451E-07 | 1.00E-06 |
| 3, moderately acidic | 2.50000E-08 | 3.37165E-11 | 1.00E-06 | 1.00E-06 | 0 | 0 | 0 |
| 3, highly acidic | 2.50000E-08 | 3.37165E-11 | 1.00E-07 | 1.00E-07 | 0 | 0 | 0 |



**Table 4. Characteristics of the datasets used for the box model intercomparison. Values are reported as mean followed be standard deviation in parenthesis. n is the number of data points. (RH, T, and concentration are the same as those in Nenes et al., 2019).**

| Dataset ID (reference) | Location (Period) | RH (%) | T (K) | Sulfate ($\mu g\ m^{-3}$) | Total Ammonium ($\mu g\ m^{-3}$) | Total Nitrate ($\mu g\ m^{-3}$) | n |
|---|---|---|---|---|---|---|---|
| Tianjin (Shi et al., 2019) | Tianjin, China (9/Aug/2015- 22/Aug/2015) | 56.6 (12.4) | 301.8 (2.79) | 21.46 (10.99) | 37.74 (7.68) | 18.12 (11.50) | 227 |
| CALNEX (Guo et al., 2017b) | Pasadena, CA, USA (17/May/2010- 15/Jun/2010) | 71.3 (15.5) | 291.1 (4.26) | 2.86 (1.70) | 3.44 (1.81) | 10.23 (9.74) | 482 |
| Cabauw (Guo et al., 2018b) | Cabauw, Netherlands (2/May/2012- 04/Jun/2013) | 78.2 (14.8) | 282.2 (7.3) | 1.92 (1.57) | 9.3 (6.8) | 4.1 (3.9) | 2612 |
| WINTER (Guo et al., 2016) | Eastern USA (03/Feb/2015) | 56.1 (18.9) | 270.8 (6.52) | 1.02 (0.08) | 0.53 (0.44) | 2.12 (2.08) | 3121 |
| SOAS (Guo et. al, 2015) | Centreville, USA (06/Jun/2013- 14/Jul/2013) | 72.7 (17.4) | 297.9 (3.45) | 1.81 (1.18) | 0.78 (0.50) | 0.12 (0.15) | 780 |



**Table 5. (a) Comparison of ISORROPIA II-derived pH approximations against pH. (b, c) Comparison of pH approximations against pH when computed by the same model in (b) AIOMFAC-GLE and in (c) E-AIM.**

| | RMSE compared to AIOMFAC-GLE pH | MB compared to AIOMFAC-GLE pH | RMSE compared to E-AIM pH | MB compared to E-AIM pH |
|---|---|---|---|---|
| **(a)** pH approximation by ISORROPIA II[1] | | | | |
| $pH_F$ | 0.457 | -0.065 | 0.452 | 0.097 |
| $pH_\pm$ ($H^+$, $HSO_4^-$) | 0.505 | -0.230 | 0.393 | -0.068 |
| $pH_\pm$ ($H^+$, $Cl^-$) | 0.449 | 0.170 | 0.527 | 0.331 |
| $pH_\pm$ ($H^+$, $NO_3^-$) | 0.513 | 0.372 | 0.634 | 0.534 |
| **(b)** pH approximation by AIOMFAC-GLE[2] | | | | |
| $pH_F$ | 0.611 | -0.544 | - | - |
| $pH_\pm$ ($H^+$, $HSO_4^-$) | 0.272 | -0.235 | - | - |
| $pH_\pm$ ($H^+$, $Cl^-$) | 0.230 | -0.200 | - | - |
| $pH_\pm$ ($H^+$, $NO_3^-$) | 0.181 | 0.073 | - | - |
| **(c)** pH approximation by E-AIM[2] | | | | |
| $pH_F$ | - | - | 0.546 | 0.123 |
| $pH_\pm$ ($H^+$, $HSO_4^-$) | - | - | 0.354 | 0.182 |
| $pH_\pm$ ($H^+$, $Cl^-$) | - | - | 0.497 | 0.233 |
| $pH_\pm$ ($H^+$, $NO_3^-$) | - | - | 0.728 | 0.493 |

[1] The pH, as defined by Eq. (1), was calculated using both AIOMFAC-GLE and E-AIM. The comparisons are presented in
5  terms of the root mean square error (RMSE) and mean bias (MB) as pH(approx.) - pH, all in pH units. Results were
calculated using all of the SOAS, Cabauw, CalNex, WINTER, and Tianjin datasets (combined) described in Table 4 (n = 7222 points), with RMSE and MB calculations limited to data points with RH > 35 %.
[2] Calculations by (b) AIOMFAC-GLE and (c) E-AIM covering the same combined data sets as in (a).





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
