# Peer review of "The Acidity of Atmospheric Particles and Clouds"

_Atmospheric Chemistry and Physics, 2019_

## Referee Comment (RC1) · Anonymous Referee #1 · 2 Nov 2019

This is an excellent and thorough review of the current state-of-science on the acidity of atmospheric aerosols and droplets. Below are some comments intended to improve the manuscript:

1. The Abstract is focused on the electrolytes and does not discuss the organic fraction and its contribution. Something should be mentioned about organic bases, such as amines, and organic acids such carboxylics and dicarboxylics.

2. Same in section 1 around line 22.

3. The use of the world "occult" is completely foreign to me. It should at least be defined, but better would be to use a word more familiar.

4. Page 4, line 32. Nicotine (and cocaine for that matter) are organic bases. increasing

pH partitions these compounds to the gas phase (so-called free-basing). The gaseous compounds have a much higher deposition efficiency to the respiratory tract than particles increasing the dose.

5. section 3, page 17, line 26. Change The to This.

6. Section 5, page 30, line 17: Why do you say that cloud liquid water is not in equilibrium with the gas phase? Well, nothing is completely in equilibrium but the surface area in a cloud is huge so equilibrium is nearly attained.

7. The authors many not be aware of the work by Beverly Cohen published in 2000 using a metal plate to assess aerosol acidity.

8. Page 33, section 5.1.3, line 31: Comment should be common.

9. Page 58, lines 22-27: Something should be said here about the role of volatile acids (HCl, HNO3) and bases (NH3, amines). If these exist in the gas phase, they serve to equilibrate acidity across the particle size distribution. If the particles are so acidic that the volatile bases are depleted from the gas phase, I can see how there could be a gradient in acidity with particle size. Likewise for the particles being alkaline and the volatile acids being depleted. As the authors discuss elsewhere, these gases are key to understanding the acidity of particles but unfortunately they are infrequently measured. I encourage a simple discussion of this logic.

---

## Referee Comment (RC2) · Anonymous Referee #2 · 13 Nov 2019

Pye et al. provides a detailed review concerning aerosol and cloud acidity. This is a much needed review, as the studies concerning aerosol acidity has increased. I applaud the large author list on this work, as it is very extensive and impressive. The review provides a much needed discussion concerning various aspects of cloud and aerosol pH, including definition, measurements and calculations, observations, and comparisons with models. I foresee this paper becoming an important source, both in the field and for introducing this subject in classes. This paper should be published upon consideration of the following comments:

1.) Similar to Reviewer #1, occult is not a term that I am familiar with. If it could be defined, or a synonym could be used, that would be appreciated.

2.) I appreciate Table 1, as there are numerous abbreviations throughout the

manuscript. However, there appears to be some abbreviations missing, such as NVCs. Please include all abbreviations in the manuscript into Table 1, as it can be hard to find them in this large manuscript.

3.) I think a Table that summarizes Section 2.6 would improve the quality of the paper. This will be one of the many reasons people will want to read and cite the paper, concerning a discussion and comparison of the thermodynamic models in order to calculate pH. Including a table that summarizes how each model calculates pH, the pros and cons of each model, any assumptions, and etc., would help the readers remember that discussion better.

4.) Throughout the sections, it is apparent that different people wrote them with different styles coming through in each section. For example, some sections briefly state future research while other sections devote a subsection about future research (and some sections do not have any discussion about questions/future work). Starting with at least Section 3, if not some aspects of Section 2 (i.e., Measuring pH), it would be beneficial for the authors to include a description of what they consider future questions/work to be.

5.) For Section 3, it would be good to include a couple of items in discussion, including: (a) How many of the reactions have been conducted for dilute, laboratory conditions, therefore, for aerosol, where the ion activity is higher and water is lower, there is large uncertainty in how the reactions may occur. (b) How there is debate occurring the field about various reactions (e.g., production of sulfate in aerosol in China) and the questions/future studies needed to move this questions forward. (c) N2O5 chemistry appears to be missing in your discussion throughout, including in Section 5.

6.) For Section 8, a table that summarizes the CTMs with the thermodynamic models they use and the species they use to calculate pH would help with the discussion.

7.) For Section 9, I appreciate that it summarizes the very large review. However, at this point, I really think a description of remaining questions, studies, observations and

future outlook is necessary so that we, as a community, know what should be done to move forward.

---

## Author Response (AR1)

Author responses in blue. Planned additions to the manuscript are underlined (or shown via screen shot).

**Anonymous Referee #1**

This is an excellent and thorough review of the current state-of-science on the acidity of atmospheric aerosols and droplets. Below are some comments intended to improve the manuscript:

We thank the reviewer for the supportive comments.

1. The Abstract is focused on the electrolytes and does not discuss the organic fraction and its contribution. Something should be mentioned about organic bases, such as amines, and organic acids such carboxylics and dicarboxylics.

We revised a sentence in the abstract to explicitly mention organic acid/base partitioning. The role of organic species in driving pH is beginning to be examined, so we highlighted the well-known role of pH on organic acid/base partitioning.

> In the atmosphere, the acidity of condensed phases (aerosol particles, cloud water, and fog droplets) governs the phase partitioning of semi-volatile gases such as $HNO_3$, $NH_3$, HCl, and organic acids and bases as well as chemical reaction rates.

2. Same in section 1 around line 22.

A list of specific organic acids/bases were added:

> Semi-volatile species, for which significant fractions typically exist in both the gas and condensed phases, include ammonia ($NH_3$), nitric acid ($HNO_3$), hydrochloric acid (HCl), and low molecular weight organic acids (formic, acetic, oxalic, malonic, succinic, glutaric, and maleic acids) and/or bases (e.g. amines).

3. The use of the world "occult" is completely foreign to me. It should at least be defined, but better would be to use a word more familiar.

Occult is now defined (we keep the term for compactness in the figure):

> The acidity of atmospheric deposition for dry, wet, and occult (wind-driven cloud-water) pathways is directly affected by aerosol and cloud pH (Fig. 1).

4. Page 4, line 32. Nicotine (and cocaine for that matter) are organic bases. increasing pH partitions these compounds to the gas phase (so-called free-basing). The gaseous compounds have a much higher deposition efficiency to the respiratory tract than particles increasing the dose.

We clarified the statement to indicate the role of particle acidity was to drive nicotine off surfaces and into the indoor air. We also added a reference to the partitioning mechanism for alkaloids to indicate the behavior may apply to other species as well.

> Aerosol acidity was also recently shown to enhance airborne nicotine levels and resulting thirdhand smoke exposure by promoting volatilization from surfaces (such as clothes) and allowing distribution throughout a building's indoor air (DeCarlo et al., 2018). Similar behavior may be possible for other alkaloids (Pankow, 2001).

5. section 3, page 17, line 26. Change The to This.

Updated

6. Section 5, page 30, line 17: Why do you say that cloud liquid water is not in equilibrium with the gas phase? Well, nothing is completely in equilibrium but the surface area in a cloud is huge so equilibrium is nearly attained.

In a relative sense, most (non-glassy) aerosols are far closer to equilibrium than clouds can ever be. This is because their size, hence equilibration times, is orders of magnitude smaller. For this reason you can  *frequently (word choice revised Feb 2020)* assume thermodynamic equilibrium for submicron, but not supermicron aerosol. Droplets are tens of microns in diameter and reside typically in a moving air parcel (e.g. in a convective cloud) which experiences changes in temperature and local saturation ratio, so the equilibrium assumption is often less satisfied (Seinfeld and Pandis, 1998).

The comment "water is not in equilibrium with the gas phase" refers to the effect of perturbing the RH of the air surrounding the sample. For aerosol, even a 10% change in RH can profoundly impact liquid water, as equilibration with RH is almost instantaneous. Cloud liquid water, on the other hand, is not driven by RH but by the cooling rate that generates them. After sampling, the evaporation biases have a much smaller effect precisely because the liquid volume is so large. We have removed the reference to "water equilibration" to avoid unnecessary confusion.

Sentence to be modified from:

> The same limitations do not exist for fog/cloud water or precipitation, where larger sample volumes can be collected from accessible clouds, the associated water is not in equilibrium with the gas phase (although evaporation artifacts may still cause biases), and solutions are dilute enough to allow for a direct pH measurement. The latter has been done with electrochemical pH probes for decades (Sect. 5.2).

To:

> The sample volumes for fog/cloud water or precipitation are orders of magnitude larger than for aerosols and can be collected from clouds using well established instrumentation. This, together with their dilute concentration allow for a direct pH measurement, which has been done with electrochemical pH probes for decades (Sect. 5.2).

7. The authors many not be aware of the work by Beverly Cohen published in 2000 using a metal plate to assess aerosol acidity.

We were not aware but have located a report and article:
1. Field Evaluation of Nanofilm Detectors for Measuring Acidic Particles in Indoor and Outdoor Air By: Beverly S Cohen, Maire SA Heikkinen, Yair Hazi, Hai Gao, Paul Peters, and Morton Lippmann, https://www.healtheffects.org/system/files/Cohen.pdf
2. Cohen et al., Detecting H+ in ultrafine ambient aerosol using iron nano-film detectors and scanning probe microscopy, https://doi.org/10.1080/104732200301881.

The technique is used to determine the number of acidic vs nonacidic particles. It provides some qualitative information (e.g. H2SO4 and NH4HSO4 are more acidic than (NH4)2SO4). We have not added a reference due to the qualitative nature of the study.

8. Page 33, section 5.1.3, line 31: Comment should be common.

Updated

9. Page 58, lines 22-27: Something should be said here about the role of volatile acids (HCl, HNO3) and bases (NH3, amines). If these exist in the gas phase, they serve to equilibrate acidity across the particle size distribution. If the particles are so acidic that the volatile bases are depleted from the gas phase, I can see how there could be a gradient in acidity with particle size. Likewise for the particles being alkaline and the volatile acids being depleted. As the authors discuss elsewhere, these gases are key to understanding the acidity of particles but unfortunately they are infrequently measured. I encourage a simple discussion of this logic.

We added a new sentence in the paragraph highlighting the role for semivolatile acids and bases:

Overall, atmospheric particle pH is size dependent and generally higher for coarse mode particles due to variations in inorganic composition with particle size. Differences as large 4 pH units have been reported between fine and coarse particles (Fang et al., 2017; Young et al., 2013). Bulk $PM_1$ and $PM_{2.5}$ acidity is more similar than fine vs coarse mode acidity ($pH_F$ within $1 – 2$ units, e.g., Bougiatioti et al., 2016; Guo et al., 2017b), but submicron (diameter < 1 μm) particles still show higher acidity than bulk $PM_{2.5}$. The reason for this is the strong enrichment of aerosol with NVCs from dust and sea salt at the larger sizes (even in the fine mode) and role of sulfate in new particle formation and surface-area driven condensation at the small sizes (Fig. 2). While semivolatile acids and bases act to homogenize acidity across the size distribution, mass transfer limitations (next section) and the heterogeneity of emission composition lead to variation in pH with size. Significant pH changes can occur in the 1 to 2.5 μm size range (Fang et al., 2017; Ding et al., 2019). The size dependent pH is also seen for sea salt aerosol (Fridlind and Jacobson, 2000) as well as in urban aerosols in China (Ding et al., 2019) where the fine mode is consistently 2-3 pH units lower than the coarse mode. The implications of this acidity gradient are considerable, for metal solubility and their impacts on public health and ecosystem productivity, as well as chemistry and semi-volatile partitioning of pH-sensitive species.

**Anonymous Referee #2**

Pye et al. provides a detailed review concerning aerosol and cloud acidity. This is a much needed review, as the studies concerning aerosol acidity has increased. I applaud the large author list on this work, as it is very extensive and impressive. The review provides a much needed discussion concerning various aspects of cloud and aerosol pH, including definition, measurements and calculations, observations, and comparisons with models. I foresee this paper becoming an important source, both in the field and for introducing this subject in classes. This paper should be published upon consideration of the following comments:

We thank the reviewer for the supportive comments.

1.) Similar to Reviewer #1, occult is not a term that I am familiar with. If it could be defined, or a synonym could be used, that would be appreciated.

 Now defined in response to reviewer #1

2.) I appreciate Table 1, as there are numerous abbreviations throughout the manuscript. However, there appears to be some abbreviations missing, such as NVCs. Please include all abbreviations in the manuscript into Table 1, as it can be hard to find them in this large manuscript.

Appendix A has been expanded to include additional abbreviations. A second appendix (B) was added listing the models, datasets, and other related information. Items in appendix B are often abbreviations (e.g. WRF-Chem), but knowing their definition is not critical to the message of the paper. In some cases (e.g. ISORROPIA, SPECIATE) the name in B is not an abbreviation.

Updated Appendix A and new Appendix B:

Appendix A: Nomenclature

| Symbol | Description |
|--------|-------------|
| $a_{H^+}$ | The activity of hydrogen ions in aqueous solution on a molality basis |
| $a_{H^+}^{(c)}$ | The activity of $H^+$ ions on a molarity (concentration) basis |
| $a_{H^+}^{(x)}$ | The activity of $H^+$ ions on a mole fraction basis |
| $\alpha_{HSO4}$ | Fraction of $HSO_4^-$ dissociated into $H^+$ and $SO_4^{2-}$ |
| $a_i$ | The activity of species $i$ (usually molality-based for ions in aqueous solutions) |
| ACSM | Aerosol chemical speciation monitor |
| adiGR | Adjusted gas ratio (see Table 2) |
| ALPHA | Adapted Low-cost Passive High-Absorption |
| ALWC | Aerosol liquid water content (mass per volume of air) |
| AMS | Aerosol mass spectrometer |
| $c^{\ominus}$ | The standard state (unit) molarity |
| $c_{H^+}$ | The molarity or "molar concentration" of hydrogen ions in an aqueous solution (also written using square brackets as [$H^+$]) |
| CCN | Cloud condensation nuclei |
| CTM | Chemical transport model |
| DELTA | Denuder for Long Term Atmospheric sampling |
| DON | Degree of neutralization (see Table 2) |
| DSN | Degree of sulfate neutralization (see Table 2) |
| $f_{H^+}^*$ | The (rational) activity coefficient based on the mole fraction concentration scale |
| $F_{p,i}$ | Fraction of species $i$ in the particle vs particle + gas phase |
| FR | Flex Ratio, identifies the $NH_3$ emissions level at which the nitrate concentration switches from $NH_3$-insensitive (or negative sensitivity) to positive $NH_3$ sensitivity |
| $f_{H^+}^*$ | The (rational) activity coefficient based on the mole fraction concentration scale |
| GR | Gas ratio (see Table 2) |
| $H_{air}^+$ | Concentration of aerosol $H^+$ per volume of air (e.g., moles per m$^{-3}$ of air) |
| $H_{air,cb}^+$ | $H_{air}^+$ determined from charge balance (see Table 2) |

| | |
|---|---|
| $HO_x$ | Hydrogen oxides ($OH + HO_2$) |
| $K_a$ | Acid dissociation constant for $HX \rightleftharpoons H^+ + X^-$ |
| $K_b$ | Acid association constant for $H^+ + X^- \rightleftharpoons HX$ or $H^+ + X \rightleftharpoons XH^+$ |
| $K_H$ | Dimensionless Henry's law constant |
| $K_w$ | Activity-based equilibrium constant for the dissociation of water into $H^+$ and $OH^-$ (see Bandura and Lvov (2005) for tabulation of values) |
| LLPS | Liquid–liquid phase separation |
| KMT | Kinetic mass transfer |
| LLPS | Liquid–liquid phase separation |
| $m^\ominus$ | The standard state (unit) molality |
| $m_{H^+}$ | Molality of $H^+$ (mol $kg^{-1}$ solvent) |
| $M_w$ | Molar mass of water: 0.018015 kg $mol^{-1}$ |
| $n_i$ | Number (e.g., moles) of species $i$ |
| NEI | National Emission Inventory (for the United States) |
| $NO_x$ | Nitrogen oxides ($NO + NO_2$) |
| NVC | Non-volatile cations |
| PAHs | polycyclic aromatic hydrocarbons |
| PFASs | polyfluoroalkyl substances |
| PFSAs | Perfluoroalkyl sulfonic acidsperflur­oroalkane sulfonic acid |
| PFCAs | Perfluoroalkyl carboxylic acids |
| pH | Hydrogen ion potential with activity coefficient and concentration expressed on a molality concentration scale (see Table 1) |
| $pH_c$ | pH on a concentration (molarity) basis |
| $pH_x$ | pH on a mole fraction basis |
| $pH_T$ | "Total" pH based on the molality of sulfate and bisulfate ions (see Table 1) |
| $pH_F$ | "Free ion" approximation of pH obtained when the activity coefficient of $H^+$ is unity (see Table 1) |
| $pH_\pm$ $(H, X)$ | Approximation of pH using the mean molal ion activity coefficient of an $H^+$ and anion $X$ pair (see Table 1) |
| $pK_{wi}$ | $-\log_{10}$ of ($K_w K_i$) (see Bandura and Lvov (2005) for tabulation of values) |

| | |
|---|---|
| PM | Particulate matter, synonymous with aerosol |
| $PM_1$ | Particulate mass with an equivalent diameter below 1 μm |
| $PM_{2.5}$ | Particulate mass with an aerodynamic equivalent diameter below 2.5 μm |
| PSC | Pitzer-Simonson-Clegg (model) |
| $R$ | Universal gas constant |
| $r^2$ | Coefficient of determination |
| RH | Relative humidity |
| RRF | Relative response factor, relative change in concentration due to relative change in emission |
| $SO_x$ | Sulfur oxides (usually $SO_2 + TSO_4$) |
| $T$ | Temperature |
| TMI | Transition Metal Ions |
| TCl | Total chloride (sum of gas-phase hydrochloric acid and aerosol chloride) |
| $TNO_3$ | Total nitrate (sum of gas-phase nitric acid and particulate nitrate) |
| $TNH_4$ | Total ammonia (sum of gas-phase ammonia and particulate ammonium) |
| $TSO_4$ | Total particulate sulfate (sum of sulfate and bisulfate) |
| VOCs | Volatile organic compounds |
| WSOC | Water-soluble organic compounds |
| $x_{H^+}$ | The mole fraction of $H^+$ in the solution |
| $X_T$ | Molal sulfate ratio indicating sulfate- rich vs poor domain (Eq. 9) |
| $Z$ | Charge balance on total gas and particle phases used to estimate initial amount of $H^+$ (Eq. 19) |
| ZSR | Zdanovskii-Stokes-Robinson (method for calculation of aerosol water) |
| $\gamma_{H^+}$ | The molal activity coefficient of species $i$ |
| $\gamma_{H^+}^{(c)}$ | The molarity-based activity coefficient of $H^+$ |
| $\gamma_{\pm,HX}$ | The mean ion activity coefficient (for monovalent acid HX) |
| $\rho_0$ | The density of the reference solvent (water) |

*Added Feb 2020 to Appendix A:*

*IN     Ice nuclei*

**Appendix B: Models, data, and related methods discussed in the text**

| Type | Examples |
|---|---|
| Activity coefficient models | AIOMFAC, UNIFAC |
| Gas-particle thermodynamic models | AIOMFAC-GLE, ADDEM, E-AIM, EQSAM, EQUISOLV II, GFEMN, ISORROPIA II, MOSAIC, SCAPE, UHAERO |
| 3-D models | CAM-Chem, CESM, CMAQ, GEOS-Chem, GISS, PM-CAMx, TM4-ECPL, WRF-Chem |
| Supporting algorithms/chemistry/databases | ASTEM, CAM6, HETV, MAM, MESA, MOZART, MTEM, SPECIATE |
| Observational data sets (networks, satellites, field campaigns) | ACTRIS, AIRS, AMoN, CalNex, CASTNET, CrIS, CSN, EMEP/EBAS, IASI, IMPROVE, MAN, NAMN, SEARCH, SEAC4RS, SOAS, WINTER |

3.) I think a Table that summarizes Section 2.6 would improve the quality of the paper. This will be one of the many reasons people will want to read and cite the paper, concerning a discussion and comparison of the thermodynamic models in order to calculate pH. Including a table that summarizes how each model calculates pH, the pros and cons of each model, any assumptions, and etc., would help the readers remember that discussion better.

We added a table summarizing the four thermodynamic models used most extensively in the paper. One sentence was added to the first paragraph of section 2.6:

> Advantages and disadvantages of four common thermodynamic models are summarized in Table 2.

The following new table was added (*The italicized text was added Feb 2020 after posting the initial response to reviewer 2)*:

Table 2: Common box models used to calculate acidity.

| Model | Input | Acidity Output | Advantages | Disadvantages |
|---|---|---|---|---|
| E-AIM | Gas + particle or equilibrium particle composition ($H^+$, $NH_4^+$, $Na^+$, $SO_4^{2-}$, $HSO_4^-$, $NO_3^-$, $Cl^-$, $Br^-$, organic acids, amines) in moles in overall electroneutral conditions (see Eq. 19 for Z); RH, T. | pH at equilibrium | pH via recommended Eq. 1

Considered the most accurate inorganic thermodynamic model

Some ionizing organic species (e.g. organic acids, amines) considered | Computationally intensive

T and RH restricted for some compositions to preserve accuracy |
| AIOMFAC-GLE | Gas + particle or equilibrium particle composition ($H^+$, $Li^+$, $Na^+$, $K^+$, $NH_4^+$, $Mg^{2+}$, $Ca^{2+}$, $Cl^-$, $Br^-$, $NO_3^-$, $HSO_4^-$, $SO_4^{2-}$, organic species and/or organic functional groups) in mol m$^{-3}$ air for electroneutral conditions; RH, T | pH at equilibrium | pH via recommended Eq. 1

Accounts for organic--inorganic interactions and liquid-liquid equilibrium in consistent framework

Code publicly distributed through repository | Limited support for solid-liquid equilibria of diverse inorganic salts (presently)

Optimized for temperatures near 298 K, with limited accuracy for much colder atmospheric temperatures

*Organic species do not ionize (Feb 2020 update)* |
| MOSAIC | Distinct gas and particle composition ($H^+$, $NH_4^+$, $Na^+$, $Ca^{2+}$, $SO_4^{2-}$, $HSO_4^-$, $CH_3SO_3^-$, $NO_3^-$, $Cl^-$, and $CO_3^{2-}$) in mol m$^{-3}$ air; RH, and T. Automatic adjustments applied to non-electroneutral input particle-phase composition. | $pH_F$ by default ($pH_\pm$ with modification) for each particle size bin (or mode) at each time step while dynamically solving gas-particle mass transfer | Provides size-resolved $pH_F$ and $pH_\pm$ to account for compositional heterogeneity across particles of different sizes and origins

Does not require equilibrium assumption | Gas-particle and solid-liquid equilibrium constants depend on temperature, but activity coefficients are limited to 298.15 K. |
| ISORROPIA II | Gas + particle or particle composition (TSO$_4$, TCl, TNO$_3$, TNH$_4$, Na, K, Ca, Mg) in mol m$^{-3}$ or µg m$^{-3}$ air; RH, T. Automatic adjustments applied to non-electroneutral input particle-phase composition. | $pH_F$ by default ($pH_\pm$ with modification) at equilibrium | Computationally efficient

Code has widespread public distribution and incorporation in CTMs | Approximations employed (e.g. some activity coefficients treated as 1, minor species do not perturb equilibrium, *higher numerical tolerances (Feb 2020 update)*)

Segmented solution approach leads to discontinuous solution surface |

4.) Throughout the sections, it is apparent that different people wrote them with different styles coming through in each section. For example, some sections briefly state future research while other sections devote a subsection about future research (and some sections do not have any discussion about questions/future work). Starting with at least Section 3, if not some aspects of Section 2 (i.e., Measuring pH), it would be beneficial for the authors to include a description of what they consider future questions/work to be.

In some cases, concepts span multiple sections. For example, proxies were introduced in section 4, but the box model intercomparison in section 6 had summarizing messages. We have reorganized the paper to improve the flow/connection throughout. The new order will be:

      i.     Introduction
     ii.     Definition
    iii.     Proxies (former 4)
    iv.     Box model comparison & w/proxies (former 6)
     v.     Role of processes: Aqueous Chemistry (former 3)
    vi.     Role of size, composition, mass transfer (former 7)
   vii.     Observations (former 5)
  viii.     CTM results
    ix.     Conclusions

This moves the box model intercomparison (*iv)* immediately after definitions of pH and proxies. The box model intercomparison contains a recap of concepts introduced in the preceding two sections (in Section 4.3: Recommendations on the calculation of pH by approximation and proxy) and thus serves as a synthesis of sections *ii-iv*.

New section *v* (old section 3) covers the role of aqueous chemistry. Since the section is already short, summary in nature, and the subject of a companion paper, an additional summary paragraph was not added.

For new Section *vi* (old 7) on size, composition, mass transfer, the last two subsections end with specific conclusions regarding the roles of size, composition, mass transfer, and organics/LLPS. No further conclusions/future directions were added.

Section *vii* (old 5/Observations) is divided into section 7.1 (aerosols) and 7.2 (clouds). Section 7.2 already had a section on the need for future monitoring of cloud pH. Section 7.2.4 was promoted up to 7.3 and expanded to include aerosol pH:

> **7.3**     Need for future monitoring of cloud and aerosol pH
>
> Although cloud and fog sampling is generally more challenging than aerosol collection, pH measurement of the collected cloud/fog water is simpler due to its much larger volume and much lower ionic strength. As a result,  fogs and clouds have been sampled and their pH determined in areas around the globe with more temporal and spatial coverage than for aerosol pH. Depending on inputs of key acids and bases, cloud/fog pH has been observed to range from below 2 to greater than 7, slightly higher, but similar to fine aerosol pH that ranges from below 0 to near 7. Programs designed to target reductions in acid rain have had direct impacts on cloud and fog pH ., but aerosol pH has been much more constant than cloud pH in the southeastern US and southeastern Canada over time. Analysis of cloud pH observations over the past 25 – 30 years reveals that cloud/fog acidity in many regions has decreased as anthropogenic emissions of the important acid precursors, $SO_2$ and $NO_x$, have decreased. A continued rise in cloud/fog pH is likely in many regions with planned, future decreases in NOx and $SO_2$ emissions and stable or increasing $NH_3$ emissions. Future changes in emissions could eventually be significant enough to lead to fine aerosol pH changes as well. Increases in cloud pH are expected to enhance the solubility of gas phase organic acids, potentially shortening their atmospheric lifetimes. while increases in aerosol pH could lead to more nitrate aerosol formation and allow previously unfavorable kinetic reactions to occur.

As emissions evolve with time, continued characterization of cloud and particle pH is needed to understand how anthropogenic activities affect condensed-phase acidity and downstream endpoints in the earth system. Much remains to be learned about factors controlling cloud/fog pH in the atmosphere and the influence of this acidity on aqueous phase chemistry, including the aqueous phase uptake and oxidation of soluble gases to form secondary inorganic or organic aerosol. More detailed measurements of organic acids and bases, and their influence on cloud pH, will be increasingly important as sulfate and nitrate concentrations decline. Likewise, there is a need for more systematic monitoring of cloud and fog composition in key environments, as opposed to the more ad hoc past sampling approaches driven primarily by the objectives of process-based research. Because fogs and clouds are good integrators of atmospheric acids and bases in both the gas and particle phases, they may offer a convenient and practical basis for ongoing monitoring of atmospheric acidity. Future monitoring strategies should consider long-term monitoring at surface sites as well as periodic measurements of cloud, particle, and gas-phase composition from aircraft in order to enhance our understanding of acidity at higher elevations in the troposphere. Future measurements should also better document heterogeneity of acidity across individual drops within a cloud or/ fog or aerosol population, for example looking atby determining the size-dependence of drop pH. Aerosol pH estimates will likely continue to be primarily based on thermodynamic models in the near future and thus require simultaneous particle- and gas-phase measurements (specifically of ammonia) to improve the spatial and temporal scales over which fine particle pH is currently characterized.

Section *viii* (CTM predictions) already contains a summary/future directions section.

Multiple minor changes were made throughout the manuscript for the new section order (will be provided in the future tracked changes document).

5.) For Section 3, it would be good to include a couple of items in discussion, including: (a) How many of the reactions have been conducted for dilute, laboratory conditions, therefore, for aerosol, where the ion activity is higher and water is lower, there is large uncertainty in how the reactions may occur. (b) How there is debate occurring the field about various reactions (e.g., production of sulfate in aerosol in China) and the questions/future studies needed to move this questions forward. (c) N2O5 chemistry appears to be missing in your discussion throughout, including in Section 5.

The current paper is already quite long (~140 pages). The topic of kinetic drivers of pH and how pH affects kinetics is a large topic and warrants a separate companion paper. Former section 3 (now 5) was meant to highlight some examples of the kinetics-pH interplay, but not intended to be comprehensive. We will defer the bulk of these reviewer suggestions to the companion paper. However, we added mention of $N_2O_5$ in the introduction along with two references:

> The concentration of fine particulate matter ($PM_{2.5}$) is directly modulated by pH through its effects on gas–particle partitioning,  pH-dependent condensed-phase reactions, and other particle processes influenced by pH. For example, $N_2O_5$ heterogeneous hydrolysis significantly affects tropospheric chemistry (Dentener and Crutzen, 1993) and depends strongly on particle composition (Chang et al., 2011), including formation of organic coatings due to liquid-liquid phase separation influenced by acidity (see Section 6.3 for a discussion of phase separation in the context of acidity).
>
> Chang, W. L., Bhave, P.V., Brown, S.S., Riemer, N., Stutz, J., and Dabdub, D.: Heterogeneous atmospheric chemistry, ambient measurements, and model calculations of $N_2O_5$: A review, Aerosol Sci. Technol., 45, 6665-6695, https://doi.org/10.1080/02786826.2010.551672, 2011.
>
> Dentener, F. J., and Crutzen, P. J.: Reaction of $N_2O_5$ on tropospheric aerosols: Impact on the global distributions of $NO_x$, $O_3$, and OH, J. Geophys. Res., 98(D4), 7149– 7163, https://doi.org/10.1029/92JD02979, 1993.

6.) For Section 8, a table that summarizes the CTMs with the thermodynamic models they use and the species they use to calculate pH would help with the discussion.

We added a new table, Table 7, that summarizes the CTM calculations of pH. We included models used in this work only (rather than characterizing all CTMs in literature). A sentence was added to the first paragraph of Section 8:

> Table 7 summarizes the species considered in the calculation of pH for each model displayed in this work.

The following new table was added:

Table 7: Species and methods used to calculate acidity in CTMs. Bulk cloudwater pH is calculated assuming electroneutrality, generally using model-specific algorithms. Dissolved gases in cloudwater are determined using Henry's law coefficients. Configurations are specific to this work.

| Model | Aerosol size information | Species/sources considered in aerosol pH calculation | Fine aerosol pH calculation method | Species/sources considered in cloud pH calculation |
|---|---|---|---|---|
| CMAQ v5.3 | Fine aerosol: explicit Aitken and accumulation modes.

Coarse mode acidity not explicitly calculated but included in determination of dynamic mass transfer and composition. | $TSO_4$, $TCl$, $TNO_3$, $TNH_4$, Na, K, Ca, Mg from sea salt, dust, wildland fires, and anthropogenic activities. | ISORROPIA II $pH_F$ for inorganic-only composition of combined fine modes.

Condensed water associated with organic species is also predicted (not considered in fine aerosol $pH_F$ in this work). | Aqueous species: $H^+$, $OH^-$, $HSO_3^-$, $SO_3^{2-}$, $HSO_4^-$, $SO_4^{2-}$, $HCO_3^-$, $CO_3^{2-}$, $HCO_2^-$, $NH_4^+$, $NO_3^-$, $Cl^-$, $Ca^{2+}$, $Na^+$, $K^+$, $Mg^{2+}$

Dissolved gases: $SO_2$, $CO_2$, $NH_3$, HCl, $HNO_3$, HCOOH, $H_2SO_4$ (as sulfate), $N_2O_5$ (as $2 \times HNO_3$) |
| GEOS-Chem v12.0.0 | Bulk fine aerosol.

Coarse mode acidity not explicitly calculated but included in determination of dynamic mass transfer and composition. | $TSO_4$, HCl, $TNO_3$, $TNH_4$, and fine mode Ca, Mg, Na, Cl from anthropogenic, sea salt, and dust sources (dust contributions not considered in default GEOS-Chem predictions but Ca and Mg from dust considered in this work). | ISORROPIA II $pH_F$. | Aqueous species: $SO_4^{2-}$, $NO_3^-$, $NH_4^+$

Dissolved gases: $CO_2$, $SO_2$, $NH_3$, $HNO_3$ |
| TM4-ECPL | Fine (externally mixed dust) and coarse (internally mixed dust) aerosol. | $SO_4^{2-}$, $NH_3$, $NH_4^+$, $HNO_3$ and $NO_3^-$; sea salt and dust assumed to be externally mixed with fine mode sulfate and not considered in the fine acidity calculation. | ISORROPIA II $pH_F$ for inorganic-only composition of fine and coarse modes (each in equilibrium with gas).

Condensed water associated with organic species is also predicted (not considered in fine aerosol $pH_F$ in this work). | Aqueous species: $SO_4^{2-}$, $CH_3O_3S^-$, $NO_3^-$, $NH_4^+$, $Na^+$, $Ca^{2+}$, $K^+$, $Cl^-$ $Mg^{2+}$

Dissolved gases: $SO_2$, $CO_2$, $HNO_3$, $NH_3$, oxalic acid |
| WRF-Chem | Four aerosol size bins (0.039–0.156, 0.156–0.625, 0.625–2.5, 2.5–10 μm in diameter) treated dynamically. | sulfate, $HNO_3/NO_3^-$, $NH_3/NH_4^+$, $CH_3O_3S^-$, $Cl^-$, $CO_3^{2-}$, Na, Ca; HCl not considered with MOZART chemistry (no displacement of $Cl^-$ from sea salt aerosols allowed). | MOSAIC size-resolved $pH_F$. | Aqueous species: $OH^-$, $HCO_3^-$, $CO_3^{2-}$, $CO_3^-$, $HSO_3^-$, $SO_3^{2-}$, $HSO_4^-$, $SO_4^{2-}$, $SO_4^-$, $SO_5^-$, $HSO_5^-$, $HOCH_2SO_3^-$, $^-OCH_2SO_3^-$, $NO_2^-$, $NO_3^-$, $HO_2^-$, $O_2^-$, $HCOO^-$, $Cl^-$, $Cl_2^-$, $ClOH^-$, $NH_4^+$, $Fe^{3+}$, $Mn^{2+}$

Dissolved gases: $SO_2$, $CO_2$, $HNO_3$, $NH_3$, $HO_2$, HCOOH, $H_2O_2$ |
| CAM-Chem | Four log-normal modes. | Inorganic aerosol composition considered: $SO_4^{2-}$, $NH_4^+$, soil dust, sea salt. | Not considered in this work. | Aqueous species: $OH^-$, $HCO_3^-$, $NO_3^-$, $HSO_3^-$, $SO_3^{2-}$, $SO_4^{2-}$, $NH_4^+$

Dissolved gases: $SO_2$, $H_2SO_4$, $HNO_3$, $CO_2$, $NH_3$ |

7.) For Section 9, I appreciate that it summarizes the very large review. However, at this point, I really think a description of remaining questions, studies, observations and future outlook is necessary so that we, as a community, know what should be done to move forward.

The synthesized messages in section 9 were developed at a workshop involving the coauthors of this study. Coauthors submitted their thoughts on major messages ahead of discussion then messages were discussed, refined, and agreed upon by the group. As a result, we think the most important major messages (which include both summary information and future directions) have been captured. The individual sections contain additional information. Some future directions (e.g. understanding/improving bisulfate dissociation predictions to improve consistency among box models) were raised and discussed by the group but considered too detailed for the main messages.

We think that section 9 already provides guidance to the community. The following guidance is contained in section 9:

- We recommend researchers use specific nomenclature to document and communicate what metric of acidic they report
- pH is the ideal indicator of acidity and researchers should aim to report that value or an approximation ($pH_\pm$ is best approximation)
- The role of kinetic-pH interactions is likely underappreciated and should be further examined to understand where $H^+$ is chemically generated (this is discussed in the companion paper)
- Experimental determination of aerosol pH is a current knowledge gap
- Heterogeneity of pH across the aerosol/cloud droplet population and within a given particle needs investigation (most methods/models use bulk techniques)
- Ammonia measurements are needed to facilitate determination of aerosol pH
- Ambient characterization of pH is spatially and temporally incomplete (more observations or observationally constrained estimates are needed-Section 7 highlights current areas without measurements and where trends are available).
- pH should be considered in the context of CTM evaluation and endpoints of interest.
- Considerable model diversity in predicted pH exists and Section 8 points to mixing state assumptions and composition as reasons and locations where model diversity is high and could be measured

**Other Revisions**

1. A few studies reporting observationally-constrained estimates of fine aerosol pH were added to SI Table S6 and fully synchronized with Fig 2b (global distribution histogram) and Fig 14 (map). In cases where only a min and max fine aerosol $pH_F$ value was provided by an observational study (Table S6), a mean value was created by averaging the min and max (revisions in Table S6). When the min/max were averaged, it was noted in the Method column (see tracked changes). Additional cleanup in Table S6 included state abbreviations and minor lat/lon adjustments. We added one sentence regarding new data:

   However, more acidic particles (pH ranging from -0.8 to 3.0) have been observed near the Kilauea volcano in Hawaii (Kroll et al., 2015).

   Data added to SI Table S6:

| Location | Altitude (m) | Lati-tude (°N) | Lati-tude (°E) | Time | Aerosol Size | n | Mean (pH) | σ (pH) | Min (pH) | Max (pH) | Method | Reference |
|---|---|---|---|---|---|---|---|---|---|---|---|---|
| Pellston, Michigan, USA | | 45.56 | -84.86 | Jul-16 | <0.4 μm | | 1.5 | | | | pH paper | Craig et al. 2018 |
| Near Kilauea, Hawaii, USA | | 19.2026-19.4327 | 155.4775-155.2739 | Jan - Feb 2013 | PM1 | | 1.1 | | -0.8 | 3.0 | E-AIM II with ACSM measured composition | Kroll et al., 2015 |
| Tianjin | | 39.11 | 117.16 | 12-23 August 2015 | PM2.5 | 387 | 3.4 | 0.5 | 2.6 | 4.6 | ISORROPIA (forward, metastable) | Shi et al., 2019 |
| Po Valley Italy | mean of 6 sites | 45.4 | 12.2 | winter 2012-2013 | PM2.5 | | 3.9 | 0.3 | | | ISORROPIA (forward, metastable, no NH3) | Masiol et al. 2020 |
| Po Valley Italy | mean of 6 sites | 45.4 | 12.2 | summer 2012 | PM2.5 | | 2.3 | 0.3 | | | ISORROPIA (forward, metastable, no NH3) | Masiol et al. 2020 |

2. The Barbados value in Fig 14 was updated to reflect an average for 0.4 to 1.7 μm size particles rather than only 0.4 to 0.8 μm.
3. A reference (Ding, J., et al, Atmos. Chem. Phys., 2019) was updated with the correct title: 'Aerosol pH and its driving factors in Beijing'.
4. SI references were formatted and cleaned up (including addition of DOIs and a few missing references).
5. We added an acknowledgement to Chris Nolte and Donna Schwede for their helpful comments during EPA internal review.
6. The number of significant digits reported in the main text tables was reduced.
7. References to discussion papers were updated to final versions.
8. Some abbreviations were cleaned up in the text (along with appendix revisions in response to reviewer 2).
9. Minor grammatical edits.
10. Figure 8 (comparison of acidity with proxies using field data) panel order was revised to follow order of discussion in text (minor formatting changes).
11. Strong acidity terminology was clarified. Now, charge/ion balance is used for the quantity in Table 2, Figure 8, and the primary term used in the text for summation over measured anions and cations. The order of some paragraphs was revised in proxy Section 3 as well as some minor wording edits.

12. A sentence in the conclusions, "This is a direct consequence of the liquid water content and other aerosol species being in equilibrium with the ambient relative humidity – while in clouds all the species can vary independently of each other," was reworded for clarity. Now: "This is a direct consequence of the difference in liquid water content which is higher in clouds than fine aerosols."
13. Some numbers in the introduction of idealized systems for box model pH calculation in 4.1.1 were not synced with Fig. 3. Section 4.1.1 text updated to match Fig. 3.
14. Multiple observations are available for Tianjin. Figure 5 (diurnal variation) used Shi et al. 2019 not Shi et al. 2017 as was labeled. Caption revised.
15. Figure 5 was updated (formatting and correction to plotting error for Atlanta data).
16. Equation S11 was updated (pH appeared on wrong side) and a typo in Section S1 corrected.
17. Note: For the purposes of tracked changes, section order was updated before changes were recorded to avoid large sections appearing as edits solely due to movement of text.

[revised manuscript text omitted]

Font: 10 pt

| Page 132: [1] Formatted | Pye, Havala | 12/16/2019 4:24:00 PM |

Font: 10 pt

| Page 132: [1] Formatted | Pye, Havala | 12/16/2019 4:24:00 PM |

Font: 10 pt

| Page 132: [2] Formatted | Pye, Havala | 12/16/2019 4:30:00 PM |

Font: 10 pt, Superscript

| Page 132: [2] Formatted | Pye, Havala | 12/16/2019 4:30:00 PM |

Font: 10 pt, Superscript

| Page 132: [3] Formatted | Pye, Havala | 12/16/2019 4:31:00 PM |

Font: 10 pt, Superscript

| Page 132: [3] Formatted | Pye, Havala | 12/16/2019 4:31:00 PM |

Font: 10 pt, Superscript

| Page 132: [4] Formatted | Pye, Havala | 12/16/2019 4:24:00 PM |

Font: 10 pt

| Page 132: [4] Formatted | Pye, Havala | 12/16/2019 4:24:00 PM |

Font: 10 pt

| Page 132: [4] Formatted | Pye, Havala | 12/16/2019 4:24:00 PM |

Font: 10 pt

| Page 132: [5] Formatted | Pye, Havala | 12/16/2019 4:30:00 PM |

Font: 10 pt, Superscript

| Page 132: [5] Formatted | Pye, Havala | 12/16/2019 4:30:00 PM |

Font: 10 pt, Superscript

| Page 132: [6] Formatted | Pye, Havala | 12/16/2019 4:31:00 PM |

Font: 10 pt, Superscript

| Page 132: [6] Formatted | Pye, Havala | 12/16/2019 4:31:00 PM |

Font: 10 pt, Superscript

| Page 132: [7] Formatted | Pye, Havala | 12/16/2019 4:30:00 PM |

Font: 10 pt, Superscript

| Page 132: [7] Formatted | Pye, Havala | 12/16/2019 4:30:00 PM |

Font: 10 pt, Superscript

| **Page 132: [8] Formatted** | **Pye, Havala** | **12/16/2019 4:31:00 PM** |

Font: 10 pt, Superscript

| **Page 132: [8] Formatted** | **Pye, Havala** | **12/16/2019 4:31:00 PM** |

Font: 10 pt, Superscript

| **Page 132: [9] Formatted** | **Pye, Havala** | **12/16/2019 4:31:00 PM** |

Font: 10 pt, Superscript

| **Page 132: [9] Formatted** | **Pye, Havala** | **12/16/2019 4:31:00 PM** |

Font: 10 pt, Superscript

| **Page 132: [10] Formatted** | **Pye, Havala** | **12/16/2019 4:31:00 PM** |

Font: 10 pt, Superscript

| **Page 132: [10] Formatted** | **Pye, Havala** | **12/16/2019 4:31:00 PM** |

Font: 10 pt, Superscript

| **Page 132: [11] Formatted** | **Pye, Havala** | **12/16/2019 4:30:00 PM** |

Font: 10 pt, Superscript

| **Page 132: [11] Formatted** | **Pye, Havala** | **12/16/2019 4:30:00 PM** |

Font: 10 pt, Superscript

| **Page 132: [12] Formatted** | **Pye, Havala** | **12/16/2019 4:31:00 PM** |

Font: 10 pt, Superscript

| **Page 132: [12] Formatted** | **Pye, Havala** | **12/16/2019 4:31:00 PM** |

Font: 10 pt, Superscript

| **Page 132: [13] Formatted** | **Pye, Havala** | **12/16/2019 4:31:00 PM** |

Font: 10 pt, Superscript

| **Page 132: [13] Formatted** | **Pye, Havala** | **12/16/2019 4:31:00 PM** |

Font: 10 pt, Superscript

| **Page 132: [14] Formatted** | **Pye, Havala** | **12/16/2019 4:31:00 PM** |

Font: 10 pt, Superscript

| **Page 132: [14] Formatted** | **Pye, Havala** | **12/16/2019 4:31:00 PM** |

Font: 10 pt, Superscript

| **Page 132: [15] Formatted** | **Pye, Havala** | **12/16/2019 4:24:00 PM** |

Font: 10 pt

| **Page 132: [15] Formatted** | **Pye, Havala** | **12/16/2019 4:24:00 PM** |

Font: 10 pt

| Page 132: [15] Formatted | Pye, Havala | 12/16/2019 4:24:00 PM |

Font: 10 pt

| Page 132: [16] Formatted | Pye, Havala | 12/16/2019 4:30:00 PM |

Font: 10 pt, Superscript

| Page 132: [16] Formatted | Pye, Havala | 12/16/2019 4:30:00 PM |

Font: 10 pt, Superscript

| Page 132: [17] Formatted | Pye, Havala | 12/16/2019 4:31:00 PM |

Font: 10 pt, Superscript

| Page 132: [17] Formatted | Pye, Havala | 12/16/2019 4:31:00 PM |

Font: 10 pt, Superscript

| Page 132: [18] Formatted | Pye, Havala | 12/16/2019 4:31:00 PM |

Font: 10 pt, Superscript

| Page 132: [18] Formatted | Pye, Havala | 12/16/2019 4:31:00 PM |

Font: 10 pt, Superscript

| Page 132: [19] Formatted | Pye, Havala | 12/16/2019 4:24:00 PM |

Font: 10 pt

| Page 132: [19] Formatted | Pye, Havala | 12/16/2019 4:24:00 PM |

Font: 10 pt

| Page 132: [19] Formatted | Pye, Havala | 12/16/2019 4:24:00 PM |

Font: 10 pt

| Page 132: [20] Formatted | Pye, Havala | 12/16/2019 4:31:00 PM |

Font: 10 pt, Superscript

| Page 132: [20] Formatted | Pye, Havala | 12/16/2019 4:31:00 PM |

Font: 10 pt, Superscript

| Page 132: [21] Formatted | Pye, Havala | 12/16/2019 4:31:00 PM |

Font: 10 pt, Superscript

| Page 132: [21] Formatted | Pye, Havala | 12/16/2019 4:31:00 PM |

Font: 10 pt, Superscript

| Page 132: [22] Formatted | Pye, Havala | 12/16/2019 4:31:00 PM |

Font: 10 pt, Superscript

| Page 132: [22] Formatted | Pye, Havala | 12/16/2019 4:31:00 PM |

Font: 10 pt, Superscript

[revised manuscript text omitted]
 $|$ KCl(aq), $\left(m_{KCl} \geq 3.5 \text{ mol kg}^{-1}\right)$ $\vdots\vdots$ solution (S or X) $|$ glass electrode.     (C2)

The reference electrode is usually of the silver–silver chloride type, with a salt bridge, e.g. a porous plug junction in contact with the solution to be tested. The working principle of a glass electrode is based on the development of an electrical potential at H$^+$-sensitive glass–liquid interfaces. The potential at the outside glass surface depends on the pH

of the sample solution measured, while the potential at the inside surface is established by the constant pH of the filling solution (e.g. concentrated KCl$_{(aq)}$). Sometimes the glass and reference electrodes are combined into a single- probe *combination electrode*.

The pH of sample solution X is determined via the measured potential difference $E_2(X) - E_2(S)$ using an adequate standard buffer solution of known pH(S),

$$\quad \text{pH(X)} = \text{pH(S)} - \frac{E_2(X) - E_2(S)}{(RT/F)\ln(10)}. \quad (S6)$$

The direct application of Eq. (S6) is an example of a simple one-point calibration. Higher precision measurements are
carried out by using a two-point or multi-point calibration procedure, in which at least two standard buffers are used
that bracket the (unknown) pH(X). Ideally, the standard buffers chosen are close (above and below) in pH to pH(X),
leading to reduced uncertainties. Two-point or multipoint calibrations are needed to achieve a target pH uncertainty
of about 0.02 – 0.03 near 25 °C. Details about such methods, proper instrument calibration procedures and associated
uncertainties are outlined in Buck et al. (2002) as well as the manuals and guidelines of commercial pH-meter
manufacturers. Moreover, these references point out that special considerations are necessary for pH measurements
in non-aqueous solutions or solutions containing substantial amounts of organic components, which may affect the
behavior of the electrodes and junctions.

### S1.2  Derivation of pH scale conversions

Conversions among pH values calculated using different concentration scales (molarity, molality, mole fraction, etc.)
are possible and necessary for an adequate comparison of model predictions. It is recommended to convert all pH
values to the molality scale. For clarity, the molality-scale pH is denoted by symbol "pH" while the pH on other scales
is indicated by a subscript (e.g. $\mathrm{pH}_x$ for the mole-fraction-based pH). Generally, formulas for the conversion of pH
scales are derived using the equivalence of the (electro-)chemical potential of single ions expressed in any
concentration scale. For example, in the case of an electroneutral liquid phase (i.e. cancellation of the local electrostatic
potential within the phase), the chemical potential of $\mathrm{H}^+$, $\mu_{\mathrm{H}^+}^l$, is given by

$$\mu_{\mathrm{H}^+}^l = \mu_{\mathrm{H}^+}^{\ominus,(m)} + RT \ln\left(\frac{m_{\mathrm{H}^+}}{m^{\ominus}}\gamma_{\mathrm{H}^+}\right) = \mu_{\mathrm{H}^+}^{\ominus,(x)} + RT \ln\left(x_{\mathrm{H}^+} f_{\mathrm{H}^+}^*\right). \tag{
[revised manuscript text omitted]

$K_2 = 150 \text{ M}^{-2/3}$
$k_3 = 10^9 \text{M}^{-2} \text{ s}^{-1}$
$k_4 = 10^{-3} \text{ s}^{-1}$
$k_5 = 10^3 \text{ M}^{-1}\text{s}^{-1}$
$k_6 = 680 \text{ M}^{-1}\text{s}^{-1}$ | Martin and Hill (1987); Martin et al. (1991) |
| $R_{O3} = \left(k_7[SO_2 \cdot H_2O] + k_8[HSO_3^-] + k_9[SO_3^{-2}]\right)[O_3]$ | $k_7 = 2.4 \times 10^4$ M⁻¹s⁻¹
$k_8 = 3.7 \times 10^5 \exp\left(-5530\left(\frac{1}{T} - \frac{1}{298}\right)\right)$ M⁻¹s⁻¹
$k_9 = 1.5 \times 10^9 \exp\left(-5280\left(\frac{1}{T} - \frac{1}{298}\right)\right)$ M⁻¹s⁻¹ | Hoffmann (1986) |
| $R_{NO2} = k_{10}[NO_2][S(IV)]$ | $k_{10}$
$= \begin{cases} 1.4 \times 10^5 \text{ M}^{-1}\text{s}^{-1} \text{ for } pH < 5.3 \\ 1.24 \times 10^7 \text{ M}^{-1}\text{s}^{-1} \text{ for } pH = 5.3 \\ 1.6 \times 10^7 \text{ M}^{-1}\text{s}^{-1} \text{ for } pH = 8.7 \\ linear\ interpolation\ for\ 5.3 < pH < 8.7 \end{cases}$ | Lee and Schwartz (1983); Clifton (1988); Cheng et al. (2016) |

**Table S2.** Henry's Law constants (Sander 2015) used for the S(IV) –S(VI) conversion processes in Figure 3.

| Henry's Law Constant | Value |
|---|---|
| $H_{SO2}$ | $1.317225 \exp\left(2900\left(\frac{1}{T}-\frac{1}{298}\right)\right) \text{M atm}^{-1}$ |
| $H_{NO2}$ | $1.22 \times 10^{-2} \exp\left(2400\left(\frac{1}{T}-\frac{1}{298}\right)\right) \text{M atm}^{-1}$ |
| $H_{O3}$ | $1.01 \times 10^{-2} \exp\left(2800\left(\frac{1}{T}-\frac{1}{298}\right)\right) \text{M atm}^{-1}$ |
| $H_{HOOH}$ | $8.41 \times 10^{4} \exp\left(7600\left(\frac{1}{T}-\frac{1}{298}\right)\right) \text{M atm}^{-1}$ |

**Table S3.** Predictions of molality-based pH and related $H^+$ properties for system 1, water $+$ $(NH_4)_2SO_4 + H_2SO_4 +$
$NH_3$ at 298.15 K. Model calculations include the partial dissociation of $HSO_4^-$ and the gas–liquid equilibria of $NH_3$
and water.

[revised manuscript text omitted]

**Formatted Table**

[revised manuscript text omitted]

---

## Author Response (AR2)

Two technical corrections were made:

1. Page 34, line 24: reference is now to Fig. 6 (not 13).

2.The following was added to footnote (1) of Table 6:

RMSE and MB are calculated as follows: $\text{RMSE} = \sqrt[2]{\frac{1}{N}\sum_{j=1}^{N}\left(\text{pH}_{approx,j} - \text{pH}_j\right)^2}$,

$\text{MB} = \frac{1}{N}\sum_{j=1}^{N}\left(\text{pH}_{approx,j} - \text{pH}_j\right)$, where $N$ denotes the number of data points within the evaluated data set and where $\text{pH}_{approx,j}$ and $\text{pH}_j$ are the pH approximation and reference values of data point $j$.